# Learning One-hidden-layer Neural Networks on Gaussian Mixture Models with Guaranteed Generalizability

## Abstract

We analyze the learning problem of fully connected neural networks with the sigmoid activation function for binary classification in the teacher-student setup, where the outputs are assumed to be generated by a ground-truth teacher neural network with unknown parameters, and the learning objective is to estimate the teacher network model by minimizing a non-convex cross-entropy risk function of the training data over a student neural network. This paper analyzes a general and practical scenario that the input features follow a Gaussian mixture model of a finite number of Gaussian distributions of various mean and variance. We propose a gradient descent algorithm with a tensor initialization approach and show that our algorithm converges linearly to a critical point that has a diminishing distance to the ground-truth model with guaranteed generalizability. We characterize the required number of samples for successful convergence, referred to as the sample complexity, as a function of the parameters of the Gaussian mixture model. We prove analytically that when any mean or variance in the mixture model is large, or when all variances are close to zero, the sample complexity increases, and the convergence slows down, indicating a more challenging learning problem. Although focusing on one-hidden-layer neural networks, to the best of our knowledge, this paper provides the first explicit characterization of the impact of the parameters of the input distributions on the sample complexity and learning rate.

## 1 Introduction

Deep neural networks (LeCun et al., 2015) have demonstrated superior empirical performance in various applications such as speech recognition (Krizhevsky et al., 2012) and computer vision (Graves et al., 2013; He et al., 2016). Despite the numerical success, the theoretical underpin of learning neural networks is much less investigated. One bottleneck for the wide acceptance of deep learning in critical applications is the lack of the theoretical generalization guarantees, i.e., why a model learned from the training data would achieve a high accuracy on the testing data.

This paper studies the generalization performance of neural networks in the "teacher-student" setup, where the training data are generated by a teacher neural network, and the learning is performed on a student network by minimizing the empirical risk of the training data. This teacher-student setup has been studied in the statistical learning community for a long time (Engel & Broeck, 2001; Seung et al., 1992) and applied to neural networks recently (Goldt et al., 2019a; Zhong et al., 2017b;a; Zhang et al., 2019; 2020b; Fu et al., 2020; Zhang et al., 2020a). Assuming that the student network has the same architecture as the teacher network, the existing generalization analyses mostly focus on one-hidden-layer networks, because the optimization problem is already nonconvex, and the analytical complexity increases tremendously when the number of hidden layers increases.

One critical assumption of most works in this line is that the input features follow the standard Gaussian distribution. Although other distributions are considered in (Du et al., 2017; Ghorbani et al., 2020; Goldt et al., 2019b; Li & Liang, 2018; Mei et al., 2018b; Mignacco et al., 2020; Yoshida & Okada, 2019), the generalization performance beyond the standard Gaussian input is less investigated. On the other hand, the learning performance clearly depends on the input data distribution. (LeCun et al., 1998) states that the learning method converges faster if the inputs are whitened to

be the standard Gaussian. Batch normalization (Ioffe & Szegedy, 2015) modifies the mean and variance in each layer and is a popular practical method to achieve a fast and stable convergence. Various explanations such as (Bjorck et al., 2018; Chai et al., 2020; Santurkar et al., 2018) have been proposed to explain the enormous success of Batch normalization, but little consensus exists on the exact mechanism.

**Contributions**: This paper provides a theoretical analysis of learning one-hidden-layer neural networks when the input distribution follows a Gaussian mixture model containing an arbitrary number of Gaussian distributions with arbitrary mean and variance. The Gaussian mixture model has been employed in many applications such as data clustering and unsupervised learning (Dasgupta, 1999; Figueiredo & Jain, 2002; Jain, 2010), and image classification and segmentation (Permuter et al., 2006). The parameters of the mixture model can be estimated from data by the EM algorithm (Redner & Walker, 1984) or the moment-based method (Hsu & Kakade, 2013), with theoretical performance guarantees, see, e.g., (Ho & Nguyen, 2016; Ho et al., 2020; Dwivedi et al., 2020a;b).

For the binary classification problem with the cross entropy loss function, this paper proposes a gradient descent algorithm with tensor initialization to estimate the weights of the one-hidden-layer fully-connected neural network. Our algorithm converges to a critical point linearly, and the returned critical point converges to the ground-truth model at a rate of $\sqrt{d \log n / n}$, where $d$ is the dimension of the feature, and $n$ is the number of samples. We also characterize the required number of samples for accurate estimation, referred to as the sample complexity, as a function of $d$, the number of neurons $K$, and the input distribution. Our explicit bounds imply (1) when the absolute value of any mean in the Gaussian mixture model increases from zero, the sample complexity increases, and the algorithm converges slower, indicating that it will be more challenging to learn a model with a small test error; (2) The same phenomenon happens when any variance in the mixture model increases to infinity from a certain positive value, or if all the variances in the mixture model approach zero. Our results indicate that the training converges faster and requires a less number of samples if the input data are zero mean with a certain non-zero variance. This can be viewed as one theoretical explanation in one-hidden-layer for the success of Batch normalization. Moreover, to the best of our knowledge, *this paper provides the first theoretical and explicit characterization about how the mean and variance of the input distribution affect the sample complexity and learning rate*.

## 1.1 RELATED WORK

**Learning over-parameterized neural networks.** One line of theoretical research on the learning performance considers the over-parameterized setting where the number of network parameters is greater than the number of training samples. (Bousquet & Elisseeff, 2002; Hardt et al., 2016; Keskar et al., 2016; Livni et al., 2014; Neyshabur et al., 2017; Rumelhart et al., 1988; Soltanolkotabi et al., 2018; Allen-Zhu et al., 2019a). (Allen-Zhu et al., 2019b; Du et al., 2019; Zou & Gu, 2019) show the deep neural networks can fit all training samples in polynomial time. The optimization problem has no spurious local minima (Livni et al., 2014; Zhang et al., 2016; Soltanolkotabi et al., 2018), and the global minimum of the empirical risk function can be obtained by gradient descent (Li & Yuan, 2017; Du et al., 2018b; Zou et al., 2020). Although the returned model can achieve a zero training error, these works do not discuss whether it achieves a small test error or not. (Allen-Zhu et al., 2019a; Li & Liang, 2018) analyze the generalization error by characterizing the training error and test error separately. Still, there is no guarantee that a learned model with a small training error would have a small test error. (Cao & Gu, 2019) provides the bounds of the generalization error of the learned model by stochastic gradient descent (SGD) in deep neural networks, based on the assumption that there exists a good model with a small test error around the initialization of the SGD algorithm, and no discussion is provided about how to find such an initialization. In contrast, our tensor initialization method in this paper provides an initialization that is close to the ground-truth teacher model such that our algorithm can find this model with a zero test error.

**Generalization performance with the standard Gaussian input.** In the teacher-student setup of one-hidden-layer neural networks, (Brutzkus & Globerson, 2017; Du et al., 2018a; Ge et al., 2018; Liang et al., 2018; Li & Yuan, 2017; Shamir, 2018; Safran & Shamir, 2018; Tian, 2017) consider the ideal case of an infinite number of training samples so that the training and test accuracy coincide and can be analyzed simultaneously. When the number of training samples is finite, (Zhong et al., 2017b;a) characterize the sample complexity, i.e., the required number of samples, of learning one-hidden-layer fully connected neural networks with smooth activation functions and propose a

gradient descent algorithm that converges to the ground-truth model linearly. (Zhang et al., 2019; 2020b) extend the analyses to the non-smooth ReLU for fully-connected and convolutional neural networks, respectively. (Zhang et al., 2020a) analyzes the generalizability of graph neural networks for both regression and binary classification problems. (Fu et al., 2020) analyzes the cross entropy loss function for binary classification problems. Compared with other common loss functions such as the squared loss, the cross entropy loss function is harder to analyze due to the complicated forms and the saturation phenomenon of its Gradient and Hessian (Fu et al., 2020).

**Theoretical characterization of learning performance from other input distributions.** (Du et al., 2017) considers rotationally invariant distributions, but the results only apply to a perceptron (i.e., a single-node network). (Mei et al., 2018b) analyzes the generalization error of one-hidden-layer neural networks in the mean-field limit trained on a large class of distributions, including a mixture of Gaussian distributions with the same mean. The results only hold in the high-dimensional region where both the number of neurons $K$ and the input dimension $d$ are sufficiently large, and no sample complexity analysis is provided. (Li & Liang, 2018) studies the generalization error of over-parameterized one-hidden-layer networks when the data come from mixtures of well-separated distribution, but the separation requirement excludes Gaussian distributions and Gaussian mixture models. (Yoshida & Okada, 2019) analyzes the Plateau Phenomenon that the decrease of the risk slows down significantly partway and speeds up again in one-hidden-layer neural networks with inputs drawn from a single Gaussian with an arbitrary covariance. (Goldt et al., 2019b; 2020) analyze the dynamics of learning one-hidden-layer networks with SGD when the inputs are drawn from a wide class of generative models. (Mignacco et al., 2020) provides analytical equations for SGD evolution in a perceptron trained on the Gaussian mixture model. (Ghorbani et al., 2020) considers inputs with low-dimensional structures and compares neural networks with kernel methods.

**Notations**: Vectors are in bold lowercase, matrices and tensors in are bold uppercase. Scalars are in normal fonts. For instance, $\boldsymbol{Z}$ is a matrix, and $\boldsymbol{z}$ is a vector. $z_i$ denotes the $i$-th entry of $\boldsymbol{z}$, and $Z_{i,j}$ denotes the $(i, j)$-th entry of $\boldsymbol{Z}$. $[K]$ ($K > 0$) denotes the set including integers from 1 to $K$. $\boldsymbol{I}_d \in \mathbb{R}^{d \times d}$ and $\boldsymbol{e}_i$ represent the identity matrix in $\mathbb{R}^{d \times d}$ and the $i$-th standard basis vector, respectively. We use $\delta_i(\boldsymbol{Z})$ to denote the $i$-th largest singular value of $\boldsymbol{Z}$. $\boldsymbol{A} \succeq 0$ means $\boldsymbol{A}$ is a positive semi-definite (PSD) matrix. The gradient and the Hessian of a function $f(\boldsymbol{W})$ are denoted by $\nabla f(\boldsymbol{W})$ and $\nabla^2 f(\boldsymbol{W})$, respectively. The outer product of vectors $\boldsymbol{z}_i \in \mathbb{R}^{n_i}$, $i \in [l]$, is defined as $\boldsymbol{T} = \boldsymbol{z}_1 \otimes \cdots \times \boldsymbol{z}_l \in \mathbb{R}^{n_1 \times \cdots \times n_l}$ with $\boldsymbol{T}_{j_1 \cdots j_l} = (\boldsymbol{z}_1)_{j_1} \cdots (\boldsymbol{z}_l)_{j_l}$.
Given a tensor $\boldsymbol{T} \in \mathbb{R}^{n_1 \times n_2 \times n_3}$ and matrices $\boldsymbol{A} \in \mathbb{R}^{n_1 \times d_1}$, $\boldsymbol{B} \in \mathbb{R}^{n_2 \times d_2}$, $\boldsymbol{C} \in \mathbb{R}^{n_3 \times d_3}$, the $(i_1, i_2, i_3)$-th entry of the tensor $\boldsymbol{T}(\boldsymbol{A}, \boldsymbol{B}, \boldsymbol{C})$ is given by

$$\sum_{i_1'}^{n_1} \sum_{i_2'}^{n_2} \sum_{i_3'}^{n_3} \boldsymbol{T}_{i_1', i_2', i_3'} \boldsymbol{A}_{i_1', i_1} \boldsymbol{B}_{i_2', i_2} \boldsymbol{C}_{i_3', i_3}. \tag{1}$$

We follow the convention that $f(x) = O(g(x))$ (or $\Omega(g(x))$, $\Theta(g(x))$) means that $f(x)$ increases at most, at least, or in the order of $g(x)$, respectively.

## 2 PROBLEM FORMULATION

We consider a one-hidden-layer fully connected neural network where all the weights in the second layer have the same fixed value. This structure is also known as the committee machine, see, e.g., (Aubin et al., 2018; Monasson & Zecchina, 1995; Schwarze & Hertz, 1992; 1993). Let $\boldsymbol{x} \in \mathbb{R}^d$ denote the input features. Let $K \geq 1$ be the number of neurons in the hidden layer. Following the teacher-student setup, see e.g., (Fu et al., 2020), the output labels are generated by a teacher neural network with unknown ground-truth weights $\boldsymbol{w}_j^* \in \mathbb{R}^d$ ($j \in [K]$). Let $\boldsymbol{W}^* = [\boldsymbol{w}_1^*, ..., \boldsymbol{w}_K^*] \in \mathbb{R}^{d \times K}$ contain all the weights. Let $\delta_i(\boldsymbol{W}^*)$ denote the $i$-th largest singular value of $\boldsymbol{W}^*$. Let $\kappa = \frac{\delta_1(\boldsymbol{W}^*)}{\delta_K(\boldsymbol{W}^*)}$, and define $\eta_1 = \prod_{i=1}^{K} \frac{\delta_i(\boldsymbol{W}^*)}{\delta_K(\boldsymbol{W}^*)}$. The nonlinear activation function here is the sigmoid function $\phi(x) = \frac{1}{1+\exp(-x)}$. We consider binary classification, and the binary output $y$ is generated by the teacher committee machine through

$$\mathbb{P}(y = 1|\boldsymbol{x}) = H(\boldsymbol{W}^*, \boldsymbol{x}) := \frac{1}{K} \sum_{j=1}^{K} \phi(\boldsymbol{w}_j^{*\top} \boldsymbol{x}). \tag{2}$$

Learning is performed over a student neural network that has the same architecture as the teacher network, and its weights are denoted by $\boldsymbol{W} \in \mathbb{R}^{d \times K}$. Given $n$ pairs of training samples $\{\boldsymbol{x}_i, y_i\}_{i=1}^n$, the empirical risk function is

$$f_n(\boldsymbol{W}) = \frac{1}{n} \sum_{i=1}^n \ell(\boldsymbol{W}; \boldsymbol{x}_i, y_i) \tag{3}$$

where $\ell(\boldsymbol{W}; \boldsymbol{x}_i, y_i)$ is the cross-entropy loss function, i.e.,

$$\ell(\boldsymbol{W}; \boldsymbol{x}_i, y_i) = -y_i \cdot \log(H(\boldsymbol{W}, \boldsymbol{x}_i)) - (1 - y_i) \cdot \log(1 - H(\boldsymbol{W}, \boldsymbol{x}_i)). \tag{4}$$

To estimate $\boldsymbol{W}^*$ from training samples, we solve the following nonconvex minimization problem

$$\min_{\boldsymbol{W} \in \mathbb{R}^{d \times K}} f_n(\boldsymbol{W}). \tag{5}$$

Here we assume the input features $\boldsymbol{x}_i$ are generated i.i.d. from the Gaussian mixture model (Pearson, 1894; Titterington et al., 1985; Hsu & Kakade, 2013), which we denote as

$$\boldsymbol{x} \sim \sum_{l=1}^L \lambda_l \mathcal{N}(\boldsymbol{\mu}_l, \sigma_l^2 \boldsymbol{I}_d), \tag{6}$$

where $\mathcal{N}$ denotes the multi-variate Gaussian distribution with mean $\boldsymbol{\mu}_l \in \mathbb{R}^d$, and covariance $\sigma_l \boldsymbol{I}_d$ for $\sigma_l \in \mathbb{R}_+$ for all $l \in [L]$. The Gaussian mixture model can be viewed as

$$\boldsymbol{x} := \boldsymbol{\mu}_h + \boldsymbol{z}_h \in \mathbb{R}^d \tag{7}$$

where $h$ is a discrete random variable with $\Pr(h = l) = \lambda_l$ for $l \in [L]$, and $\boldsymbol{z}_l$ follows the multivariate Gaussian $\mathcal{N}(\boldsymbol{0}, \sigma_l^2 \boldsymbol{I}_d)$ with zero mean and covariance $\sigma_l^2 \boldsymbol{I}_d$[1].

If the Gaussian mixture model is symmetric, the symmetric distribution can be written as

$$\boldsymbol{x} \sim \begin{cases} \sum_{l=1}^{\frac{L}{2}} \lambda_l \big(\mathcal{N}(\boldsymbol{\mu}_l, \sigma_l^2 \boldsymbol{I}_d) + \mathcal{N}(-\boldsymbol{\mu}_l, \sigma_l^2 \boldsymbol{I}_d)\big) & L \text{ is even} \\ \lambda_1 \mathcal{N}(\boldsymbol{0}, \sigma_1^2 \boldsymbol{I}_d) + \sum_{l=2}^{\frac{L-1}{2}} \lambda_l \big(\mathcal{N}(\boldsymbol{\mu}_l, \sigma_l^2 \boldsymbol{I}_d) + \mathcal{N}(-\boldsymbol{\mu}_l, \sigma_l^2 \boldsymbol{I}_d)\big) & L \text{ is odd} \end{cases} \tag{8}$$

We assume without loss of generality that $\boldsymbol{\mu}_l$ belongs to the column space of $\boldsymbol{W}^*$ for all $l \in [L]$. To see this, note that an arbitrary $\boldsymbol{\mu}_l$ can be written as $\boldsymbol{\mu}_{l\parallel} + \boldsymbol{\mu}_{l\perp}$, where $\boldsymbol{\mu}_{l\parallel}$ belongs to the column space of $\boldsymbol{W}^*$, and $\boldsymbol{\mu}_{l\perp}$ is perpendicular to the column space. Then, from (2) and (7) we have

$$H(\boldsymbol{W}^*, \boldsymbol{x}) = \frac{1}{K} \sum_{j=1}^K \phi(\boldsymbol{w}_j^{*\top}(\boldsymbol{\mu}_{h\parallel} + \boldsymbol{\mu}_{h\perp} + \boldsymbol{z}_h)) = \frac{1}{K} \sum_{j=1}^K \phi(\boldsymbol{w}_j^{*\top}(\boldsymbol{\mu}_{h\parallel} + \boldsymbol{z}_h)) = H(\boldsymbol{W}^*, \boldsymbol{x}')$$

$$\tag{9}$$

where $\boldsymbol{x}' \sim \sum_{l=1}^L \lambda_l \mathcal{N}(\boldsymbol{\mu}_{l\parallel}, \sigma_l^2 \boldsymbol{I}_d)$. Thus, these two cases are equivalent.

## 3 PROPOSED LEARNING ALGORITHM

We propose Algorithm 1 to solve (5) and defer its theoeretical analysis to Section 4. The method starts from a initialization $\boldsymbol{W}_0 \in \mathbb{R}^{d \times K}$ computed based on the tensor initialization method (Subroutine 1) and then updates the iterates $\boldsymbol{W}_t$ using gradient descent with the step size $\eta_0$. To analyze the general cases, we assume an i.i.d. zero-mean noise $\{\nu_i\}_{i=1}^n \in \mathbb{R}^{d \times K}$ with bounded magnitude $|(\nu_i)_{jk}| \le \xi$ $(j \in [d], k \in [K])$ for some $\xi \ge 0$ when computing the gradient of the loss in (4).

Our tensor initialization method is extended from (Janzamin et al., 2014) and (Zhong et al., 2017b). The idea is to compute quantities ($\boldsymbol{M}_j$ in (10)) that are tensors of $\boldsymbol{w}_i^*$ and then apply the tensor decomposition method to estimate $\boldsymbol{w}_i^*$. Because $\boldsymbol{M}_j$ can only be estimated from training samples, tensor decomposition does not return $\boldsymbol{w}_i^*$ exactly but provides a close approximation. Because the existing method only applies to the standard Gaussian, we exploit the relationship between probability density functions and tensor expressions developed in (Janzamin et al., 2014) to design tensors suitable for the Gaussian mixture model. Formally,

---

[1]One can easily extend our analysis to the case when the covariance is $\text{diag}(\sigma_{l1}^2, \cdots, \sigma_{ld}^2)$. One needs to revise Property 4 and Lemma 7 correspondingly. We use the same $\sigma_l$ to simplify the presentation.

---

**Algorithm 1** Our proposed learning algorithm

---

**Input:** Training data $\{(\boldsymbol{x}_i, y_i)\}_{i=1}^n$, the step size $\eta_0 = O\left(\frac{1}{\sum_{l=1}^L \lambda_l(\|\boldsymbol{\mu}_l\|_\infty + \sigma_l)^2}\right)$, iteration $T$

**Initialization:** $\boldsymbol{W}_0 \leftarrow$ Tensor initialization method via Subroutine 1

**Gradient Descent:** for $t = 0, 1, \cdots, T-1$

$$\boldsymbol{W}_{t+1} = \boldsymbol{W}_t - \eta_0 \cdot \frac{1}{n} \sum_{i=1}^n (\nabla l(\boldsymbol{W}, \boldsymbol{x}_i, y_i) + \nu_i) = \boldsymbol{W}_t - \eta_0 \left(\nabla f_n(\boldsymbol{W}) + \frac{1}{n} \sum_{i=1}^n \nu_i\right)$$

**Output:** $\boldsymbol{W}_T$

---

**Definition 1** *Let $p(\boldsymbol{x}) = \sum_{l=1}^L \lambda_l (2\pi\sigma_l)^{-\frac{d}{2}} \exp(-\frac{\|\boldsymbol{x}-\boldsymbol{\mu}_l\|^2}{2\sigma_l^2})$ be the probability density function of the Gaussian mixture model in (6). We define*

$$\boldsymbol{M}_j := \mathbb{E}_{\boldsymbol{x} \sim \sum_{l=1}^L \lambda_l \mathcal{N}(\boldsymbol{\mu}_l, \sigma_l^2 \boldsymbol{I})} [y \cdot (-1)^j p^{-1}(\boldsymbol{x}) \nabla^{(m)} p(\boldsymbol{x})], \; j = 1, 2, 3 \qquad (10)$$

*Let $\boldsymbol{\alpha} \in \mathbb{R}^d$ denote an arbitrary vector. If the Gaussian Mixture Model is symmetric as in (8), then $\boldsymbol{P}_2 := \boldsymbol{M}_3(\boldsymbol{I}_d, \boldsymbol{I}_d, \boldsymbol{\alpha})$. Otherwise, $\boldsymbol{P}_2 := \boldsymbol{M}_2$.*

$\boldsymbol{M}_j$ is a $j$th-order tensor of $\boldsymbol{w}_i^*$, e.g., $\boldsymbol{M}_3 = \frac{1}{K} \sum_{i=1}^K \mathbb{E}_{\boldsymbol{x} \sim \sum_{l=1}^L \lambda_l \mathcal{N}(\boldsymbol{\mu}_l, \sigma_l^2 \boldsymbol{I})} [\phi'''(\boldsymbol{w}_i^{*\top} \boldsymbol{x})] \boldsymbol{w}_i^{*\otimes 3}$. These quantifies cannot be directly computed from (10) but can be estimated by sample means, denoted by $\widehat{\boldsymbol{M}}_i$ ($i = 1, 2, 3$) and $\widehat{\boldsymbol{P}}_2$, from samples $\{\boldsymbol{x}_i, y_i\}_{i=1}^n$. The following assumption guarantees that these tensors are nonzero and can thus be leveraged to estimate $\boldsymbol{W}^*$.

**Assumption 1** *The Gaussian Mixture Model in (6) satisfies the following conditions:*

1. $\mathbb{E}_{\boldsymbol{x} \sim \sum_{l=1}^L \lambda_l \mathcal{N}(\boldsymbol{\mu}_l, \sigma_l^2 \boldsymbol{I})} [\phi'''(\boldsymbol{w}_i^{*\top} \boldsymbol{x})] \neq 0$ *for $i \in [K]$,* which implies that $\boldsymbol{M}_3$ is nonzero.

2. *If the distribution is not symmetric, then* $\mathbb{E}_{\boldsymbol{x} \sim \sum_{l=1}^L \lambda_l \mathcal{N}(\boldsymbol{\mu}_l, \sigma_l^2 \boldsymbol{I})} [\phi''(\boldsymbol{w}_i^{*\top} \boldsymbol{x})] \neq 0$ *for $i \in [K]$,* which implies $\boldsymbol{M}_2$ and $\boldsymbol{P}_2$ in this case are nonzero.

Note that Assumption 1 is a very mild assumption[2]. Moreover, as indicted in (Janzamin et al., 2014), in the rare case that some quantities $\boldsymbol{M}_i$ ($i = 1, 2, 3$) and $\boldsymbol{P}_2$ are zero, one can construct higher-order tensors in a similar way as in Definition 1 and then estimate $\boldsymbol{W}^*$ from higher-order tensors.

Subroutine 1 estimates the direction and magnitude of $\boldsymbol{w}_j^*, j \in [K]$, separately. The key steps are as follows. We first use the power method to decompose $\widehat{\boldsymbol{P}}_2$ to approximate the subspace spanned by $\{\boldsymbol{w}_1^*, \boldsymbol{w}_2^*, \cdots, \boldsymbol{w}_K^*\}$, denoted by $\widehat{\boldsymbol{U}}$. Then, we project $\widehat{\boldsymbol{M}}_3 \in \mathbb{R}^{d \times d \times d}$ to $\widehat{\boldsymbol{R}}_3 \in \mathbb{R}^{K \times K \times K}$ using $\widehat{\boldsymbol{U}}$ to reduce the computational and sample complexity for decomposing a third-order tensor in the next step. We then apply the KCL algorithm to decompose $\widehat{\boldsymbol{R}}_3$ into vectors $\hat{\boldsymbol{v}}_i$. Note that $\widehat{\boldsymbol{U}}^\top \hat{\boldsymbol{v}}_i = s_i \bar{\boldsymbol{w}}_i^*$, where $s_i \in \{1, -1\}$ is a random sign. Then the direction of $\boldsymbol{w}_j^*$ is determined. Finally, the magnitude of $\boldsymbol{w}_i^*$'s and the signs of $s_i$'s are determined by solving a linear system of equations using the RecMagSign method. Please refer to (Zhong et al., 2017b) and (Kuleshov et al., 2015) for more details on the power method, KCL and RecMagSign methods.

## 4 MAIN THEORETICAL RESULTS

The main idea of our analysis is to show that the empirical risk function in (3) is strongly convex in a region near $\boldsymbol{W}^*$. Then $\boldsymbol{W}_0$ returned by Subroutine 3 is in this convex region, and the iterates returned by Algorithm 1 converge to a critical point in this region. Before formally stating our result in Theorem 1, we summarize the key implications of Theorem 1 as follow.

**1. Convergence rate and estimation accuracy:** When gradients are accurate (i.e., $\xi = 0$), the iterates $\boldsymbol{W}_t$ converge to a critical point $\widehat{\boldsymbol{W}}_n$ linearly, and the distance between $\widehat{\boldsymbol{W}}_n$ and $\boldsymbol{W}^*$ is

---

[2]By mild we mean given $L$, if Assumption 1 is not met for some $(\boldsymbol{\lambda}_0, \boldsymbol{M}_0, \boldsymbol{\sigma}_0)$, there exists an infinite number of $(\boldsymbol{\lambda}', \boldsymbol{M}', \boldsymbol{\sigma}')$ in any neighborhood of $(\boldsymbol{\lambda}_0, \boldsymbol{M}_0, \boldsymbol{\sigma}_0)$ such that Assumption 1 holds for $(\boldsymbol{\lambda}', \boldsymbol{M}', \boldsymbol{\sigma}')$,

---

**Subroutine 1** Tensor Initialization Method

    **Input:** Partition $n$ pairs of data $\{(\boldsymbol{x}_i, y_i)\}_{i=1}^n$ into three subsets $\mathcal{D}_1, \mathcal{D}_2, \mathcal{D}_3$
    Compute $\widehat{\boldsymbol{P}}_2$ using $\mathcal{D}_1$ and an arbitrary vector $\boldsymbol{\alpha}$
    $\widehat{\boldsymbol{U}} \longleftarrow \text{PowerMethod}(\widehat{\boldsymbol{P}}_2, K)$
    Compute $\widehat{\boldsymbol{R}}_3 = \widehat{\boldsymbol{M}}_3(\widehat{\boldsymbol{U}}, \widehat{\boldsymbol{U}}, \widehat{\boldsymbol{U}})$ from data set $\mathcal{D}_2$
    $\{\widehat{\boldsymbol{v}}_i\}_{i \in [K]} \longleftarrow \text{KCL}(\widehat{\boldsymbol{R}}_3)$
    $\{\boldsymbol{W}_0\} \longleftarrow \text{RecMagSign}(\widehat{\boldsymbol{U}}, \{\hat{\boldsymbol{v}}_i\}_{i \in [K]}, \mathcal{D}_3)$
    **Return:** $\boldsymbol{W}_0$

---

$O(\sqrt{d \log n / n})$. With the noise in the gradient, there is an additional error term of $O(\xi \sqrt{d \log n / n})$. For example, when $n$ is $\Theta(d \log^2 d)$, the estimation error decays as $O(\frac{1+\xi}{\log d})$.

**2. Sample complexity**: The sample complexity for accurate estimation is $\Theta(d \log^2 d)$ where $d$ is the feature dimension. This result is in the same order as the sample complexity for the standard Gaussian input in (Fu et al., 2020) and (Zhong et al., 2017b), indicating that our method can handle input from the Gaussian mixture model without increasing the order of the sample complexity. Our bound is almost order-wise optimal with respect to $d$ because the degree of freedom is $dK$. The additional multiplier of $\log^2 d$ results from the concentration bound in the proof technique.

**3. Impact of the mean**: If everything else is fixed, and at least one entry of a mean $\mu_{l(i)}$ (the $i$th entry of $\boldsymbol{\mu}_l$) of the Gaussian mixture model increases from 0 (in terms of the absolute value), the sample complexity increases to infinity and the convergence slows down. The intuition is that as the absolute value of some mean increases, some training samples have significantly large magnitude such that the sigmoid function saturates. These training samples are not informative for the estimation of $\boldsymbol{W}^*$, and the gradient of these samples is close to zero. Therefore, the required number of samples to estimate $\boldsymbol{W}^*$ needs to increase, and the gradient descent algorithm slows down.

**4. Impact of the variance**: If everything else is fixed, and at least one variance $\sigma_l$ of the Gaussian mixture model increases from a certain positive value, the sample complexity increases to infinity and the convergence slows down. The intuition is the same as increasing $|\mu_{l(i)}|$ in point 3. On the other hand, when all variances in the Gaussian mixture model approach zero, the sample complexity increases to infinity, and the convergence slows down. The intuition is that when the input data are concentrated on a few vectors, the optimization problem does not have a benign landscape.

Combining points 3 and 4, one can see that to learn the teacher network characterized by (2), the training samples shall have zero mean and a medium level of variance to reduce the sample complexity and speed up the convergence. If the variance is too large, some samples become non-informative and affect the learning negatively. If the variance is too small, the learning problem becomes mathematically challenging to solve. This theoretical characterization can be viewed as one motivation of the empirical techniques to improve learning rate such as whiting (LeCun et al., 1998) and Batch normalization (Ioffe & Szegedy, 2015). We state our main theoretical result as follows.

**Theorem 1** *Consider the binary classification problem with one hidden-layer fully connected neural network as in (2). Suppose Assumption 1 holds, then there exist $\epsilon_0 \in (0, \frac{1}{4})$ and positive value functions $\mathcal{B}(\boldsymbol{\lambda}, \boldsymbol{M}, \boldsymbol{\sigma}, \boldsymbol{W}^*)$ and $q(\boldsymbol{\lambda}, \boldsymbol{M}, \boldsymbol{\sigma}, \boldsymbol{W}^*)$ such that as long as the sample size $n$ satisfies*

$$n \geq n_{sc} := poly(\epsilon_0^{-1}, \kappa, K)\mathcal{B}(\boldsymbol{\lambda}, \boldsymbol{M}, \boldsymbol{\sigma}, \boldsymbol{W}^*)d \log^2 d, \tag{11}$$

*we have that with probability at least $1 - d^{-10}$, the iterates $\{\boldsymbol{W}_t\}_{t=1}^T$ returned by Algorithm 1 with step size $\eta_0 = O\left(\frac{1}{\sum_{l=1}^L \lambda_l (\|\boldsymbol{\mu}_l\|_\infty + \sigma_l)^2}\right)$ converge linearly with a statistical error to a critical point $\widehat{\boldsymbol{W}}_n$ with the rate of convergence $v = 1 - K^{-2}q(\boldsymbol{\lambda}, \boldsymbol{M}, \boldsymbol{\sigma}, \boldsymbol{W}^*)$, i.e.,*

$$\|\boldsymbol{W}_t - \widehat{\boldsymbol{W}}_n\|_F \leq v^t \|\boldsymbol{W}_0 - \widehat{\boldsymbol{W}}_n\|_F + \frac{\eta_0 \xi}{1-v}\sqrt{dK \log n / n}, \tag{12}$$

*Moreover, the distance between $\boldsymbol{W}^*$ and $\widehat{\boldsymbol{W}}_n$ is bounded by*

$$\|\widehat{\boldsymbol{W}}_n - \boldsymbol{W}^*\|_F \leq O\left(K^{\frac{5}{2}}(1+\xi) \cdot \sqrt{d \log n / n}\right). \tag{13}$$

We next quantify the impact of the parameters of the Gaussian mixture model on the sample complexity $n_{\text{sc}}$ and the convergence rate $v$ discussed in Theorem 1 as follows.

**Corollary 1** *(Impact of the Gaussian mixture model on $n_{sc}$ and $v$)*

*(1) When everything else is fixed, $n_{sc}$ increases to infinity, and $v$ increases to 1, as $|\mu_{l(i)}|$ with any $l \in [L]$ and $i \in [d]$ increases, where $\mu_{l(i)}$ is the $i$-th entry of $\boldsymbol{\mu}_l$.*

*(2) When everything else is fixed except for some $\sigma_l$ for any $l \in [L]$, $n_{sc}$ increases to infinity, and $v$ increases to 1, as $\sigma_l$ increases from $\zeta_s$ for some constant $\zeta_s > 0$.*

*(3) $n_{sc}$ increases to infinity, and $v$ increases to 1 if all $\sigma_l$'s go to zero for all $l \in [L]$.*

To the best of our knowledge, Theorem 1 provides the first explicit characterization of the sample complexity and learning rate when the input follows the Gaussian mixture model. Although we consider the sigmoid activation in this paper, our results apply to any activation function $\phi$ provided that $\phi'$ is an even function, and $\phi$, $\phi'$ and $\phi''$ are bounded. Examples include $\tanh$ and $\text{erf}$. Algorithm 1 employs a constant step size. One can potentially speed up the convergence, i.e., reduce $v$, by using a variable step size. We leave the corresponding theoretical analysis for future work.

If we scale the weights $\boldsymbol{W}^{*'} = \boldsymbol{W}^*/c$ and the input feature $\boldsymbol{x}' = c\boldsymbol{x}$ simultaneously, the output remains the same for any nonzero constant $c$. Therefore, the learning problems in these two cases are equivalent in terms of the sample complexity and convergence rate. Theorem 1 reflects such equivalence. One can check that $\mathcal{B}(\boldsymbol{\lambda}, \boldsymbol{M}, \boldsymbol{\sigma}, \boldsymbol{W}^*) = \mathcal{B}(\boldsymbol{\lambda}, \boldsymbol{M}', \boldsymbol{\sigma}', \boldsymbol{W}^{*'})$ from the proof in Section B. Similarly, the convergence rate in (12) remains the same in both cases.

One main component in the proof of Theorem 1 to show that if (11) holds, the landscape of the empirical risk is close to that of the population risk in a local neighborhood of $\boldsymbol{W}^*$. (Mei et al., 2018a) quantified the similarity of these two functions when $K = 1$, but it is not clear if their approach can be extended to the case $K > 1$. Here, focusing on the Gaussian mixture model, we explicitly quantify the impact of the parameters of the input distribution on the landscapes of these functions. Please see Appendix-C for details.

Compared with the analyses for the standard Gaussian in (Fu et al., 2020; Zhong et al., 2017b), we develop new techniques in the following aspects. First, a direct extension of the matrix concentration inequalities in these works leads to a sample complexity bound of $O(d^3)$, while we develop new concentration bounds to tighten it to $O(d \log^2 d)$. Second, the existing analysis to bound the Hessian of the population risk function does not extend to the Gaussian mixture model. We develop new tools that also apply to other activation functions like $\tanh$ or $\text{erf}$. Third, we design new tensors for the initialization, and the proof about the tensor initialization is revised accordingly.

The above results assume the parameters of the Gaussian mixture are known. In practice, they can be estimated by the EM algorithm (Redner & Walker, 1984) and the moment-based method (Hsu & Kakade, 2013). The EM algorithm returns model parameters within Euclidean distance $O((\frac{d}{n})^{\frac{1}{2}})$ when the number of mixture components $L$ is known. When $L$ is unknown, one usually over-specifies an estimate $\bar{L} > L$, then the estimation error by the EM algorithm scales as $O((\frac{d}{n})^{\frac{1}{4}})$. Please refer to (Ho & Nguyen, 2016; Ho et al., 2020; Dwivedi et al., 2020a;b) for details.

## 5 NUMERICAL EXPERIMENTS

We verify Theorem 1 through numerical experiments. We generate a ground-truth $\boldsymbol{W}^* \in \mathbb{R}^{d \times K}$ from the Gaussian distribution. The training samples $\{\boldsymbol{x}_i, y_i\}_{i=1}^n$ are generated using (6) and (2). The maximum number of iterations of Algorithm 1 is set as 12000.

### 5.1 TENSOR INITIALIZATION

Fig. 1 shows the accuracy of the returned model by Algorithm 1. Here $d = 5$, $K = 2$, $\lambda_1 = \lambda_2 = 0.5$, $\boldsymbol{\mu}_1 = -\mathbf{1}$ and $\boldsymbol{\mu}_2 = \mathbf{0}$. We compare the tensor initialization with a random initialization in a local region $\{\boldsymbol{W} \in \mathbb{R}^{d \times K} : \frac{||\boldsymbol{W} - \boldsymbol{W}^*||_F}{||\boldsymbol{W}^*||_F} \leq \epsilon\}$. Tensor initialization in Subroutine 1 returns an initial point close to $\boldsymbol{W}^*$ with a relative error of 0.61. If the random initialization is also close to $\boldsymbol{W}^*$, e.g.,

$\epsilon = 0.1$, then the gradient descent algorithm converges to a critical point from both initializations, and the linear convergence rate is the same. If the random initialization is far away, e.g., $\epsilon = 1.5$, the algorithm does not converge. On a MacBook Pro with Intel(R) Core(TM) i5-7360U CPU at 2.30GHz and MATLAB 2017a, it takes 0.55 second to compute the tensor initialization. We consider a random initialization with $\epsilon = 0.1$ in the following experiments to simplify the computation.

## 5.2 SAMPLE COMPLEXITY

Consider the case that $K = 3$, $L = 2$, $\lambda_1 = \lambda_2 = \frac{1}{2}$. Let $\boldsymbol{\mu}_1$ be an all one vector in $\mathbb{R}^d$ and let $\boldsymbol{\mu}_2 = -\boldsymbol{\mu}_1$. Let $\sigma_1 = \sigma_2 = 1$. We vary $d$ and evaluate the sample complexity bound in (11) with respect to $d$. We randomly initialize $M$ times and let $\widehat{\boldsymbol{W}}_n^{(m)}$ denote the output of Algorithm 1 in the $m$th trail. Let $\bar{\boldsymbol{W}}_n$ denote the mean values of all $\widehat{\boldsymbol{W}}_n^{(m)}$, and let $d_W = \sqrt{\sum_{m=1}^{M} ||\widehat{\boldsymbol{w}}_n^m - \bar{\boldsymbol{W}}_n||^2/M}$ denote the variance. An experiment is successful if $d_W \leq 10^{-4}$ and fails otherwise. $M$ is set as 20.

We vary $d$ and the number of samples $n$. For each pair of $d$ and $n$, 20 independent sets of $\boldsymbol{W}^*$ and the corresponding training samples are generated. Fig. 2 shows the success rate of these independent experiments. A black block means that all the experiments fail. A white block means that they all succeed. The sample complexity is indeed almost linear in $d$, as predicted by (11). Moreover, the coefficient $n/d$ can be large depending on the problem setup.

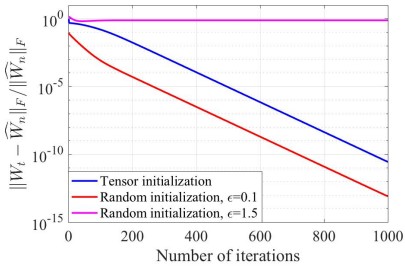

Figure 1: Comparison between gradient descent with tensor initialization and random initialization

Figure 2: The sample complexity against the feature dimension $d$

We then fix $d = 5$ and study the impact on the sample complexity when the mean and variance in the Gaussian mixture model change. In Fig. 3.(a), we fix $\sigma_1 = \sigma_2 = 1$ and let $\boldsymbol{\mu}_1 = \mu \cdot \mathbf{1}$, $\boldsymbol{\mu}_2 = -\mathbf{1}$. $\mu$ varies from 0 to 7.5. Fig. 3.(a) shows that when the mean increases, the sample complexity increases. This coincides with our theoretical analyses in Section 4. In Fig. 3.(b), we fix $\boldsymbol{\mu}_1 = \mathbf{1}$, $\boldsymbol{\mu}_2 = -\mathbf{1}$, and let $\sigma_1 = \sigma$ and $\sigma_2 = 1$. $\sigma$ varies from $10^{-1.4}$ to $10^{1.4}$. The sample complexity increases both when $\sigma$ increases and when $\sigma$ approaches zero. The results match our theoretical prediction in Section 4.

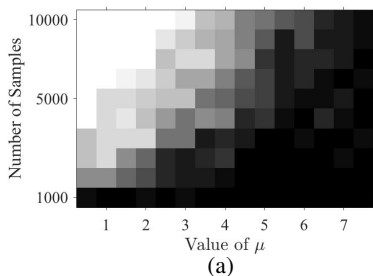

(a)

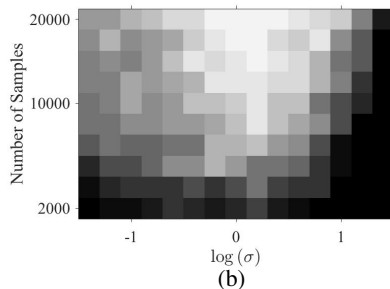

(b)

Figure 3: The sample complexity (a) when one mean changes, (b) when one variance changes.

### 5.3 CONVERGENCE ANALYSIS

We next study the convergence rate of Algorithm 1. $d$ is fixed as 5. Fig. 4.(a) shows the impact of the mean of the Gaussian mixture model on the convergence rate. We set $\lambda_1 = \lambda_2 = 0.5$, $\boldsymbol{\mu}_1 = \mu \cdot \mathbf{1}$, $\boldsymbol{\mu}_2 = -\mathbf{1}$, and $\sigma_1 = \sigma_2 = 1$. The sample complexity $n$ is set to 10000. One can see that Algorithm 1 always converges linearly when $\mu$ changes. Moreover, as $\mu$ increases, Algorithm 1 converges slower, as predicted by our theoretical analyses in Section 4. In Fig. 4.(b) shows the impact of the variance of the Gaussian mixture model. We set $\lambda_1 = \lambda_2 = 0.5$, $\boldsymbol{\mu}_1 = \mathbf{1}$, $\boldsymbol{\mu}_2 = -\mathbf{1}$, $\sigma_1 = \sigma_2 = \sigma$. The sample complexity $n$ is set to 50000. Among different $\sigma$ we test, Algorithm 1 converges fastest when $\sigma = 1$. The convergence rate slows down when $\sigma$ increases to 2 or when $\sigma$ decreases to 0.5. The result is consistent with our theoretical results in Section 4.

We then verify the convergence rate in (12), which shows that $v = 1 - \Theta(K^{-2})$. We set $\lambda_1 = \lambda_2 = 0.5$, $\boldsymbol{\mu}_1 = \mathbf{1}$, $\boldsymbol{\mu}_2 = -\mathbf{1}$, $\sigma_1 = \sigma_2 = 1$. $K$ ranges from 2 to 8. One can see from Fig. 5 that, as predicted, the convergence rate is almost linear in $1/K^2$.

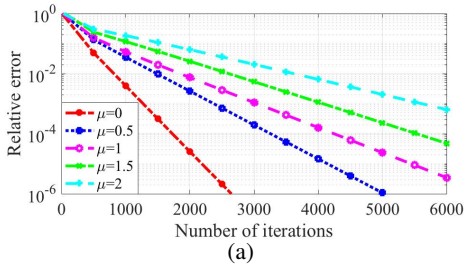 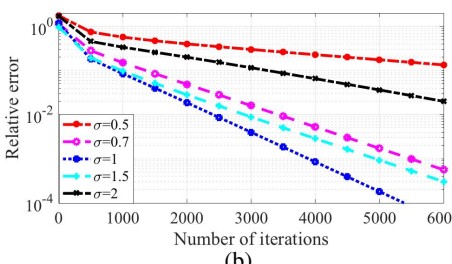

(a)  (b)

Figure 4: (a) The convergence rate with different $\mu$, (b) The convergence rate with different $\sigma$.

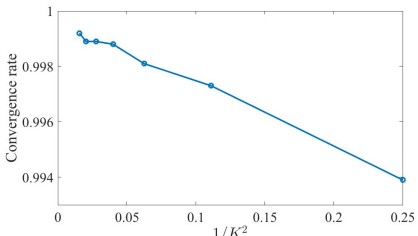 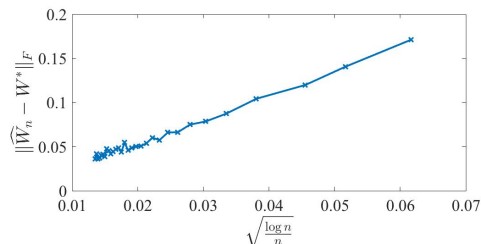

Figure 5: Convergence rate when the number of neurons $K$ changes

Figure 6: The relative error of the learned model with the ground-truth when $n$ changes

We then evaluate the distance between $\widehat{\boldsymbol{W}}_n$ returned by Algorithm 1 and $\boldsymbol{W}^*$, measured by $||\widehat{\boldsymbol{W}}_n - \boldsymbol{W}^*||_F$. $d$ is 5. $n$ ranges from $2 \times 10^3$ to $6 \times 10^4$. $\sigma_1 = \sigma_2 = 3$, $\boldsymbol{\mu}_1 = \mathbf{1}$, $\boldsymbol{\mu}_2 = -\mathbf{1}$. Each point in Fig. 6 is averaged over 100 independent experiments of different $\boldsymbol{W}^*$ and the corresponding training set. $||\boldsymbol{W}^*||_F$ is normalized to 1. The error is indeed linear in $\sqrt{\log(n)/n}$, as predicted by (12).

## 6 CONCLUSIONS

This paper analyzes the theoretical performance guarantee of learning one-hidden-layer neural networks for binary classification when the input follows the Gaussian mixture model. We develop an algorithm that converges linearly to a model that has a diminishing difference from the ground-truth model that has guaranteed generalizability. We also provide the first explicit characterization of the impact of the input distribution on the sample complexity and convergence rate. Future works include the analysis of multiple-hidden-layer neural networks and multi-class classification. Because of the concatenation of nonlinear activation functions, the analysis of the landscape of the empirical risk and the design of a proper initialization is more challenging and requires the development of new tools.

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

## A    PRELIMINARIES

In this section, we introduce some definitions and properties that will be used in proving the main results.

First we define the sub-Gaussian random variable and sub-Gaussian norm.

**Definition 2** *We say $X$ is a sub-Gaussian random variable with sub-Gaussian norm $K > 0$, if $(\mathbb{E}|X|^p)^{\frac{1}{p}} \leq K\sqrt{p}$ for all $p \geq 1$. In addition, the sub-Gaussian norm of X, denoted $\|X\|_{\psi_2}$, is defined as $\|X\|_{\psi_2} = \sup_{p \geq 1} p^{-\frac{1}{2}} (\mathbb{E}|X|^p)^{\frac{1}{p}}$.*

Then we define the following three quantities. $\rho(\boldsymbol{\mu}, \sigma)$ is motivated by the $\rho$ parameter for the standard Gaussian distribution in (Zhong et al., 2017b), and we generalize it to a Gaussian with an arbitrary mean and variance. We define the new quantities $\Gamma(\boldsymbol{\lambda}, \boldsymbol{M}, \boldsymbol{\sigma}, \boldsymbol{W}^*)$ and $D_m(\boldsymbol{\lambda}, \boldsymbol{M}, \boldsymbol{\sigma})$ for the Gaussian mixture model.

**Definition 3** *($\rho$-function). Let $\boldsymbol{z} \sim \mathcal{N}(\boldsymbol{u}, \boldsymbol{I}_d) \in \mathbb{R}^d$. Define $\alpha_q(i, \boldsymbol{u}, \sigma) = \mathbb{E}_{z_i \sim \mathcal{N}(u_i, 1)}[\phi'(\sigma \cdot z_i)z_i^q]$ and $\beta_q(i, \boldsymbol{u}, \sigma) = \mathbb{E}_{z_i \sim \mathcal{N}(u_i, 1)}[\phi'^2(\sigma \cdot z_i)z_i^q], \forall q \in \{0, 1, 2\}$, where $z_i$ and $u_i$ is the i-th entry of $\boldsymbol{z}$ and $\boldsymbol{u}$, respectively. Define $\rho(\boldsymbol{u}, \sigma)$ as*

$$\rho(\boldsymbol{u}, \sigma) = \min_{i,j \in [d], j \neq i} \{(u_j^2 + 1)(\beta_0(i, \boldsymbol{u}, \sigma) - \alpha_0(i, \boldsymbol{u}, \sigma)^2), \beta_2(i, \boldsymbol{u}, \sigma) - \frac{\alpha_2(i, \boldsymbol{u}, \sigma)^2}{u_i^2 + 1}\} \quad (14)$$

**Definition 4** *($\Gamma$-function). With (6), (14) and $\kappa, \eta$ defined in Section 2, we define*

$$\Gamma(\boldsymbol{\lambda}, \boldsymbol{M}, \boldsymbol{\sigma}, \boldsymbol{W}^*) = \sum_{l=1}^{L} \frac{\lambda_l}{\kappa^2 \eta} \frac{\sigma_l^2}{\sigma_{\max}^2} \rho(\frac{\boldsymbol{W}^{*\top} \boldsymbol{\mu}_l}{\sigma_l \delta_K(\boldsymbol{W}^*)}, \sigma_l \delta_K(\boldsymbol{W}^*)) \quad (15)$$

**Definition 5** *(D-function). Given the Gaussian Mixture Model in (6) and any positive integer $m$, define $D_m(\boldsymbol{\lambda}, \boldsymbol{M}, \boldsymbol{\sigma})$ as*

$$D_m(\boldsymbol{\lambda}, \boldsymbol{M}, \boldsymbol{\sigma}) = \sum_{l=1}^{L} \lambda_l (\frac{\|\boldsymbol{\mu}_l\|_\infty}{\sigma_l} + 1)^m, \quad (16)$$

*where $\boldsymbol{\lambda} = (\lambda_1, \cdots, \lambda_L) \in \mathbb{R}^L$, $\boldsymbol{M} = (\boldsymbol{\mu}_1, \cdots, \boldsymbol{\mu}_L) \in \mathbb{R}^{d \times L}$ and $\boldsymbol{\sigma} = (\sigma_1, \cdots, \sigma_L) \in \mathbb{R}^L$.*

$\rho$-function is defined to compute the lower bound of the Hessian of the population risk with Gaussian input. $\Gamma$ function is the weighted sum of $\rho$-function under mixture Gaussian distribution. This function is positive and upper bounded by a small value. It is increasing when $|\mu_{l(i)}|$ increases. When $\sigma_l$ increases, $\Gamma$ increases first and then decreases. $\Gamma$ goes to zero if all $\|\boldsymbol{\mu}_l\|_\infty$ or all $\sigma_l$ goes to infinity. $D$-function is a normalized parameter for the means and variances. It is lower bounded by 1. $D$-function is an increasing function of $\|\boldsymbol{\mu}_l\|_\infty$ and a decreasing function of $\sigma_l$.

**Property 1** *We have that $\|\nu_i\|_F$ is a sub-Gaussian random variable with its sub-Gaussian norm bounded bu $\xi\sqrt{dK}$.*

**Proof:**

$$(\mathbb{E}\|\nu_i\|_F^p)^{\frac{1}{p}} \leq (\mathbb{E}|\sqrt{dK}\xi|^p)^{\frac{1}{p}} \leq \xi\sqrt{dK} \quad (17)$$

**Property 2** *$\rho(\boldsymbol{u}, \sigma)$ in Definition 3 satisfies the following properties,*

1. *$\rho(\boldsymbol{u}, \sigma) > 0$ for any $\boldsymbol{u} \in \mathbb{R}^d$ and $\sigma \neq 0$.*

2. *$\rho(\boldsymbol{u}, \sigma)$ converges to a positive value function of $\sigma$ as $u_i$ goes to 0, i.e. $\lim_{u_i \to 0} \rho(\boldsymbol{u}, \sigma) := \mathcal{C}_m(\sigma)$.*

3. *When all $u_i \neq 0$ $(i \in [d])$, $\rho(\frac{\boldsymbol{u}}{\sigma}, \sigma)$ converges to a positive value function of $\boldsymbol{u}$ as $\sigma$ goes to 0, i.e. $\lim_{\sigma \to 0} \rho(\frac{\boldsymbol{u}}{\sigma}, \sigma) := \mathcal{C}_s(\boldsymbol{u})$. When $u_i = 0$ for some $i \in [d]$, $\lim_{\sigma \to 0} \rho(\frac{\boldsymbol{u}}{\sigma}, \sigma) = 0$.*

4. *When everything else except $|u_i|$ is fixed, $\rho(\frac{\boldsymbol{W}^{*\top}\boldsymbol{u}}{\sigma \delta_K(\boldsymbol{W}^*)}, \sigma \delta_K(\boldsymbol{W}^*))$ is lower bounded by a positive value function, $\mathcal{L}_m(\frac{\boldsymbol{W}^{*\top}\boldsymbol{u}}{\sigma \delta_K(\boldsymbol{W}^*)}, \sigma \delta_K(\boldsymbol{W}^*))$, which is monotonically decreasing to 0 as $|u_i|$ increases.*

5. *When everything else except $\sigma$ is fixed, $\rho(\frac{\boldsymbol{W}^{*\top}\boldsymbol{u}}{\sigma \delta_K(\boldsymbol{W}^*)}, \sigma \delta_K(\boldsymbol{W}^*))$ is lower bounded by a positive value function, $\mathcal{L}_s(\frac{\boldsymbol{W}^{*\top}\boldsymbol{u}}{\sigma \delta_K(\boldsymbol{W}^*)}, \sigma \delta_K(\boldsymbol{W}^*))$, which satisfies the following conditions: (a) there exists $\zeta_{s'} > 0$, such that $\sigma^{-1} \mathcal{L}_s(\frac{\boldsymbol{W}^{*\top}\boldsymbol{u}}{\sigma \delta_K(\boldsymbol{W}^*)}, \sigma \delta_K(\boldsymbol{W}^*))$ is an increasing function of $\sigma$ when $\sigma \in (0, \zeta_{s'})$; (b) there exists $\zeta_s > 0$ such that $\mathcal{L}_s(\frac{\boldsymbol{W}^{*\top}\boldsymbol{u}}{\sigma \delta_K(\boldsymbol{W}^*)}, \sigma \delta_K(\boldsymbol{W}^*))$ is an decreasing function of $\sigma$ when $\sigma \in (\zeta_s, +\infty)$.*

**Proof:**

(1) From the Cauchy Schwarz's inequality, we have

$$\mathbb{E}_{z_i \sim \mathcal{N}(u_i, 1)}[\phi'(\sigma \cdot z_i)] \leq \sqrt{\mathbb{E}_{z_i \sim \mathcal{N}(u_i, 1)}[\phi'^2(\sigma \cdot z_i)]} \tag{18}$$

$$\begin{aligned}
\mathbb{E}_{z_i \sim \mathcal{N}(u_i, 1)}[\phi'(\sigma \cdot z_i) z_i \cdot z_i] &\leq \sqrt{\mathbb{E}_{z_i \sim \mathcal{N}(u_i, 1)}[\phi'^2(\sigma \cdot z_i) z_i^2]} \cdot \sqrt{\mathbb{E}_{z_i \sim \mathcal{N}(u_i, 1)}[z_i^2]} \\
&= \sqrt{\mathbb{E}_{z_i \sim \mathcal{N}(u_i, 1)}[\phi'^2(\sigma \cdot z_i) z_i^2]} \cdot \sqrt{u_i^2 + 1}
\end{aligned} \tag{19}$$

The equalities of the (18) and (19) hold if and only if $\phi'$ is a constant function. Since that $\phi$ is the sigmoid function, the equalities of (18) and (19) cannot hold.

By the definition of $\rho(\boldsymbol{u}, \sigma)$ in Definition 3, we have $\beta_0(i, \boldsymbol{u}, \sigma) - \alpha_0^2(i, \boldsymbol{u}, \sigma) > 0$ and $\beta_2(i, \boldsymbol{u}, \sigma) - \frac{\alpha_2^2(i, \boldsymbol{u}, \sigma)}{u_i^2 + 1} > 0$. Therefore,

$$\rho(\boldsymbol{u}, \sigma) > 0 \tag{20}$$

(2)

$$\begin{aligned}
&\lim_{u_i \to 0} (\frac{u_j^2}{\sigma^2} + 1)(\beta_0(i, \boldsymbol{u}, \sigma) - \alpha_0^2(i, \boldsymbol{u}, \sigma)) \\
=&\lim_{u_i \to 0} (\frac{u_j^2}{\sigma^2} + 1)\Big( \int_{-\infty}^{\infty} \phi'^2(\sigma \cdot z_i)(2\pi)^{-\frac{1}{2}} \exp(-\frac{\|z_i - u_i\|^2}{2}) dz_i \\
&- (\int_{-\infty}^{\infty} \phi'(\sigma \cdot z_i)(2\pi)^{-\frac{1}{2}} \exp(-\frac{\|z_i - u_i\|^2}{2}) dz_i)^2 \Big) \\
=&(\frac{u_j^2}{\sigma^2} + 1)\Big( \int_{-\infty}^{\infty} \phi'^2(\sigma \cdot z_i)(2\pi)^{-\frac{1}{2}} \exp(-\frac{\|z_i\|^2}{2}) dz_i - (\int_{-\infty}^{\infty} \phi'(\sigma \cdot z_i)(2\pi)^{-\frac{1}{2}} \exp(-\frac{\|z_i\|^2}{2}) dz_i)^2 \Big)
\end{aligned} \tag{21}$$

$$\begin{aligned}
&\lim_{u_i \to 0} (\beta_2(i, \boldsymbol{u}, \sigma) - \frac{1}{u_i^2 + 1}\alpha_2^2(i, \boldsymbol{u}, \sigma)) \\
=&\lim_{u_i \to 0} \int_{-\infty}^{\infty} \phi'^2(\sigma \cdot z_i) z_i^2 (2\pi)^{-\frac{1}{2}} \exp(-\frac{\|z_i - u_i\|^2}{2}) dz_i \\
&- (\frac{1}{u_i^2 + 1} \int_{-\infty}^{\infty} \phi'(\sigma \cdot z_i) z_i^2 (2\pi)^{-\frac{1}{2}} \exp(-\frac{\|z_i - u_i\|^2}{2}) dz_i)^2 \\
=&\int_{-\infty}^{\infty} \phi'^2(\sigma \cdot z_i) z_i^2 (2\pi)^{-\frac{1}{2}} \exp(-\frac{\|z_i\|^2}{2}) dz_i - (\int_{-\infty}^{\infty} \phi'(\sigma \cdot z_i) z_i^2 (2\pi)^{-\frac{1}{2}} \exp(-\frac{\|z_i\|^2}{2}) dz_i)^2
\end{aligned} \tag{22}$$

Combining (21) and (22), we can derive that $\rho(\boldsymbol{u}, \sigma)$ converges to a positive value function of $\sigma$ as $u_i$ goes to 0, i.e. $\lim_{u_i \to 0} \rho(\boldsymbol{u}, \sigma) := \mathcal{C}_m(\sigma)$

(3) When all $u_i \neq 0$ $(i \in [d])$,

$$\lim_{\sigma \to 0} \left(\beta_2(i, \frac{\boldsymbol{u}}{\sigma}, \sigma) - \frac{1}{\frac{u_i^2}{\sigma^2} + 1}\alpha_2^2(i, \frac{\boldsymbol{u}}{\sigma}, \sigma)\right)$$

$$= \lim_{\sigma \to 0} \int_{-\infty}^{\infty} \phi'^2(\sigma \cdot z_i) z_i^2 (2\pi)^{-\frac{1}{2}} \exp(-\frac{\|z_i - \frac{u_i}{\sigma}\|^2}{2})dz_i$$

$$- \frac{1}{\frac{u_i^2}{\sigma^2} + 1}\left(\int_{-\infty}^{\infty} \phi'(\sigma \cdot z_i) z_i^2 (2\pi)^{-\frac{1}{2}} \exp(-\frac{\|z_i - \frac{u_i}{\sigma}\|^2}{2})dz_i\right)^2$$

$$= \lim_{\sigma \to 0} \int_{-\infty}^{\infty} \phi'^2(u_i \cdot x_i)\frac{u_i^2}{\sigma^2} x_i^2 (2\pi\frac{\sigma^2}{u_i^2})^{-\frac{1}{2}} \exp(-\frac{\|x_i - 1\|^2}{2\frac{\sigma^2}{u_i^2}})dx_i$$

$$- \frac{1}{\frac{u_i^2}{\sigma^2} + 1}\left(\int_{-\infty}^{\infty} \phi'(u_i \cdot x_i)\frac{u_i^2}{\sigma^2} x_i^2 (2\pi\frac{\sigma^2}{u_i^2})^{-\frac{1}{2}} \exp(-\frac{\|x_i - 1\|^2}{2\frac{\sigma^2}{u_i^2}})dx_i\right)^2 \qquad z_i = \frac{u_i}{\sigma}x_i \quad (23)$$

$$= \lim_{\sigma \to 0} \phi'^2(u_i)\frac{u_i^2}{\sigma^2} - \frac{1}{\frac{u_i^2}{\sigma^2} + 1}\left(\phi'(u_i)\frac{u_i^2}{\sigma^2}\right)$$

$$= \lim_{\sigma \to 0} \phi'^2(u_i)\frac{u_i^2}{\sigma^2}\left(1 - \frac{\frac{u_i^2}{\sigma^2}}{1 + \frac{\sigma^2}{u_i^2}}\right)^2$$

$$= \lim_{\sigma \to 0} \phi'^2(u_i)\frac{1}{1 + \frac{\sigma^2}{u_i^2}}$$

$$= \phi'^2(u_i)$$

The third step of (23) is by the fact that the Gaussian distribution goes to a Dirac delta function when $\sigma$ goes to $0$. Then the integral will take the value when $x_i = 1$. Similarly, we can obtain the following

$$\lim_{\sigma \to 0} \left(\beta_0(i, \frac{\boldsymbol{u}}{\sigma}, \sigma) - \alpha_0^2(i, \frac{\boldsymbol{u}}{\sigma}, \sigma)\right)$$

$$= \lim_{\sigma \to 0} \int_{-\infty}^{\infty} \phi'^2(\sigma \cdot z_i)(2\pi)^{-\frac{1}{2}} \exp(-\frac{\|z_i - \frac{u_i}{\sigma}\|^2}{2})dz_i$$

$$- \left(\int_{-\infty}^{\infty} \phi'(\sigma \cdot z_i)(2\pi)^{-\frac{1}{2}} \exp(-\frac{\|z_i - \frac{u_i}{\sigma}\|^2}{2})dz_i\right)^2 \qquad (24)$$

$$= \phi'^2(u_i) - \phi'^2(u_i) = 0$$

$$\lim_{\sigma \to 0} \left(\frac{\partial}{\partial \sigma}\left(\beta_0(i, \frac{\boldsymbol{u}}{\sigma}, \sigma) - \alpha_0^2(i, \frac{\boldsymbol{u}}{\sigma}, \sigma)\right)\right)$$

$$= \lim_{\sigma \to 0} \left(\frac{\partial}{\partial \sigma}\left(\int_{-\infty}^{\infty} \phi'^2(x_i)(2\pi\sigma^2)^{-\frac{1}{2}} \exp(-\frac{\|x_i - u_i\|^2}{2\sigma^2})dx_i\right.\right.$$

$$\left.\left. - \left(\int_{-\infty}^{\infty} \phi'(x_i)(2\pi\sigma^2)^{-\frac{1}{2}} \exp(-\frac{\|x_i - u_i\|^2}{2\sigma^2})dx_i\right)^2\right)\right) \qquad x_i = \sigma \cdot z_i$$

$$= \lim_{\sigma \to 0} \left(\int_{-\infty}^{\infty} \phi'^2(x_i)(2\pi\sigma^2)^{-\frac{1}{2}} \exp(-\frac{\|x_i - u_i\|^2}{2\sigma^2})(-\sigma^{-1} + \|x_i - u_i\|^2\sigma^{-2})dx_i\right.$$

$$- 2\left(\int_{-\infty}^{\infty} \phi'(x_i)(2\pi\sigma^2)^{-\frac{1}{2}} \exp(-\frac{\|x_i - u_i\|^2}{2\sigma^2})dx_i\right) \qquad (25)$$

$$\left. \cdot \int_{-\infty}^{\infty} \phi'(x_i)(2\pi\sigma^2)^{-\frac{1}{2}} \exp(-\frac{\|x_i - u_i\|^2}{2\sigma^2})(-\sigma^{-1} + \|x_i - u_i\|^2\sigma^{-2})dx_i\right)$$

$$= \lim_{\sigma \to 0} \left(\frac{\phi'^2(u_i)}{-\sigma} - 2\phi'(u_i)\frac{\phi'(u_i)}{-\sigma}\right)$$

$$= \lim_{\sigma \to 0} \frac{\phi'^2(u_i)}{\sigma} = +\infty$$

Therefore, by L'Hopital's rule and (24), (25), we have

$$
\begin{aligned}
&\lim_{\sigma \to 0} (\frac{u_j^2}{\sigma^2} + 1)(\beta_0(i, \frac{\boldsymbol{u}}{\sigma}, \sigma) - \alpha_0(i, \frac{\boldsymbol{u}}{\sigma}, \sigma)) \\
&= \lim_{\sigma \to 0} \frac{u_i^2}{2\sigma} \frac{\partial}{\partial \sigma}(\beta_0(i, \frac{\boldsymbol{u}}{\sigma}, \sigma) - \alpha_0(i, \frac{\boldsymbol{u}}{\sigma}, \sigma)) \\
&= +\infty
\end{aligned}
\tag{26}
$$

Combining (26) and (23), we can derive that $\rho(\frac{\boldsymbol{u}}{\sigma}, \sigma)$ converges to a positive value function of $\boldsymbol{u}$ as $\sigma$ goes to 0, i.e. $\lim_{\sigma \to 0} \rho(\frac{\boldsymbol{u}}{\sigma}, \sigma) := \mathcal{C}_s(\boldsymbol{u})$.

When $u_i = 0$ for some $i \in [d]$, $\lim_{\sigma \to 0}(\frac{u_i^2}{\sigma^2} + 1)(\beta_0(j, \frac{\boldsymbol{u}}{\sigma}, \sigma) - \alpha^2(j, \frac{\boldsymbol{u}}{\sigma}, \sigma)) = 0$ by (24). Then from the Definition 3, we have $\lim_{\sigma \to 0} \rho(\frac{\boldsymbol{u}}{\sigma}, \sigma) = 0$.

(4) We show the statement by contradiction. Suppose that for any positive value function, $h(u_i)$, which is monotonically decreasing to 0 as $|u_i|$ increases, there exists a $u_i^* \in \mathbb{R}$ such that $h(u_i) \geq \rho(\frac{\boldsymbol{W}^{*\top}\boldsymbol{u}}{\sigma\delta_K(\boldsymbol{W}^*)}, \sigma\delta_K(\boldsymbol{W}^*))\big|_{u_i = u_i*}$. Then we can derive that $\lim_{u_i \to u_i^*} \rho(\frac{\boldsymbol{W}^{*\top}\boldsymbol{u}}{\sigma\delta_K(\boldsymbol{W}^*)}, \sigma\delta_K(\boldsymbol{W}^*))\big|_{u_i = u_i*} = 0$. Since that $\rho(\frac{\boldsymbol{W}^{*\top}\boldsymbol{u}}{\sigma\delta_K(\boldsymbol{W}^*)}, \sigma\delta_K(\boldsymbol{W}^*))$ is continuous, we can obtain that $\rho(\frac{\boldsymbol{W}^{*\top}\boldsymbol{u}}{\sigma\delta_K(\boldsymbol{W}^*)}, \sigma\delta_K(\boldsymbol{W}^*))\big|_{u_i = u_i*} = 0$, which contradicts to the conclusion in Property 2.1.

(5) The condition (b) can be easily proved as (4). Therefore, we only need to show the condition (a). When $(\boldsymbol{W}^{*\top}\boldsymbol{u})_i \neq 0$ for all $i \in [K]$, $\lim_{\sigma \to 0} \rho(\frac{\boldsymbol{W}^{*\top}\boldsymbol{u}}{\sigma\delta_K(\boldsymbol{W}^*)}, \sigma\delta_K(\boldsymbol{W}^*)) = \mathcal{C}_s(\boldsymbol{u}) > 0$. Therefore, there exists $\zeta_s > 0$, such that when $0 < \sigma < \zeta_s$, $\rho(\frac{\boldsymbol{W}^{*\top}\boldsymbol{u}}{\sigma\delta_K(\boldsymbol{W}^*)}, \sigma\delta_K(\boldsymbol{W}^*)) > \frac{\mathcal{C}_s(\boldsymbol{W}^{*\top}\boldsymbol{u})}{2}$. Then we can define $\mathcal{L}_s(\frac{\boldsymbol{W}^{*\top}\boldsymbol{u}}{\sigma\delta_K(\boldsymbol{W}^*)}, \sigma\delta_K(\boldsymbol{W}^*)) := \frac{\mathcal{C}_s(\boldsymbol{W}^{*\top}\boldsymbol{u})}{2\zeta_s}\sigma^2$ such that $\sigma^{-1}\mathcal{L}_s(\frac{\boldsymbol{W}^{*\top}\boldsymbol{u}}{\sigma\delta_K(\boldsymbol{W}^*)}, \sigma\delta_K(\boldsymbol{W}^*))$ is an increasing function of $\sigma$ below $\rho(\frac{\boldsymbol{W}^{*\top}\boldsymbol{u}}{\sigma\delta_K(\boldsymbol{W}^*)}, \sigma\delta_K(\boldsymbol{W}^*))$. When $(\boldsymbol{W}^{*\top}\boldsymbol{u})_i = 0$ for some $i \in [K]$, then $\lim_{\sigma \to 0} \rho(\frac{\boldsymbol{W}^{*\top}\boldsymbol{u}}{\sigma\delta_K(\boldsymbol{W}^*)}, \sigma\delta_K(\boldsymbol{W}^*)) = 0$. We can derive

$$
\lim_{\sigma \to 0} \frac{\rho(\frac{\boldsymbol{W}^{*\top}\boldsymbol{u}}{\sigma\delta_K(\boldsymbol{W}^*)}, \sigma\delta_K(\boldsymbol{W}^*))}{\sigma} = \lim_{\sigma \to 0} \frac{\partial}{\partial \sigma} \rho(\frac{\boldsymbol{W}^{*\top}\boldsymbol{u}}{\sigma\delta_K(\boldsymbol{W}^*)}, \sigma\delta_K(\boldsymbol{W}^*)) \geq 0
\tag{27}
$$

The last step of (27) is because if the limit is negative, then $\rho(\frac{\boldsymbol{W}^{*\top}\boldsymbol{u}}{\sigma\delta_K(\boldsymbol{W}^*)}, \sigma\delta_K(\boldsymbol{W}^*))$ will be negative in a small neighborhood around $\sigma = 0$, which contradicts to the fact that $\rho(\frac{\boldsymbol{W}^{*\top}\boldsymbol{u}}{\sigma\delta_K(\boldsymbol{W}^*)}, \sigma\delta_K(\boldsymbol{W}^*)) > 0$.

If the limit in (27) is 0, then $\lim_{\sigma \to 0} \frac{\partial}{\partial \sigma} \frac{\rho(\frac{\boldsymbol{W}^{*\top}\boldsymbol{u}}{\sigma\delta_K(\boldsymbol{W}^*)}, \sigma\delta_K(\boldsymbol{W}^*))}{\sigma} > 0$ otherwise there will be a small neighborhood around $\sigma = 0$ in which $\frac{\rho(\frac{\boldsymbol{W}^{*\top}\boldsymbol{u}}{\sigma\delta_K(\boldsymbol{W}^*)}, \sigma\delta_K(\boldsymbol{W}^*))}{\sigma} < 0$. In this case we only need to let $\sigma^{-1}\mathcal{L}_s(\frac{\boldsymbol{W}^{*\top}\boldsymbol{u}}{\sigma\delta_K(\boldsymbol{W}^*)}, \sigma\delta_K(\boldsymbol{W}^*)) := \frac{\rho(\frac{\boldsymbol{W}^{*\top}\boldsymbol{u}}{\sigma\delta_K(\boldsymbol{W}^*)}, \sigma\delta_K(\boldsymbol{W}^*))}{\sigma}$. If the limit in (27) is positive, we can find a positive lower bound of $\frac{\rho(\frac{\boldsymbol{W}^{*\top}\boldsymbol{u}}{\sigma\delta_K(\boldsymbol{W}^*)}, \sigma\delta_K(\boldsymbol{W}^*))}{\sigma}$ in a small neighborhood around $\sigma = 0$ and an increasing function of $\sigma$, $\sigma^{-1}\mathcal{L}(\frac{\boldsymbol{W}^{*\top}\boldsymbol{u}}{\sigma\delta_K(\boldsymbol{W}^*)}, \sigma\delta_K(\boldsymbol{W}^*))$ can be defined to be less than this positive lower bound.

In conclusion, the condition (a) is proved.

**Property 3** *With the notation in (6), if a function $f(\boldsymbol{x})$ is an even function, then*

$$
\mathbb{E}_{\boldsymbol{x} \sim \mathcal{N}(\boldsymbol{\mu}, \sigma^2\boldsymbol{I}_d)}[f(\boldsymbol{x})] = \mathbb{E}_{\boldsymbol{x} \sim \frac{1}{2}\mathcal{N}(\boldsymbol{\mu}, \sigma^2\boldsymbol{I}_d) + \frac{1}{2}\mathcal{N}(-\boldsymbol{\mu}, \sigma^2\boldsymbol{I}_d)}[f(\boldsymbol{x})]
\tag{28}
$$

**Proof:**
Denote

$$
g(\boldsymbol{x}) = f(\boldsymbol{x})(2\pi\sigma^2)^{-\frac{d}{2}} \exp(-\frac{||\boldsymbol{x} - \boldsymbol{\mu}||^2}{2\sigma^2})
\tag{29}
$$

$$
\begin{aligned}
\mathbb{E}_{\boldsymbol{x} \sim \mathcal{N}(\boldsymbol{\mu}, \sigma^2 \boldsymbol{I})}[f(\boldsymbol{x})] &= \int_{\boldsymbol{x} \in \mathbb{R}^d} g(\boldsymbol{x}) d\boldsymbol{x} = \int_{-\infty}^{\infty} \cdots \int_{-\infty}^{\infty} g(x_1, \cdots, x_d) dx_1 \cdots dx_d \\
&= \int_{-\infty}^{\infty} \cdots \int_{-\infty}^{\infty} \int_{\infty}^{-\infty} g(x_1, x_2, \cdots, x_d) d(-x_1) dx_2 \cdots dx_d \\
&= \int_{-\infty}^{\infty} \cdots \int_{-\infty}^{\infty} g(-x_1, x_2 \cdots, x_d) dx_1 dx_2 \cdots dx_d \\
&= \int_{\boldsymbol{x} \in \mathbb{R}^d} g(-\boldsymbol{x}) d\boldsymbol{x} \\
&= \int_{\boldsymbol{x} \in \mathbb{R}^d} f(\boldsymbol{x})(2\pi\sigma^2)^{-\frac{d}{2}} \exp(-\frac{||\boldsymbol{x} + \boldsymbol{\mu}||^2}{2\sigma^2}) d\boldsymbol{x} \\
&= \mathbb{E}_{\boldsymbol{x} \sim \mathcal{N}(-\boldsymbol{\mu}, \sigma^2 \cdot \boldsymbol{I}_d)}[f(\boldsymbol{x})]
\end{aligned}
\tag{30}
$$

Therefore, we have

$$
\mathbb{E}_{\boldsymbol{x} \sim \mathcal{N}(\boldsymbol{\mu}, \sigma^2 \boldsymbol{I}_d)}[f(\boldsymbol{x})] = \mathbb{E}_{\boldsymbol{x} \sim \frac{1}{2}\mathcal{N}(\boldsymbol{\mu}, \sigma^2 \boldsymbol{I}_d) + \frac{1}{2}\mathcal{N}(-\boldsymbol{\mu}, \sigma^2 \boldsymbol{I}_d)}[f(\boldsymbol{x})]
\tag{31}
$$

**Property 4** *Under Gaussian Mixture Model $\boldsymbol{x} \sim \sum_{l=1}^{L} \lambda_l \mathcal{N}(\boldsymbol{\mu}_l, \sigma_l^2 \boldsymbol{I}_d)$, we have the following upper bound.*

$$
\mathbb{E}_{\boldsymbol{x} \sim \sum_{l=1}^{L} \lambda_l \mathcal{N}(\boldsymbol{\mu}_l, \sigma_l^2 \boldsymbol{I}_d)}[(\boldsymbol{u}^\top \boldsymbol{x})^{2t}] \le (2t-1)!! ||\boldsymbol{u}||^{2t} \sum_{l=1}^{L} \lambda_l (||\boldsymbol{\mu}_l||_\infty + \sigma_l)^{2t}
\tag{32}
$$

**Proof:**
The main idea is to find an upper bound with symmetric distribution assumption first, and then apply Property 3 to extend the conclusion to the general case.
(a) If the Mixed-Gaussian distribution is symmetric and $L = 2$, i.e. $\boldsymbol{x} \sim \frac{1}{2}\Big(\mathcal{N}(\boldsymbol{\mu}, \sigma^2 \boldsymbol{I}_d) + \mathcal{N}(-\boldsymbol{\mu}, \sigma^2 \boldsymbol{I}_d)\Big)$, then we first need to analyse the distribution of $\boldsymbol{u}^\top \boldsymbol{x}$ by computing the moment generating function

$$
\begin{aligned}
\mathbb{E}_{\boldsymbol{x} \sim \frac{1}{2}\big(\mathcal{N}(\boldsymbol{\mu}, \sigma^2 \boldsymbol{I}_d) + \mathcal{N}(-\boldsymbol{\mu}, \sigma^2 \boldsymbol{I}_d)\big)} & [\exp(t\boldsymbol{u}^\top \boldsymbol{x})] = \mathbb{E}[\exp(t\sum_{i=1}^{d} u_i x_i)] = \prod_{i=1}^{d} \mathbb{E}[\exp(t u_i x_i)] \\
= \prod_{i=1}^{d} \{ \sum_{j=1}^{2} & \frac{1}{2} \int_{-\infty}^{\infty} \exp(t u_i x_i) \frac{1}{\sqrt{2\pi}\sigma} \exp(-\frac{(x_i - (-1)^j \mu_i)^2}{2\sigma^2}) dx_i \} \\
= \prod_{i=1}^{d} \{ \sum_{j=1}^{2} & \frac{1}{2} \exp(t u_i (-1)^j \mu_i) \\
\cdot & \int_{-\infty}^{\infty} \exp\big(t u_i (x_i - (-1)^j \mu_i)\big) \frac{1}{\sqrt{2\pi}\sigma} \exp(-\frac{(x_i - (-1)^j \mu_i)^2}{2\sigma^2}) dx_i \} \\
= \prod_{i=1}^{d} \{ & \frac{1}{2} \exp(-t u_i \mu_i) + \frac{1}{2}\sigma^2 u_i^2 t^2 + \frac{1}{2} \exp(t u_i \mu_i + \frac{1}{2}\sigma^2 u_i^2 t^2) \} \\
:= \sum_{i=1}^{2^d} & \frac{1}{2^d} \exp(t\mu_i' + \frac{1}{2}t^2 \sigma'^2)
\end{aligned}
\tag{33}
$$

which is the Moment Generating Function of $\sum_{i=1}^{2^d} \frac{1}{2^d} \mathcal{N}(\mu_i', \sigma'^2)$. The last step of (33) is by expanding the multiplication of $d$ terms. Specifically, let $\{\boldsymbol{s}^i\}_{i=1}^{2^d}$ denote all $2^d$ vectors in $\mathbb{R}^d$ taking values from 0 and 1. Let $s_k^i$ ($k \in [d]$) denote the $k$-th entry of $\boldsymbol{s}^i$. We define $\mu_i' = \sum_{k=1}^{d} (-1)^{s_k^i} u_k \mu_k \in \mathbb{R}$ for $i \in [2^d]$, and $\sigma' = \sigma ||\boldsymbol{u}|| \in \mathbb{R}$, where $u_k$ and $\mu_k$ are the $k$-th entry of the vector $\boldsymbol{u}$ and $\boldsymbol{\mu}$, respectively. Then we can derive the first few steps of $\mathbb{E}[(\boldsymbol{u}^\top \boldsymbol{x})^{2t}]$

$$\mathbb{E}_{\boldsymbol{x} \sim \frac{1}{2}\left(\mathcal{N}(\boldsymbol{\mu}, \sigma^2 \boldsymbol{I}_d) + \mathcal{N}(-\boldsymbol{\mu}, \sigma^2 \boldsymbol{I}_d)\right)}\left[(\boldsymbol{u}^\top \boldsymbol{x})^{2t}\right]$$

$$= \int_{-\infty}^{\infty} y^{2t} \sum_{i=1}^{2^d} \frac{1}{2^d} \frac{1}{\sqrt{2\pi}\sigma'} e^{-\frac{(y-\mu_i')^2}{2\sigma'^2}} dy$$

$$= \sum_{i=1}^{2^d} \frac{1}{2^d} \int_{-\infty}^{\infty} (y - \mu_i' + \mu_i')^{2t} \frac{1}{\sqrt{2\pi}\sigma'} e^{-\frac{(y-\mu_i')^2}{2\sigma'^2}} dy$$

$$= \sum_{i=1}^{2^d} \frac{1}{2^d} \int_{-\infty}^{\infty} \sum_{p=0}^{2t} \binom{2t}{p} \mu_i'^{2t-p} (y - \mu')^p \frac{1}{\sqrt{2\pi}\sigma'} e^{-\frac{(y-\mu_i')^2}{2\sigma'^2}} dy \tag{34}$$

$$= \sum_{i=1}^{2^d} \frac{1}{2^d} \sum_{p=0}^{2t} \binom{2t}{p} \mu_i'^{2t-p} \cdot \begin{cases} 0, & p \text{ is odd} \\ (p-1)!!\sigma'^2, & p \text{ is even} \end{cases}$$

$$= \sum_{i=1}^{2^d} \frac{1}{2^d} \sum_{k=0}^{t} \binom{2t}{2k} \mu_i'^{2t-2k} \sigma'^{2k} (2k-1)!!$$

$$= \frac{1}{2^d} \sum_{k=0}^{t} \binom{2t}{2k} \sigma'^{2k} (2k-1)!! \sum_{i=1}^{2^d} \mu_i'^{2t-2k}$$

The first step is by the distribution of $\boldsymbol{u}^\top \boldsymbol{x}$ we obtain from (33). The third step follows from the binomial expansion. The forth step results from the calculation of high-order moment of Gaussian distribution. The second to last step is derived from the inverse of binomial expansion. The last step is due to the substitution of summation. To compute the inner summation in the last step of (34), we

have

$$\sum_{i=1}^{2^d} \mu_i'^{2t}$$

$$= \sum_{i=1}^{2^d} (u_1(-1)^{s_1^i}\mu_1 + u_2(-1)^{s_2^i}\mu_2 + ... + u_d(-1)^{s_d^i}\mu_d)^{2t}$$

$$= \sum_{i=1}^{2^d} \sum_{p_1^{(i)}+\cdots+p_d^{(i)}=2t} \frac{(2t)!}{p_1^{(i)}!p_2^{(i)}!...p_d^{(i)}!} (u_1(-1)^{s_1^i}\mu_1)^{p_1^{(i)}}...(u_d(-1)^{s_d^i}\mu_d)^{p_d^{(i)}}$$

$$= \sum_{i=1}^{2^d} \sum_{p_1^{(i)}+\cdots+p_d^{(i)}=2t} \frac{(2t)!}{p_1^{(i)}!p_2^{(i)}!...p_d^{(i)}!} (u_1\mu_1)^{p_1^{(i)}}...(u_d\mu_d)^{p_d^{(i)}} \qquad \text{all the } p_i \text{ are even}$$

$$= \sum_{i=1}^{2^d} \sum_{p_1^{(i)}+\cdots+p_d^{(i)}=2t} \frac{(2t)!}{p_1^{(i)}!p_2^{(i)}!...p_d^{(i)}!} (u_1^2\mu_1^2)^{q_1^{(i)}}...(u_d^2\mu_d^2)^{q_d^{(i)}} \qquad q_j^{(i)} = \frac{p_j^{(i)}}{2}$$

$$\leq \sum_{i=1}^{2^d} \max \frac{(2t)!}{p_1^{(i)}!p_2^{(i)}!...p_d^{(i)}!} \Big/ \frac{(t)!}{q_1^{(i)}!q_2^{(i)}!...q_d^{(i)}!} \} \sum_{\sum_{h=1}^d q_h^{(i)}=t} \frac{(t)!}{q_1^{(i)}!q_2^{(i)}!...q_d^{(i)}!} (u_1^2\mu_1^2)^{q_1^{(i)}}...(u_d^2\mu_d^2)^{q_d^{(i)}}$$

$$\leq \sum_{i=1}^{2^d} \max\{\frac{(2t)!}{p_1^{(i)}!p_2^{(i)}!...p_d^{(i)}!} \Big/ \frac{(t)!}{q_1^{(i)}!q_2^{(i)}!...q_d^{(i)}!}\} \cdot (u_1^2\mu_1^2 + \cdots + u_d^2\mu_d^2)^t$$

$$\leq \sum_{i=1}^{2^d} \max\{\frac{(2t)!}{p_1^{(i)}!p_2^{(i)}!...p_d^{(i)}!} \Big/ \frac{(t)!}{q_1^{(i)}!q_2^{(i)}!...q_d^{(i)}!}\} \cdot (u_1^2 + ... + u_d^2)^t \cdot \max_j\{|\mu_j|\}^{2t}$$

$$\leq 2^d \|\boldsymbol{u}\|^{2t} \cdot \max_j\{|\mu_j|\}^{2t} \cdot (2t-1)!!$$

(35)

Firstly we explain the third step. For any odd $p_\ell$, there is a term $a_0 = (u_\ell(-1)^{s_\ell^i}\mu_\ell)^{p_\ell} \cdot \prod_{k\neq\ell}(u_k(-1)^{s_k^i}\mu_k)^{p_k}$ among the expansion of $\mu_i'^{2t}$, whose corresponding vector $\boldsymbol{s}^i$ is $(s_i^i, \cdots, s_\ell^i, \cdots, s_d^i)$. We can find a $\mu_j'$ such that its corresponding vector is $(s_1^i, \cdots, 1-s_\ell^i, \cdots, s_d^i)$, which is only different from the tuple of $\mu_i'$ in the $\ell$-th entry. Therefore, in the expansion of $\mu_j'^{2t}$, there exists a term $a_0' = (u_\ell(-1)^{1-s_\ell^i}\mu_\ell)^{p_\ell} \cdot \prod_{k\neq\ell}(u_k(-1)^{s_k^i}\mu_k)^{p_k}$ that can be cancelled out by $a_0$. Therefore, there will be no odd power terms left. The third to last step of (35) is by the inverse binomial expansion. The second to last step is by the inequality $\sum_{i=1}^N a_i b_i \leq \max\{b_i\} \cdot \sum_{i=1}^N a_i$, where $a_i$ and $b_i$ are positive. The last step is because

$$\frac{(2t)!}{p_1^{(i)}!p_2^{(i)}!...p_d^{(i)}!} \Big/ \frac{(t)!}{q_1^{(i)}!q_2^{(i)}!...q_d^{(i)}!}$$

$$= \frac{(2t)!}{t!} \cdot \frac{\frac{p_{d_1}^{(i)}}{2}!\frac{p_{d_2}^{(i)}}{2}!\cdots\frac{p_{d_m}^{(i)}}{2}!}{p_{d_1}^{(i)}!p_{d_2}^{(i)}!\cdots p_{d_m}^{(i)}!}$$

$$\leq \frac{(2t)!}{t!} \cdot (\frac{1}{2})^m$$

$$\leq \frac{(2t)!}{t!} \cdot (\frac{1}{2})^t = (2t-1)!!$$

(36)

In the first equality of (36), $p_{d_1}, ..., p_{d_m}$ denote all the positive $p_i$. Thus, we have $\sum_{i=1}^m p_{d_i} = 2t$ where $p_{d_i} \geq 2$. Therefore, $m \leq \frac{2t}{2} = t$ which is used in the second inequality. Therefore, combining

(35), we can continue the derivation of (34) as follows.

$$\mathbb{E}_{\boldsymbol{x}\sim\frac{1}{2}\left(\mathcal{N}(\boldsymbol{\mu},\sigma^2\boldsymbol{I}_d)+\mathcal{N}(-\boldsymbol{\mu},\sigma^2\boldsymbol{I}_d)\right)}[(\boldsymbol{u}^\top\boldsymbol{x})^{2t}]$$

$$=\frac{1}{2^d}\sum_{k=0}^{t}\binom{2t}{2k}\sigma_i'^{2k}(2k-1)!!\sum_{i=1}^{2^d}\mu_i'^{2t-2k}$$

$$\leq\sum_{k=0}^{t}\binom{2t}{2k}\cdot\sigma'^{2k}(2k-1)!!||\boldsymbol{u}||^{2t}\cdot\max_j\{|\mu_j|\}^{2t-2k}(2t-1-2k)!!$$

$$\leq(2t-1)!!||\boldsymbol{u}||^{2t}(||\boldsymbol{\mu}||_\infty+\sigma')^{2t} \tag{37}$$

The last step is because that

$$(2t-1-2k)!!(2k-1)!!\leq(2t-1-2k)!!\underbrace{(2t-1)(2t-3)\cdots(2t-2k+1)}_{k\text{ terms}}=(2t-1)!!$$

(b) From Property 3, since that $(\boldsymbol{u}^\top\boldsymbol{x})^{2t}$ is an even function, we have a result for a general Gaussian distribution

$$\mathbb{E}_{\boldsymbol{x}\sim\mathcal{N}(\boldsymbol{\mu},\sigma^2\boldsymbol{I}_d)}[(\boldsymbol{u}^\top\boldsymbol{x})^{2t}]=\mathbb{E}_{\boldsymbol{x}\sim\frac{1}{2}\mathcal{N}(\boldsymbol{\mu},\sigma^2\boldsymbol{I}_d)+\frac{1}{2}\mathcal{N}(-\boldsymbol{\mu},\sigma^2\boldsymbol{I}_d)}[(\boldsymbol{u}^\top\boldsymbol{x})^{2t}]$$

$$\leq(2t-1)!!||\boldsymbol{u}||^{2t}(||\boldsymbol{\mu}||_\infty+\sigma)^{2t} \tag{38}$$

Therefore, if there are $L$ components in the Gaussian Mixture Model, then

$$\mathbb{E}_{\boldsymbol{x}\sim\sum_{l=1}^{L}\lambda_l\mathcal{N}(\boldsymbol{\mu}_l,\sigma_l^2\boldsymbol{I}_d)}[(\boldsymbol{u}^\top\boldsymbol{x})^{2t}]\leq(2t-1)!!||\boldsymbol{u}||^{2t}\sum_{l=1}^{L}\lambda_l(||\boldsymbol{\mu}_l||_\infty+\sigma_l)^{2t} \tag{39}$$

**Property 5** *With Gaussian Mixture Model (7), we have*

$$\mathbb{E}_{\boldsymbol{x}\sim\sum_{l=1}^{L}\lambda_l\mathcal{N}(\boldsymbol{\mu}_l,\sigma_l^2\boldsymbol{I}_d)}[||\boldsymbol{x}||^{2t}]\leq d^t(2t-1)!!\sum_{l=1}^{L}\lambda_l(||\boldsymbol{\mu}_l||_\infty+\sigma_l)^{2t} \tag{40}$$

**Proof:**

$$\mathbb{E}_{\boldsymbol{x}\sim\sum_{l=1}^{L}\lambda_l\mathcal{N}(\boldsymbol{\mu}_l,\sigma_l^2\boldsymbol{I}_d)}[||\boldsymbol{x}||_2^{2t}]$$

$$=\mathbb{E}_{\boldsymbol{x}\sim\sum_{l=1}^{L}\lambda_l\mathcal{N}(\boldsymbol{\mu}_l,\sigma_l^2\boldsymbol{I}_d)}[(\sum_{i=1}^{d}x_i^2)^t]$$

$$=\mathbb{E}_{\boldsymbol{x}\sim\sum_{l=1}^{L}\lambda_l\mathcal{N}(\boldsymbol{\mu}_l,\sigma_l^2\boldsymbol{I}_d)}[d^t(\sum_{i=1}^{d}\frac{x_i^2}{d})^t]$$

$$\leq\mathbb{E}_{\boldsymbol{x}\sim\sum_{l=1}^{L}\lambda_l\mathcal{N}(\boldsymbol{\mu}_l,\sigma_l^2\boldsymbol{I}_d)}[d^t\sum_{i=1}^{d}\frac{x_i^{2t}}{d}]$$

$$=d^{t-1}\sum_{i=1}^{d}\sum_{j=1}^{L}\int_{-\infty}^{\infty}(x_i-\mu_{ji}+\mu_{ji})^{2t}\lambda_j\frac{1}{\sqrt{2\pi}\sigma}\exp(-\frac{(x_i-\mu_{ji})^2}{2\sigma_j^2})dx_i \tag{41}$$

$$=d^{t-1}\sum_{i=1}^{d}\sum_{j=1}^{L}\sum_{k=1}^{2t}\binom{2t}{k}\lambda_j|\mu_{ji}|^{2t-k}\cdot\begin{cases}0, & k\text{ is odd}\\(k-1)!!\sigma_j^k, & k\text{ is even}\end{cases}$$

$$\leq d^{t-1}\sum_{i=1}^{d}\sum_{j=1}^{L}\sum_{k=1}^{2t}\binom{2t}{k}\lambda_j|\mu_{ji}|^{2t-k}\sigma_j^k\cdot(2t-1)!!$$

$$=d^{t-1}\sum_{i=1}^{d}\sum_{j=1}^{L}\lambda_j(|\mu_{ji}|+\sigma_j)^{2t}(2t-1)!!$$

$$\leq d^t(2t-1)!!\sum_{l=1}^{L}\lambda_l(||\boldsymbol{\mu}||_\infty+\sigma_l)^{2t}$$

In the 3rd step, we apply Jensen inequality because $f(x) = x^t$ is convex when $x \geq 0$ and $t \geq 1$. In the 4th step we apply the Binomial theorem and the result of k-order central moment of Gaussian variable.

**Property 6** *The population risk function $f(\boldsymbol{W})$ is defined as*

$$f(\boldsymbol{W}) = \mathbb{E}_{\boldsymbol{x} \sim \sum_{l=1}^{L} \lambda_l \mathcal{N}(\boldsymbol{\mu}_l, \sigma_l^2 \boldsymbol{I}_d)}[f_n(\boldsymbol{W})]$$

$$= \mathbb{E}_{\boldsymbol{x} \sim \sum_{l=1}^{L} \lambda_l \mathcal{N}(\boldsymbol{\mu}_l, \sigma_l^2 \boldsymbol{I}_d)} \Big[\frac{1}{n} \sum_{i=1}^{n} \ell(\boldsymbol{W}; \boldsymbol{x}_i, y_i)\Big] \tag{42}$$

$$= \mathbb{E}_{\boldsymbol{x} \sim \sum_{l=1}^{L} \lambda_l \mathcal{N}(\boldsymbol{\mu}_l, \sigma_l^2 \boldsymbol{I}_d)}[\ell(\boldsymbol{W}; \boldsymbol{x}_i, y_i)]$$

*Based on (2), (3) and (4), we can derive its gradient and Hessian as follows.*

$$\frac{\partial \ell(\boldsymbol{W}; \boldsymbol{x}, y)}{\partial \boldsymbol{w}_j} = -\frac{1}{K} \frac{y - H(\boldsymbol{W})}{H(\boldsymbol{W})(1 - H(\boldsymbol{W}))} \phi'(\boldsymbol{w}_j^\top \boldsymbol{x}) \boldsymbol{x} = \zeta(\boldsymbol{W}) \cdot \boldsymbol{x} \tag{43}$$

$$\frac{\partial^2 \ell(\boldsymbol{W}; \boldsymbol{x}, y)}{\partial \boldsymbol{w}_j \partial \boldsymbol{w}_l} = \xi_{j,l} \cdot \boldsymbol{x} \boldsymbol{x}^\top \tag{44}$$

$$\xi_{j,l}(\boldsymbol{W}) = \begin{cases} \frac{1}{K^2} \phi'(\boldsymbol{w}_j^\top \boldsymbol{x}) \phi'(\boldsymbol{w}_l^\top \boldsymbol{x}) \frac{H(\boldsymbol{W})^2 + y - 2y \cdot H(\boldsymbol{W})}{H^2(\boldsymbol{W})(1 - H(\boldsymbol{W}))^2}, & j \neq l \\ \frac{1}{K^2} \phi'(\boldsymbol{w}_j^\top \boldsymbol{x}) \phi'(\boldsymbol{w}_l^\top \boldsymbol{x}) \frac{H(\boldsymbol{W})^2 + y - 2y \cdot H(\boldsymbol{W})}{H^2(\boldsymbol{W})(1 - H(\boldsymbol{W}))^2} - \frac{1}{K} \phi''(\boldsymbol{w}_j^\top \boldsymbol{x}) \frac{y - H(\boldsymbol{W})}{H(\boldsymbol{W})(1 - H(\boldsymbol{W}))}, & j = l \end{cases} \tag{45}$$

**Property 7** *With $D_m(\boldsymbol{\lambda}, \boldsymbol{M}, \boldsymbol{\sigma})$ defined in definition 5, we have*

$$(i) \ D_m(\boldsymbol{\lambda}, \boldsymbol{M}, \boldsymbol{\sigma}) D_{2m}(\boldsymbol{\lambda}, \boldsymbol{M}, \boldsymbol{\sigma}) \leq D_{3m}(\boldsymbol{\lambda}, \boldsymbol{M}, \boldsymbol{\sigma}) \tag{46}$$

$$(ii) \ \big(D_m(\boldsymbol{\lambda}, \boldsymbol{M}, \boldsymbol{\sigma})\big)^2 \leq D_{2m}(\boldsymbol{\lambda}, \boldsymbol{M}, \boldsymbol{\sigma}) \tag{47}$$

**Proof:**
To prove (46), we can first compare the terms $\sum_{i=1}^{L} \lambda_i a_i \sum_{i=1}^{L} \lambda_i a_i^2$ and $\sum_{i=1}^{L} \lambda_i a_i^3$, where $a_i \geq 1$, $i \in [L]$ and $\sum_{i=1}^{L} \lambda_i = 1$.

$$\begin{aligned}
\sum_{i=1}^{L} \lambda_i a_i^3 - \sum_{i=1}^{L} \lambda_i a_i \sum_{i=1}^{L} \lambda_i a_i^2 &= \sum_{i=1}^{L} \lambda_i a_i \cdot \big(a_i^2 - \sum_{j=1}^{L} \lambda_j a_j^2\big) \\
&= \sum_{i=1}^{L} \lambda_i a_i \cdot \big((1 - \lambda_i) a_i^2 - \sum_{1 \leq j \leq L, j \neq i} \lambda_j a_j^2\big) \\
&= \sum_{i=1}^{L} \lambda_i a_i \cdot \big(\sum_{1 \leq j \leq L, j \neq i} \lambda_j a_i^2 - \sum_{1 \leq j \leq L, j \neq i} \lambda_j a_j^2\big) \\
&= \sum_{i=1}^{L} \lambda_i a_i \cdot \big(\sum_{1 \leq j \leq L, j \neq i} \lambda_j (a_i^2 - a_j^2)\big) \\
&= \sum_{1 \leq i,j \leq L, i \neq j} \big(\lambda_i \lambda_j a_i(a_i^2 - a_j^2) + \lambda_i \lambda_j a_j(a_j^2 - a_i^2)\big) \\
&= \sum_{1 \leq i,j \leq L, i \neq j} \lambda_i \lambda_j (a_i - a_j)^2 (a_i + a_j) \geq 0
\end{aligned} \tag{48}$$

The second to last step is because we can find the pairwise terms $\lambda_i a_i \cdot \lambda_j(a_i^2 - a_j^2)$ and $\lambda_j a_j \cdot \lambda_i(a_j^2 - a_i^2)$ in the summation that can be putted together. From (48), we can obtain

$$\sum_{i=1}^{L} \lambda_i a_i \sum_{i=1}^{L} \lambda_i a_i^2 \leq \sum_{i=1}^{L} \lambda_i a_i^3 \tag{49}$$

Combining (49) and the definition of $D_m(\boldsymbol{\lambda}, \boldsymbol{M}, \boldsymbol{\sigma})$ in (5), we can derive (46).

Similarly, to prove (47), we can first compare the terms $(\sum_{i=1}^{L} \lambda_i a_i)^2$ and $\sum_{i=1}^{L} \lambda_i a_i^2$, where $a_i \geq 1$, $i \in [L]$ and $\sum_{i=1}^{L} \lambda_i = 1$.

$$
\begin{aligned}
\sum_{i=1}^{L} \lambda_i a_i^2 - (\sum_{i=1}^{L} \lambda_i a_i)^2 &= \sum_{i=1}^{L} \lambda_i a_i \cdot \big( a_i - \sum_{j=1}^{L} \lambda_j a_j \big) \\
&= \sum_{i=1}^{L} \lambda_i a_i \cdot \big( (1-\lambda_i) a_i - \sum_{1 \leq j \leq L, j \neq i} \lambda_j a_j \big) \\
&= \sum_{i=1}^{L} \lambda_i a_i \cdot \big( \sum_{1 \leq j \leq L, j \neq i} \lambda_j a_i - \sum_{1 \leq j \leq L, j \neq i} \lambda_j a_j \big) \\
&= \sum_{i=1}^{L} \lambda_i a_i \cdot \big( \sum_{1 \leq j \leq L, j \neq i} \lambda_j (a_i - a_j) \big) \\
&= \sum_{1 \leq i,j \leq L, i \neq j} \big( \lambda_i \lambda_j a_i (a_i - a_j) + \lambda_i \lambda_j a_j (a_j - a_i) \big) \\
&= \sum_{1 \leq i,j \leq L, i \neq j} \lambda_i \lambda_j (a_i - a_j)^2 \geq 0
\end{aligned}
\tag{50}
$$

The derivation of (50) is close to (48). By (50) we have

$$
(\sum_{i=1}^{L} \lambda_i a_i)^2 \leq \sum_{i=1}^{L} \lambda_i a_i^2
\tag{51}
$$

Combining (51) and the definition of $D_m(\boldsymbol{\lambda}, \boldsymbol{M}, \boldsymbol{\sigma})$ in (5), we can derive (47).

## B  PROOF OF THEOREM 1

Theorem 1 is built upon three lemmas. Lemma 1 shows that with $O(dK^5 \log^2 d)$ samples, the empirical risk function is strongly convex in the neighborhood of $\boldsymbol{W}^*$. Lemma 2 shows that if initialized in the convex region, that the gradient descent algorithm converges linearly to a critical point $\widehat{\boldsymbol{W}}_n$, that is close to $\boldsymbol{W}^*$. Lemma 3 shows that the Tensor Initialization Method in Subroutine 1 initializes $\boldsymbol{W}_0 \in \mathbb{R}^{d \times K}$ in the local convex region. Theorem 1 follows naturally by combining these three lemmas.

This proving approach is built upon those in (Fu et al., 2020). One of our major technical contribution is extending Lemmas 1 and 2 to the Gaussian mixture model, while the results in (Fu et al., 2020) only apply to Standard Gaussian models. The second major contribution is a new tensor initialization method for Gaussian mixture model such that the initial point is in the convex region (see Lemma 3). Both contributions require the development of new tools, and our analyses are much more involved than those for the standard Gaussian due to the complexity introduced by the Gaussian mixture model.

To present these lemmas, the Euclidean ball $\mathbb{B}(\boldsymbol{W}^*, r)$ is used to denote the neighborhood of $\boldsymbol{W}^*$, where $r$ is the radius of the ball.

$$
\mathbb{B}(\boldsymbol{W}^*, r) = \{ \boldsymbol{W} \in \mathbb{R}^{d \times K} : ||\boldsymbol{W} - \boldsymbol{W}^*||_F \leq r \}
\tag{52}
$$

The radius of the convex region is

$$
r := \Theta \Bigg( \frac{C_3 \epsilon_0 \cdot \sum_{l=1}^{L} \lambda_l \frac{\sigma_l^2}{\eta \kappa^2} \rho(\frac{\boldsymbol{W}^{*\top} \boldsymbol{\mu}_l}{\sigma_l \delta_K(\boldsymbol{W}^*)}, \sigma_l \delta_K(\boldsymbol{W}^*))}{K^{\frac{7}{2}} \Big( \sum_{l=1}^{L} \lambda_l (\|\boldsymbol{\mu}_l\|_\infty + \sigma_l)^4 \sum_{l=1}^{L} \lambda_l (\|\boldsymbol{\mu}_l\|_\infty + \sigma_l)^8 \Big)^{\frac{1}{4}}} \Bigg)
\tag{53}
$$

with some constant $C_3 > 0$.

**Lemma 1** *(Strongly local convexity) Consider the classification model with FCN (2) and the sigmoid activation function. There exists a constant C such that as long as the sample size*

$$n \geq C_1 \epsilon_0^{-2} \cdot \Big( \sum_{l=1}^{L} \lambda_l(\|\boldsymbol{\mu}\|_\infty + \sigma_l)^2 \Big)^2 \Big( \sum_{l=1}^{L} \lambda_l \frac{\sigma_l^2}{\eta \kappa^2} \rho\big( \frac{\boldsymbol{W}^{*\top} \boldsymbol{\mu}_l}{\sigma_l \delta_K(\boldsymbol{W}^*)}, \sigma_l \delta_K(\boldsymbol{W}^*) \big) \Big)^{-2} dK^5 \log^2 d \tag{54}$$

*for some constant $C_1 > 0$ and $\epsilon_0 \in (0, \frac{1}{4})$, we have for all $\boldsymbol{W} \in \mathbb{B}(\boldsymbol{W}^*, r_{FCN})$,*

$$\Omega\Big( \frac{1 - 2\epsilon_0}{K^2} \sum_{l=1}^{L} \lambda_l \frac{\sigma_l^2}{\eta \kappa^2} \rho\big( \frac{\boldsymbol{W}^{*\top} \boldsymbol{\mu}_l}{\sigma_l \delta_K(\boldsymbol{W}^*)}, \sigma_l \delta_K(\boldsymbol{W}^*) \big) \Big) \cdot \boldsymbol{I}_{dK}$$

$$\preceq \nabla^2 f_n(\boldsymbol{W}) \preceq C_2 \sum_{l=1}^{L} \lambda_l(\|\boldsymbol{\mu}_l\|_\infty + \sigma_l)^2 \cdot \boldsymbol{I}_{dK} \tag{55}$$

*with probability at least $1 - d^{-10}$ for some constant $C_2 > 0$.*

**Lemma 2** *(Linear convergence of gradient descent) Assume the conditions in Lemma 1 hold. If the local convexity holds, there exists a critical point in $\mathbb{B}(\boldsymbol{W}^*, r)$ for some constant $C_3 > 0$ and $\epsilon_0 \in (0, \frac{1}{2})$, such that*

$$\|\widehat{\boldsymbol{W}}_n - \boldsymbol{W}^*\|_F \leq O\Big( \frac{K^{\frac{5}{2}} \sqrt{\sum_{l=1}^{L} \lambda_l(\|\boldsymbol{\mu}\|_\infty + \sigma_l)^2}}{\sum_{l=1}^{L} \lambda_l \frac{\sigma_l^2}{\eta \kappa^2} \rho\big( \frac{\boldsymbol{W}^{*\top} \boldsymbol{\mu}_l}{\sigma_l \delta_K(\boldsymbol{W}^*)}, \sigma_l \delta_K(\boldsymbol{W}^*) \big)} \sqrt{\frac{d \log n}{n}} \Big) \tag{56}$$

*If the initial point $\boldsymbol{W}_0 \in \mathbb{B}(\boldsymbol{W}^*, r)$, the gradient descent linearly converges to $\widehat{\boldsymbol{W}}_n$, i.e.,*

$$\|\boldsymbol{W}_t - \widehat{\boldsymbol{W}}_n\|_F \leq \Big( 1 - \Omega\big( \frac{\sum_{l=1}^{L} \lambda_l \frac{\sigma_l^2}{\eta \kappa^2} \rho\big( \frac{\boldsymbol{W}^{*\top} \boldsymbol{\mu}_l}{\sigma_l \delta_K(\boldsymbol{W}^*)}, \sigma_l \delta_K(\boldsymbol{W}^*) \big)}{K^2 \sum_{l=1}^{L} \lambda_l(\|\boldsymbol{\mu}_l\|_\infty + \sigma_l)^2} \big) \Big)^t \|\boldsymbol{W}_0 - \widehat{\boldsymbol{W}}_n\|_F \tag{57}$$

*with probability at least $1 - d^{-10}$.*

**Lemma 3** *(Tensor initialization) For classification model, with $D_6(\boldsymbol{\lambda}, \boldsymbol{M}, \boldsymbol{\sigma})$ defined in Definition 5, we have that if the sample size*

$$n \geq \kappa^8 K^4 \tau^{12} D_6(\boldsymbol{\lambda}, \boldsymbol{M}, \boldsymbol{\sigma}) \cdot d \log^2 d, \tag{58}$$

*then the output $\boldsymbol{W}_0 \in \mathbb{R}^{d \times K}$ satisfies*[3]

$$\|\boldsymbol{W}_0 - \boldsymbol{W}^*\| \lesssim \kappa^6 K^3 \cdot \tau^6 \sqrt{D_6(\boldsymbol{\lambda}, \boldsymbol{M}, \boldsymbol{\sigma})} \sqrt{\frac{d \log n}{n}} \|\boldsymbol{W}^*\| \tag{59}$$

*with probability at least $1 - n^{-\Omega(\delta_1^4)}$*

**Proof of Theorem 1 and Corollary 1:**
From Lemma 2 and Lemma 3, we know that if $n$ is sufficiently large such that the initialization $\boldsymbol{W}_0$ by the tensor method is in the region $\mathbb{B}(\boldsymbol{W}^*, r)$, then the gradient descent method converges to a critical point $\widehat{\boldsymbol{W}}_n$ that is sufficiently close to $\boldsymbol{W}^*$. To achieve that, one sufficient condition is

$$\|\boldsymbol{W}_0 - \boldsymbol{W}^*\|_F \leq \sqrt{K} \|\boldsymbol{W}_0 - \boldsymbol{W}^*\| \leq \kappa^6 K^{\frac{7}{2}} \cdot \tau^6 \sqrt{D_6(\boldsymbol{\lambda}, \boldsymbol{M}, \boldsymbol{\sigma})} \sqrt{\frac{d \log n}{n}} \|\boldsymbol{W}^*\|$$

$$\leq \frac{C_3 \epsilon_0 \Gamma(\boldsymbol{\lambda}, \boldsymbol{M}, \boldsymbol{\sigma}, \boldsymbol{W}^*) \sigma_{\max}^2}{K^{\frac{7}{2}} \Big( \sum_{l=1}^{L} \lambda_l(\|\boldsymbol{\mu}_l\|_\infty + \sigma_l)^4 \sum_{l=1}^{L} \lambda_l(\|\boldsymbol{\mu}_l\|_\infty + \sigma_l)^8 \Big)^{\frac{1}{4}}} \tag{60}$$

where the first inequality follows from $\|\boldsymbol{W}\|_F \leq \sqrt{K} \|\boldsymbol{W}\|$ for $\boldsymbol{W} \in \mathbb{R}^{d \times K}$, the second inequality comes from Lemma 3, and the third inequality comes from the requirement to be in the region

---

[3] $\sigma_{\min}$ and $\sigma_{\max}$ denote the minimum and maximum among $\{\sigma_1, \cdots, \sigma_L\}$, respectively. $\tau = \frac{\sigma_{\max}}{\sigma_{\min}}$

$\mathbb{B}(\boldsymbol{W}^*, r)$. That is equivalent to the following condition

$$
\begin{aligned}
n \geq & C_0 \epsilon_0^{-2} \cdot \tau^{12} \kappa^{12} K^{14} \Big( \sum_{l=1}^{L} \lambda_l (\|\boldsymbol{\mu}_l\|_\infty + \sigma_l)^4 \sum_{l=1}^{L} \lambda_l (\|\boldsymbol{\mu}_l\|_\infty + \sigma_l)^8 \Big)^{\frac{1}{2}} (\delta_1(\boldsymbol{W}^*))^2 D_6(\boldsymbol{\lambda}, \boldsymbol{M}, \boldsymbol{\sigma}) \\
& \cdot \Gamma(\boldsymbol{\lambda}, \boldsymbol{M}, \boldsymbol{\sigma}, \boldsymbol{W}^*)^{-2} \sigma_{\max}^{-4} \cdot d \log^2 d
\end{aligned}
\tag{61}
$$

where $C_0 = \max\{C_4, C_3^{-2}\}$. By the definition 5, we can obtain

$$
\Big( \sum_{l=1}^{L} \lambda_l (\|\boldsymbol{\mu}_l\|_\infty + \sigma_l)^4 \sum_{l=1}^{L} \lambda_l (\|\boldsymbol{\mu}_l\|_\infty + \sigma_l)^8 \Big)^{\frac{1}{2}} \leq \sqrt{D_4(\boldsymbol{\lambda}, \boldsymbol{M}, \boldsymbol{\sigma}) D_8(\boldsymbol{\lambda}, \boldsymbol{M}, \boldsymbol{\sigma})} \sigma_{\max}^6 \tag{62}
$$

From Property 7, we have that

$$
\begin{aligned}
& \sqrt{D_4(\boldsymbol{\lambda}, \boldsymbol{M}, \boldsymbol{\sigma}) D_8(\boldsymbol{\lambda}, \boldsymbol{M}, \boldsymbol{\sigma})} D_6(\boldsymbol{\lambda}, \boldsymbol{M}, \boldsymbol{\sigma}) \\
& \leq \sqrt{D_{12}(\boldsymbol{\lambda}, \boldsymbol{M}, \boldsymbol{\sigma})} \sqrt{D_{12}(\boldsymbol{\lambda}, \boldsymbol{M}, \boldsymbol{\sigma})} = D_{12}(\boldsymbol{\lambda}, \boldsymbol{M}, \boldsymbol{\sigma})
\end{aligned}
\tag{63}
$$

Plugging (62), (63) into (61), we have

$$
n \geq C_0 \epsilon_0^{-2} \cdot \kappa^{12} K^{14} (\sigma_{\max} \delta_1(\boldsymbol{W}^*))^2 \tau^{12} \Gamma(\boldsymbol{\lambda}, \boldsymbol{M}, \boldsymbol{\sigma}, \boldsymbol{W}^*)^{-2} D_{12}(\boldsymbol{\lambda}, \boldsymbol{M}, \boldsymbol{\sigma}) \cdot d \log^2 d \tag{64}
$$

Considering the requirements on the sample complexity in (54), (58) and (64), (64) shows a sufficient number of samples. Taking the union bound of all the failure probabilities in Lemma 1, and 3, (64) holds with probability $1 - d^{-10}$.

By Property 2.3, $\rho(\frac{\boldsymbol{W}^{*\top} \boldsymbol{\mu}_l}{\sigma_l \delta_K(\boldsymbol{W}^*)}, \sigma_l \delta_K(\boldsymbol{W}^*))$ can be lower bounded by positive and monotonically decreasing functions $\mathcal{L}_m(\frac{\boldsymbol{W}^{*\top} \boldsymbol{\mu}_l}{\sigma_l \delta_K(\boldsymbol{W}^*)}, \sigma_l \delta_K(\boldsymbol{W}^*))$ when everything else except $|\mu_{l(i)}|$ is fixed, or $\mathcal{L}_s(\frac{\boldsymbol{W}^{*\top} \boldsymbol{\mu}_l}{\sigma_l \delta_K(\boldsymbol{W}^*)}, \sigma_l \delta_K(\boldsymbol{W}^*))$ when everything else except $\sigma_l$ is fixed. Then, by substituting the lower bound of $\rho(\frac{\boldsymbol{W}^{*\top} \boldsymbol{\mu}_l}{\sigma_l \delta_K(\boldsymbol{W}^*)}, \sigma_l \delta_K(\boldsymbol{W}^*))$ for itself in $\Gamma(\boldsymbol{\lambda}, \boldsymbol{M}, \boldsymbol{\sigma}, \boldsymbol{W}^*)$, we can have an upper bound of $(\sigma_{\max} \delta_1(\boldsymbol{W}^*))^2 \tau^{12} \Gamma(\boldsymbol{\lambda}, \boldsymbol{M}, \boldsymbol{\sigma}, \boldsymbol{W}^*)^{-2} D_{12}(\boldsymbol{\lambda}, \boldsymbol{M}, \boldsymbol{\sigma})$, denoted as $\mathcal{B}(\boldsymbol{\lambda}, \boldsymbol{M}, \boldsymbol{\sigma}, \boldsymbol{W}^*)$.

To be more specific, when everything else except $|\mu_{l(i)}|$ is fixed, $\mathcal{L}_m(\frac{\boldsymbol{W}^{*\top} \boldsymbol{\mu}_l}{\sigma_l \delta_K(\boldsymbol{W}^*)}, \sigma_l \delta_K(\boldsymbol{W}^*))$ is plugged in $\mathcal{B}(\boldsymbol{\lambda}, \boldsymbol{M}, \boldsymbol{\sigma}, \boldsymbol{W}^*)$. Then since that $D_{12}(\boldsymbol{\lambda}, \boldsymbol{M}, \boldsymbol{\sigma}, \boldsymbol{W}^*)$ and $\mathcal{L}_m(\frac{\boldsymbol{W}^{*\top} \boldsymbol{\mu}_l}{\sigma_l \delta_K(\boldsymbol{W}^*)}, \sigma_l \delta_K(\boldsymbol{W}^*))^{-2}$ are both increasing function of $\mu_{l(i)}$, $\mathcal{B}(\boldsymbol{\lambda}, \boldsymbol{M}, \boldsymbol{\sigma}, \boldsymbol{W}^*)$ is an increasing function of $|\mu_{l(i)}|$.

When everything else except $\sigma_l$ is fixed, if $\sigma_l = \sigma_{\max} > \zeta_s$, then $\sigma_{\max}^2 \tau^{12} D_{12}(\boldsymbol{\lambda}, \boldsymbol{M}, \boldsymbol{\sigma}, \boldsymbol{W}^*)$ is an increasing function of $\sigma_l$. Since that $\mathcal{L}_s(\frac{\boldsymbol{W}^{*\top} \boldsymbol{\mu}_l}{\sigma_l \delta_K(\boldsymbol{W}^*)}, \sigma_l \delta_K(\boldsymbol{W}^*))$ is a decreasing function, $\mathcal{L}_s(\frac{\boldsymbol{W}^{*\top} \boldsymbol{\mu}_l}{\sigma_l \delta_K(\boldsymbol{W}^*)}, \sigma_l \delta_K(\boldsymbol{W}^*))^{-2}$ is an increasing function of $\sigma_l$. Hence, $\mathcal{B}(\boldsymbol{\lambda}, \boldsymbol{M}, \boldsymbol{\sigma}, \boldsymbol{W}^*)$ is an increasing function of $\sigma_l$. Moreover, when all $\sigma_l < \zeta_{s'}$ and go to 0, two decreasing functions of $\sigma_l$, $\sigma_{\max}^2 \mathcal{L}_s(\frac{\boldsymbol{W}^{*\top} \boldsymbol{\mu}_l}{\sigma_l \delta_K(\boldsymbol{W}^*)}, \sigma_l \delta_K(\boldsymbol{W}^*))^{-2}$ and $D_{12}(\boldsymbol{\lambda}, \boldsymbol{M}, \boldsymbol{\sigma})$ will be the dominant term of $\mathcal{B}(\boldsymbol{\lambda}, \boldsymbol{M}, \boldsymbol{\sigma}, \boldsymbol{W}^*)$. Therefore, $\mathcal{B}(\boldsymbol{\lambda}, \boldsymbol{M}, \boldsymbol{\sigma}, \boldsymbol{W}^*)$ increases to infinity as all $\sigma_l$'s go to 0. Hence, we have

$$
n \geq poly(\epsilon_0^{-1}, \kappa, K) \mathcal{B}(\boldsymbol{\lambda}, \boldsymbol{M}, \boldsymbol{\sigma}, \boldsymbol{W}^*) \cdot d \log^2 d \tag{65}
$$

Similarly, by replacing $\rho(\frac{\boldsymbol{W}^{*\top} \boldsymbol{\mu}_l}{\sigma_l \delta_K(\boldsymbol{W}^*)}, \sigma_l \delta_K(\boldsymbol{W}^*))$ with $\mathcal{L}_m(\frac{\boldsymbol{W}^{*\top} \boldsymbol{\mu}_l}{\sigma_l \delta_K(\boldsymbol{W}^*)}, \sigma_l \delta_K(\boldsymbol{W}^*))$ when everything else except $|\boldsymbol{\mu}_{l(i)}|$ is fixed, or $\mathcal{L}_s(\frac{\boldsymbol{W}^{*\top} \boldsymbol{\mu}_l}{\sigma_l \delta_K(\boldsymbol{W}^*)}, \sigma_l \delta_K(\boldsymbol{W}^*))$ (or $\sigma^{-2} \mathcal{L}_s(\frac{\boldsymbol{W}^{*\top} \boldsymbol{\mu}_l}{\sigma_l \delta_K(\boldsymbol{W}^*)}, \sigma_l \delta_K(\boldsymbol{W}^*))$ for $\sigma \geq 1$) when everything else except $\sigma_l$ is fixed, (57) can also be transferred to another feasible upper bound. We denote the modified version of the convergence rate as $v = 1 - K^{-2} q(\boldsymbol{\lambda}, \boldsymbol{M}, \boldsymbol{\sigma}, \boldsymbol{W}^*)$. Since that $q(\boldsymbol{\lambda}, \boldsymbol{M}, \boldsymbol{\sigma}, \boldsymbol{W}^*)$ is a ratio between the smallest and the largest singular value of $\nabla^2 f(\boldsymbol{W}^*)$, we have $q(\boldsymbol{\lambda}, \boldsymbol{M}, \boldsymbol{\sigma}, \boldsymbol{W}^*) \in (0, 1)$. Hence, we can obtain $1 - K^{-2} q(\boldsymbol{\lambda}, \boldsymbol{M}, \boldsymbol{\sigma}, \boldsymbol{W}^*) \in (0, 1)$ by $K \geq 1$. When everything else except $|\mu_{l(i)}|$ is fixed, since that $\mathcal{L}_m(\frac{\boldsymbol{W}^{*\top} \boldsymbol{\mu}_l}{\sigma_l \delta_K(\boldsymbol{W}^*)}, \sigma_l \delta_K(\boldsymbol{W}^*))$ is monotonically decreasing and $\sum_{l=1}^{L} \lambda (\|\boldsymbol{\mu}\|_\infty + \sigma_l)^2$ is increasing as $|\mu_{l(i)}|$ increases, $v$ is an increasing function of $|\mu_{l(i)}|$ to 1. Similarly, when everything else except $\sigma_l$ is fixed where $\sigma_l \geq \max\{1, \zeta_s\}$, $\frac{1}{\sum_{l=1}^{L} \lambda_l (\|\boldsymbol{\mu}_l\|_\infty + \sigma_l)^2}$ decreases to 0 as $\sigma_l$ increases. We replace $\rho(\frac{\boldsymbol{W}^{*\top} \boldsymbol{\mu}_l}{\sigma_l \delta_K(\boldsymbol{W}^*)}, \sigma_l \delta_K(\boldsymbol{W}^*))$ by $\sigma^{-2} \mathcal{L}_s(\frac{\boldsymbol{W}^{*\top} \boldsymbol{\mu}_l}{\sigma_l \delta_K(\boldsymbol{W}^*)}, \sigma_l \delta_K(\boldsymbol{W}^*))$ and then

$\sigma^2 \cdot \sigma^{-2} \mathcal{L}_s(\frac{\boldsymbol{W}^{*\top}\boldsymbol{\mu}_l}{\sigma_l \delta_K(\boldsymbol{W}^*)}, \sigma_l \delta_K(\boldsymbol{W}^*)) = \mathcal{L}_s(\frac{\boldsymbol{W}^{*\top}\boldsymbol{\mu}_l}{\sigma_l \delta_K(\boldsymbol{W}^*)}, \sigma_l \delta_K(\boldsymbol{W}^*))$ is an decreasing function less than $\rho(\frac{\boldsymbol{W}^{*\top}\boldsymbol{\mu}_l}{\sigma_l \delta_K(\boldsymbol{W}^*)}, \sigma_l \delta_K(\boldsymbol{W}^*))$. Therefore, $v$ is an increasing function of $\sigma_l$ to 1 when $\sigma_l \geq \max\{1, \zeta_s\}$. When everything else except all $\sigma_l \leq \zeta_{s'}$'s go to 0, all $\mathcal{L}_s(\frac{\boldsymbol{W}^{*\top}\boldsymbol{\mu}_l}{\sigma_l \delta_K(\boldsymbol{W}^*)}, \sigma_l \delta_K(\boldsymbol{W}^*))$'s and $\frac{\sigma_l^2}{\sum_{l=1}^L \lambda_l (\|\boldsymbol{\mu}_l\|_\infty + \sigma_l)^2}$'s decrease to 0. Therefore, $v$ increases to 1.

The bound of $\|\widehat{\boldsymbol{W}}_n - \boldsymbol{W}^*\|_F$ is directly from (56).

## C    PROOF OF LEMMA 1

We first state some important lemmas used in proof in Section C.1 and describe the proof in Section C.2. The proofs of these lemmas are provided in Section C.3 to C.7 in sequence. The proof idea mainly follows from (Fu et al. (2020)). Lemma 6 shows the Hessian $\nabla^2 f(\boldsymbol{W})$ of the population risk function is smooth. Lemma 7 illustrates that $\nabla^2 f(\boldsymbol{W})$ is strongly convex in the neighborhood around $\boldsymbol{\mu}^*$. Lemma 8 shows the Hessian of the empirical risk function $\nabla^2 f_n(\boldsymbol{W}^*)$ is close to its population risk $\nabla^2 f(\boldsymbol{W}^*)$ in the local convex region. Summing up these three lemmas, we can derive the proof of Lemma 1. Lemma 4 is used in the proof of Lemma 7. Lemma 5 is used in the proof of Lemma 8.

The analysis of the Hessian matrix of the population loss in (Fu et al., 2020) and (Zhong et al., 2017b) can not be extended to the Gaussian mixture model. To solve this problem, we develop new tools using some good properties of symmetric distribution and even function. Our approach can also be applied to other activations like tanh or erf. Moreover, if we directly apply the existing matrix concentration inequalities in these works in bounding the error between the empirical loss and the population loss, the resulting sample complexity would be $O(d^3)$ and cannot reflect the influence of each component of the Gaussian mixture distribution. We develop a new version of Bernstein's inequality (see (137)) so that the final bound is $O(d \log^2 d)$.

(Mei et al. (2018a)) showed that the landscape of the empirical risk is close to that of the population risk when the number of samples is sufficiently large for the special case that $K = 1$. Focusing on Gaussian mixture models, our result explicitly shows how the parameters of the input distribution, including the proportion, mean and variance of each component will affect the error bound between the empirical loss and the population loss in Lemma 8.

### C.1    USEFUL LEMMAS IN THE PROOF OF LEMMA 1

**Lemma 4**

$$\mathbb{E}_{\boldsymbol{x} \sim \frac{1}{2}\mathcal{N}(\boldsymbol{\mu}, \boldsymbol{I}_d) + \frac{1}{2}\mathcal{N}(-\boldsymbol{\mu}, \boldsymbol{I}_d)} \Big[ (\sum_{i=1}^k \boldsymbol{p}_i^\top \boldsymbol{x} \cdot \phi'(\sigma \cdot x_i))^2 \Big] \geq \rho(\boldsymbol{\mu}, \sigma) \|\boldsymbol{P}\|_F^2 , \qquad (66)$$

*where $\rho(\boldsymbol{\mu}, \sigma)$ is defined in Definition 3.*

**Lemma 5**  *With the FCN model (2) and the Gaussian Mixture Model (7), for some constant $C_{12} > 0$, we have*

$$\mathbb{E}_{\boldsymbol{x} \sim \sum_{l=1}^L \lambda_l \mathcal{N}(\boldsymbol{\mu}_l, \sigma_l^2 \boldsymbol{I}_d)} \Big[ \sup_{\boldsymbol{W} \neq \boldsymbol{W}' \in \mathbb{B}(\boldsymbol{W}^*, r)} \frac{||\nabla^2 \ell(\boldsymbol{W}, \boldsymbol{x}) - \nabla^2 \ell(\boldsymbol{W}', \boldsymbol{x})||}{||\boldsymbol{W} - \boldsymbol{W}'||_F} \Big]$$

$$\leq C_{12} \cdot d^{\frac{3}{2}} K^{\frac{5}{2}} \sqrt{\sum_{l=1}^L \lambda_l (\|\boldsymbol{\mu}\|_\infty + \sigma_l)^2 \sum_{l=1}^L \lambda_l (\|\boldsymbol{\mu}\|_\infty + \sigma_l)^4} \qquad (67)$$

**Lemma 6**  *(Hessian smoothness of population loss)In the FCN model (2), assume $||\boldsymbol{w}_k^*||_2 \leq 1$ for all $k$. Then for some constant $C_5 > 0$, we have*

$$||\nabla^2 f(\boldsymbol{W}) - \nabla^2 f(\boldsymbol{W}^*)|| \leq C_5 \cdot K^{\frac{3}{2}} \cdot \Big( \sum_{l=1}^L \lambda_l (\|\boldsymbol{\mu}\|_\infty + \sigma_l)^4 \sum_{l=1}^L \lambda_l (\|\boldsymbol{\mu}_l\| + \sigma_l)^8 \Big)^{\frac{1}{4}} \cdot ||\boldsymbol{W} - \boldsymbol{W}^*||_F$$

$$(68)$$

**Lemma 7** *(Local strong convexity of population loss) In the FCN model* (2)*, if* $||\boldsymbol{W} - \boldsymbol{W}^*||_F \leq r$ *for an* $\epsilon_0 \in (0, \frac{1}{4})$*, then for some constant* $C_4 > 0$*,*

$$\frac{4(1-\epsilon_0)}{K^2} \sum_{l=1}^{L} \lambda_l \frac{\sigma_l^2}{\eta\kappa^2} \rho(\frac{\boldsymbol{W}^{*\top}\boldsymbol{\mu}_l}{\sigma_l \delta_K(\boldsymbol{W}^*)}, \sigma_l \delta_K(\boldsymbol{W}^*)) \cdot \boldsymbol{I}_{dK} \precsim \nabla^2 f(\boldsymbol{W}) \preceq C_4 \sum_{l=1}^{L} \lambda_l(||\boldsymbol{\mu}_l||_\infty + \sigma_l)^2 \cdot \boldsymbol{I}_{dK} \tag{69}$$

**Lemma 8** *In the FCN model* (2)*, as long as* $n \geq C' \cdot dK \log dK$ *for some constant* $C' > 0$*, we have*

$$\sup_{\boldsymbol{W} \in \mathbb{B}(\boldsymbol{W}^*, r_{FCN})} ||\nabla^2 f_n(\boldsymbol{W}) - \nabla^2 f(\boldsymbol{W})|| \leq C_6 \cdot \sum_{l=1}^{L} \lambda_l(||\boldsymbol{\mu}||_\infty + \sigma_l)^2 \sqrt{\frac{dK \log n}{n}}) \tag{70}$$

*with probability at least* $1 - d^{-10}$ *for some constant* $C_6 > 0$*.*

### C.2 PROOF OF LEMMA 1

From Lemma 7 and 8, with probability at least $1 - d^{-10}$,

$$\begin{aligned} \nabla^2 f_n(\boldsymbol{W}) &\succeq \nabla^2 f(\boldsymbol{W}) - ||\nabla^2 f(\boldsymbol{W}) - \nabla^2 f_n(\boldsymbol{W})|| \cdot \boldsymbol{I} \\ &\succeq \Omega\Big(\frac{1-\epsilon_0}{K^2} \sum_{l=1}^{L} \lambda_l \frac{\sigma_l^2}{\eta\kappa^2} \rho(\frac{\boldsymbol{W}^{*\top}\boldsymbol{\mu}_l}{\sigma_l \delta_K(\boldsymbol{W}^*)}, \sigma_l \delta_K(\boldsymbol{W}^*))\Big) \cdot \boldsymbol{I} \\ &\quad - O\Big(C_6 \cdot \sum_{l=1}^{L} \lambda_l(||\boldsymbol{\mu}_l|| + \sigma_l)^2 \sqrt{\frac{dK \log n}{n}}\Big) \cdot \boldsymbol{I} \end{aligned} \tag{71}$$

As long as the sample complexity is set to satisfy

$$C_6 \cdot \sum_{l=1}^{L} \lambda_l(||\boldsymbol{\mu}_l||_\infty + \sigma_l)^2 \sqrt{\frac{dK \log n}{n}} \leq \frac{\epsilon_0}{K^2} \sum_{l=1}^{L} \lambda_l \frac{\sigma_l^2}{\eta\kappa^2} \rho(\frac{\boldsymbol{W}^{*\top}\boldsymbol{\mu}_l}{\sigma_l \delta_K(\boldsymbol{W}^*)}, \sigma_l \delta_K(\boldsymbol{W}^*)) \tag{72}$$

i.e.,

$$n \geq C_1 \epsilon_0^{-2} \cdot \Big(\sum_{l=1}^{L} \lambda_l(||\boldsymbol{\mu}||_\infty + \sigma_l)^2\Big)^2 \Big(\sum_{l=1}^{L} \lambda_l \frac{\sigma_l^2}{\eta\kappa^2} \rho(\frac{\boldsymbol{W}^{*\top}\boldsymbol{\mu}_l}{\sigma_l \delta_K(\boldsymbol{W}^*)}, \sigma_l \delta_K(\boldsymbol{W}^*))\Big)^{-2} dK^5 \log^2 d \tag{73}$$

for some constant $C_1 > 0$, then we have the lower bound of the Hessian with probability at least $1 - d^{-10}$.

$$\nabla^2 f_n(\boldsymbol{W}) \succeq \Omega\Big(\frac{1-2\epsilon_0}{K^2} \sum_{l=1}^{L} \lambda_l \frac{\sigma_l^2}{\eta\kappa^2} \rho(\frac{\boldsymbol{W}^{*\top}\boldsymbol{\mu}_l}{\sigma_l \delta_K(\boldsymbol{W}^*)}, \sigma_l \delta_K(\boldsymbol{W}^*))\Big) \cdot \boldsymbol{I} \tag{74}$$

By (69) and (70), we can also derive the upper bound as follows,

$$\begin{aligned} ||\nabla^2 f_n(\boldsymbol{W})|| &\leq ||\nabla^2 f(\boldsymbol{W})|| + ||\nabla^2 f_n(\boldsymbol{W}) - \nabla^2 f(\boldsymbol{W})|| \\ &\leq C_4 \cdot \sum_{l=1}^{L} \lambda_l(||\boldsymbol{\mu}_l||_\infty + \sigma_l)^2 + C_6 \cdot \sum_{1=1}^{L} \lambda_l(||\boldsymbol{\mu}||_\infty + \sigma_l)^2 \sqrt{\frac{dK \log n}{n}} \\ &\leq C_2 \cdot \sum_{l=1}^{L} \lambda_l(||\boldsymbol{\mu}_l||_\infty + \sigma_l)^2 \end{aligned} \tag{75}$$

for some constant $C_2 > 0$. Combining (74) and (75), we have

$$\Omega\Big(\frac{1-2\epsilon_0}{K^2} \sum_{l=1}^{L} \lambda_l \frac{\sigma_l^2}{\eta\kappa^2} \rho(\frac{\boldsymbol{W}^{*\top}\boldsymbol{\mu}_l}{\sigma_l \delta_K(\boldsymbol{W}^*)}, \sigma_l \delta_K(\boldsymbol{W}^*)) \cdot \boldsymbol{I} \preceq \nabla^2 f_n(\boldsymbol{W}) \preceq C_2 \sum_{l=1}^{L} \lambda_l(||\boldsymbol{\mu}_l||_\infty + \sigma_l)^2 \cdot \boldsymbol{I} \tag{76}$$

with probability at least $1 - d^{-10}$.

### C.3 PROOF OF LEMMA 4

Following the proof idea in Lemma D.4 of (Zhong et al., 2017b), we have

$$\mathbb{E}_{\boldsymbol{x}\sim\frac{1}{2}\mathcal{N}(\boldsymbol{\mu},\boldsymbol{I}_d)+\frac{1}{2}\mathcal{N}(-\boldsymbol{\mu},\boldsymbol{I}_d)}\left[(\sum_{i=1}^{k}\boldsymbol{p}_i^\top\boldsymbol{x}\cdot\phi'(\sigma\cdot x_i))^2\right] = A_0 + B_0 \tag{77}$$

$$A_0 = \mathbb{E}_{\boldsymbol{x}\sim\frac{1}{2}\mathcal{N}(\boldsymbol{\mu},\boldsymbol{I}_d)+\frac{1}{2}\mathcal{N}(-\boldsymbol{\mu},\boldsymbol{I}_d)}\Big(\sum_{i=1}^{k}\boldsymbol{p}_i^\top\boldsymbol{x}\cdot\phi'^2(\sigma\cdot x_i)\cdot\boldsymbol{x}\boldsymbol{x}^\top\boldsymbol{p}_i\Big) \tag{78}$$

$$B_0 = \mathbb{E}_{\boldsymbol{x}\sim\frac{1}{2}\mathcal{N}(\boldsymbol{\mu},\boldsymbol{I}_d)+\frac{1}{2}\mathcal{N}(-\boldsymbol{\mu},\boldsymbol{I}_d)}\Big(\sum_{i\neq l}\boldsymbol{p}_i^\top\phi'(\sigma\cdot x_i)\phi'(\sigma\cdot x_l)\cdot\boldsymbol{x}\boldsymbol{x}^\top\boldsymbol{p}_l\Big) \tag{79}$$

In $A_0$, we know that $\mathbb{E}_{\boldsymbol{x}\sim\frac{1}{2}\mathcal{N}(\boldsymbol{\mu},\boldsymbol{I}_d)+\frac{1}{2}\mathcal{N}(-\boldsymbol{\mu},\boldsymbol{I}_d)}x_j = 0$. Therefore,

$$
\begin{aligned}
A_0 &= \sum_{i=1}^{k}\mathbb{E}_{\boldsymbol{x}\sim\frac{1}{2}\mathcal{N}(\boldsymbol{\mu},\boldsymbol{I}_d)+\frac{1}{2}\mathcal{N}(-\boldsymbol{\mu},\boldsymbol{I}_d)}\Big[\boldsymbol{p}_i^\top\Big(\phi'^2(\sigma\cdot x_i)\Big(x_i^2\boldsymbol{e}_i\boldsymbol{e}_i^\top + \sum_{j\neq i}x_ix_j(\boldsymbol{e}_i\boldsymbol{e}_j^\top \\
&\quad + \boldsymbol{e}_j\boldsymbol{e}_i^\top) + \sum_{j\neq i}\sum_{l\neq i}x_jx_l\boldsymbol{e}_j\boldsymbol{e}_l^\top\Big)\Big)\boldsymbol{p}_i\Big] \\
&= \sum_{i=1}^{k}\mathbb{E}_{\boldsymbol{x}\sim\frac{1}{2}\mathcal{N}(\boldsymbol{\mu},\boldsymbol{I}_d)+\frac{1}{2}\mathcal{N}(-\boldsymbol{\mu},\boldsymbol{I}_d)}\Big[\boldsymbol{p}_i^\top\Big(\phi'^2(\sigma\cdot x_i)\Big(x_i^2\boldsymbol{e}_i\boldsymbol{e}_i^\top + \sum_{j\neq i}x_j^2\boldsymbol{e}_j\boldsymbol{e}_j^\top\Big)\Big)\boldsymbol{p}_i\Big] \\
&= \sum_{i=1}^{k}\Big[\mathbb{E}_{\boldsymbol{x}\sim\frac{1}{2}\mathcal{N}(\boldsymbol{\mu},\boldsymbol{I}_d)+\frac{1}{2}\mathcal{N}(-\boldsymbol{\mu},\boldsymbol{I}_d)}[\phi'^2(\sigma\cdot x_i)x_i^2]\boldsymbol{p}_i^\top\boldsymbol{e}_i\boldsymbol{e}_i^\top\boldsymbol{p}_i \\
&\quad + \sum_{j\neq i}\mathbb{E}_{\boldsymbol{x}\sim\frac{1}{2}\mathcal{N}(\boldsymbol{\mu},\boldsymbol{I}_d)+\frac{1}{2}\mathcal{N}(-\boldsymbol{\mu},\boldsymbol{I}_d)}[x_j^2]\mathbb{E}_{\boldsymbol{x}\sim\frac{1}{2}\mathcal{N}(\boldsymbol{\mu},\boldsymbol{I})+\frac{1}{2}\mathcal{N}(-\boldsymbol{\mu},\boldsymbol{I})}[\phi'^2(\sigma\cdot x_i)]\boldsymbol{p}_i^\top\boldsymbol{e}_j\boldsymbol{e}_j^\top\boldsymbol{p}_i\Big] \\
&= \sum_{i=1}^{k}p_{ii}^2\beta_2(i,\boldsymbol{\mu},\sigma) + \sum_{i=1}^{k}\sum_{j\neq i}p_{ij}^2\beta_0(i,\boldsymbol{\mu},\sigma)(1+\mu_j^2)
\end{aligned} \tag{80}
$$

In $B_0$, $\alpha_1(i,\boldsymbol{\mu},\sigma) = \mathbb{E}_{\boldsymbol{x}\sim\frac{1}{2}\mathcal{N}(\boldsymbol{\mu},\boldsymbol{I}_d)+\frac{1}{2}\mathcal{N}(-\boldsymbol{\mu},\boldsymbol{I}_d)}(x_i\phi'(x_i)) = 0$. By the equation in Page 30 of (Zhong et al., 2017b), we have

$$
\begin{aligned}
B_0 &= \sum_{i\neq l}^{k}\mathbb{E}_{\boldsymbol{x}\sim\frac{1}{2}\mathcal{N}(\boldsymbol{\mu},\boldsymbol{I}_d)+\frac{1}{2}\mathcal{N}(-\boldsymbol{\mu},\boldsymbol{I}_d)}\Big[\boldsymbol{p}_i^\top\Big(\phi'(\sigma\cdot x_i)\phi'(\sigma\cdot x_l)\Big(x_i^2\boldsymbol{e}_i\boldsymbol{e}_i^\top + x_l^2\boldsymbol{e}_l\boldsymbol{e}_l^\top + x_ix_l(\boldsymbol{e}_i\boldsymbol{e}_l^\top + \\
&\quad \boldsymbol{e}_l\boldsymbol{e}_i^\top) + \sum_{j\neq i}x_jx_l\boldsymbol{e}_j\boldsymbol{e}_l^\top + \sum_{j\neq l}x_jx_i\boldsymbol{e}_j\boldsymbol{e}_i^\top + \sum_{j\neq i,j'\neq i,l}x_jx_{j'}\boldsymbol{e}_j\boldsymbol{e}_{j'}^\top\Big)\Big)\boldsymbol{p}_l\Big] \\
&= \sum_{i\neq l}p_{ii}p_{li}\alpha_2(i,\boldsymbol{\mu},\sigma)\alpha_0(l,\boldsymbol{\mu},\sigma) + \sum_{i\neq l}p_{ij}p_{lj}\alpha_0(i,\boldsymbol{\mu},\sigma)\alpha_0(l,\boldsymbol{\mu},\sigma)(1+\mu_j^2)
\end{aligned} \tag{81}
$$

Therefore,

$$
\begin{aligned}
A_0 + B_0 &= \sum_{i=1}^{k}\Big(p_{ii}\frac{\alpha_2(i,\boldsymbol{\mu},\sigma)}{\sqrt{1+\mu_i^2}} + \sum_{l\neq i}p_{li}\alpha_0(l,\boldsymbol{\mu},\sigma)\sqrt{1+\mu_i^2}\Big)^2 - \sum_{i=1}^{k}p_{ii}^2\frac{\alpha_2^2(i,\boldsymbol{\mu},\sigma)}{1+\mu_i^2} \\
&\quad - \sum_{i=1}^{k}\sum_{l\neq i}p_{li}^2\alpha_0(l,\boldsymbol{\mu},\sigma)^2(1+\mu_i^2) + \sum_{i=1}^{k}p_{ii}^2\beta_2(i,\boldsymbol{\mu},\sigma) + \sum_{i=1}^{k}\sum_{j\neq i}p_{ij}^2\beta_0(i,\boldsymbol{\mu},\sigma)(1+\mu_j^2) \\
&\geq \sum_{i=1}^{k}p_{ii}^2\Big(\beta_2(i,\boldsymbol{\mu},\sigma) - \frac{\alpha_2^2(i,\boldsymbol{\mu},\sigma)}{1+\mu_i^2}\Big) + \sum_{i=1}^{k}\sum_{j\neq i}p_{ij}^2\Big(\beta_0(i,\boldsymbol{\mu},\sigma) - \alpha_0^2(i,\boldsymbol{\mu},\sigma)\Big)(1+\mu_j^2) \\
&\geq \rho(\boldsymbol{\mu},\sigma)||P||_F^2
\end{aligned} \tag{82}
$$

## C.4 Proof of Lemma 5

Following the equation (92) in Lemma 8 of (Fu et al., 2020) and by (45)

$$||\nabla^2 \ell(\boldsymbol{W}) - \nabla^2 \ell(\boldsymbol{W}')|| \le \sum_{j=1}^{K} \sum_{l=1}^{K} |\xi_{j,l}(\boldsymbol{W}) - \xi_{j,l}(\boldsymbol{W}')| \cdot ||\boldsymbol{x}\boldsymbol{x}^\top|| \tag{83}$$

By Lagrange's inequality, we have

$$|\xi_{j,l}(\boldsymbol{W}) - \xi_{j,l}(\boldsymbol{W}')| \le (\max_k |T_{j,k,l}|) \cdot ||\boldsymbol{x}|| \cdot \sqrt{K} ||\boldsymbol{W} - \boldsymbol{W}'||_F \tag{84}$$

From Lemma 6, we know

$$\max_k |T_{j,k,l}| \le C_7 \tag{85}$$

By Property 5, we have

$$\mathbb{E}_{\boldsymbol{x} \sim \sum_{l=1}^{L} \lambda_l \mathcal{N}(\boldsymbol{\mu}_l, \sigma_l^2 \boldsymbol{I}_d)}[||\boldsymbol{x}||^{2t}||] \le d^t (2t-1)!! \sum_{l=1}^{L} \lambda_l (||\boldsymbol{\mu}_l||_\infty + \sigma_l)^{2t} \tag{86}$$

Therefore, for some constant $C_{12} > 0$

$$\begin{aligned}
\mathbb{E}_{\boldsymbol{x} \sim \sum_{l=1}^{L} \lambda_l \mathcal{N}(\boldsymbol{\mu}_l, \sigma_l^2 \boldsymbol{I}_d)} &\Big[ \sup_{\boldsymbol{W} \ne \boldsymbol{W}'} \frac{||\nabla^2 \ell(\boldsymbol{W}) - \nabla^2 \ell(\boldsymbol{W}')||}{||\boldsymbol{W} - \boldsymbol{W}'||_F} \Big] \le K^{\frac{5}{2}} \mathbb{E}[||\boldsymbol{x}||_2^3] \\
\le &K^{\frac{5}{2}} \sqrt{d \sum_{l=1}^{L} \lambda_l (||\boldsymbol{\mu}||_\infty + \sigma_l)^2} \sqrt{3d^2 \sum_{l=1}^{L} \lambda_l (||\boldsymbol{\mu}_l||_\infty + \sigma_l)^4} \\
= &C_{12} \cdot d^{\frac{3}{2}} K^{\frac{5}{2}} \sqrt{\sum_{l=1}^{L} \lambda_l (||\boldsymbol{\mu}_l||_\infty + \sigma_l)^2 \sum_{l=1}^{L} \lambda_l (||\boldsymbol{\mu}_l||_\infty + \sigma_l)^4}
\end{aligned} \tag{87}$$

## C.5 Proof of Lemma 6

Let $\boldsymbol{a} = (\boldsymbol{a}_1^\top, \cdots, \boldsymbol{a}_K^\top)^\top \in \mathbb{R}^{dK}$. Let $\Delta_{j,l} \in \mathbb{R}^{d \times d}$ be the $(j,l)$-th block of $\nabla^2 f(\boldsymbol{W}) - \nabla^2 f(\boldsymbol{W}^*) \in \mathbb{R}^{dK \times dK}$. By definition,

$$||\nabla^2 f(\boldsymbol{W}) - \nabla^2 f(\boldsymbol{W}^*)|| = \max_{||\boldsymbol{a}||=1} \sum_{j=1}^{K} \sum_{l=1}^{K} \boldsymbol{a}_j^\top \Delta_{j,l} \boldsymbol{a}_l \tag{88}$$

By the mean value theorem and (45),

$$\begin{aligned}
\Delta_{j,l} = \frac{\partial^2 f(\boldsymbol{W})}{\partial \boldsymbol{w}_j \partial \boldsymbol{w}_l} - \frac{\partial^2 f(\boldsymbol{W}^*)}{\partial \boldsymbol{w}_j^* \partial \boldsymbol{w}_l^*} &= \mathbb{E}_{\boldsymbol{x} \sim \sum_{l=1}^{L} \lambda_l \mathcal{N}(\boldsymbol{\mu}_l, \sigma_l^2 \boldsymbol{I}_d)}[(\xi_{j,l}(\boldsymbol{W}) - \xi_{j,l}(\boldsymbol{W}^*)) \cdot \boldsymbol{x}\boldsymbol{x}^\top] \\
&= \mathbb{E}_{\boldsymbol{x} \sim \sum_{l=1}^{L} \lambda_l \mathcal{N}(\boldsymbol{\mu}_l, \sigma_l^2 \boldsymbol{I}_d)}\Big[ \sum_{k=1}^{K} \Big\langle \frac{\partial \xi_{j,l}(\boldsymbol{W}')}{\partial \boldsymbol{w}_k'}, \boldsymbol{w}_k - \boldsymbol{w}_k^* \Big\rangle \cdot \boldsymbol{x}\boldsymbol{x}^\top \Big] \\
&= \mathbb{E}_{\boldsymbol{x} \sim \sum_{l=1}^{L} \lambda_l \mathcal{N}(\boldsymbol{\mu}_l, \sigma_l^2 \boldsymbol{I}_d)}\Big[ \sum_{k=1}^{K} \langle T_{j,l,k} \cdot \boldsymbol{x}, \boldsymbol{w}_k - \boldsymbol{w}_k^* \rangle \cdot \boldsymbol{x}\boldsymbol{x}^\top \Big]
\end{aligned} \tag{89}$$

where $\boldsymbol{W}' = \gamma \boldsymbol{W} + (1 - \gamma)\boldsymbol{W}^*$ for some $\gamma \in (0,1)$ and $T_{j,l,k}$ is defined such that $\frac{\partial \xi_{j,l}(\boldsymbol{W}')}{\partial \boldsymbol{w}_k'} = T_{j,l,k} \cdot x \in \mathbb{R}^d$. Then we provide an upper bound for $\xi_{j,l}$. Since that $y = 1$ or $0$, we first compute the case in which $y = 1$. From (45) we can obtain

$$\xi_{j,l}(\boldsymbol{W}) = \begin{cases} \frac{1}{K^2} \phi'(\boldsymbol{w}_j^\top \boldsymbol{x})\phi'(\boldsymbol{w}_l^\top \boldsymbol{x}) \cdot \frac{1}{H^2(\boldsymbol{W})}, & j \ne l \\ \frac{1}{K^2} \phi'(\boldsymbol{w}_j^\top \boldsymbol{x})\phi'(\boldsymbol{w}_l^\top \boldsymbol{x}) \cdot \frac{1}{H^2(\boldsymbol{W})} - \frac{1}{K} \phi''(\boldsymbol{w}_j^\top \boldsymbol{x}) \cdot \frac{1}{H(\boldsymbol{W})}, & j = l \end{cases} \tag{90}$$

We can bound $\xi_{j,l}(\boldsymbol{W})$ by bounding each components of (90). Note that we have

$$\frac{1}{K^2}\phi'(\boldsymbol{w}_j^\top \boldsymbol{x})\phi'(\boldsymbol{w}_l^\top \boldsymbol{x}) \cdot \frac{1}{H^2(\boldsymbol{W})} \leq \frac{1}{K^2}\frac{\phi(\boldsymbol{w}_j^\top \boldsymbol{x})\phi(\boldsymbol{w}_l^\top \boldsymbol{x})(1-\phi(\boldsymbol{w}_j^\top \boldsymbol{x}))(1-\phi(\boldsymbol{w}_l^\top \boldsymbol{x}))}{\frac{1}{K^2}\phi(\boldsymbol{w}_j^\top \boldsymbol{x})\phi(\boldsymbol{w}_l^\top \boldsymbol{x})} \leq 1 \tag{91}$$

$$\frac{1}{K}\phi''(\boldsymbol{w}_j^\top \boldsymbol{x}) \cdot \frac{1}{H(\boldsymbol{W})} \leq \frac{1}{K}\frac{\phi(\boldsymbol{w}_j^\top \boldsymbol{x})(1-\phi(\boldsymbol{w}_j^\top \boldsymbol{x}))(1-2\phi(\boldsymbol{w}_j^\top \boldsymbol{x}))}{\frac{1}{K}\phi(\boldsymbol{w}_j^\top \boldsymbol{x})} \leq 1 \tag{92}$$

where (91) holds for any $j, l \in [K]$. The case $y = 0$ can be computed with the same upper bound by substituting $(1 - H(\boldsymbol{W})) = \frac{1}{K}\sum_{j=1}^K(1-\phi(\boldsymbol{w}_j^\top \boldsymbol{x}))$ for $H(\boldsymbol{W})$ in (90), (91) and (92). Therefore, there exists a constant $C_9 > 0$, such that

$$|\xi_{j,l}(\boldsymbol{W})| \leq C_9 \tag{93}$$

We then need to calculate $T_{j,l,k}$. Following the analysis of $\xi_{j,l}(\boldsymbol{W})$, we only consider the case of $y = 1$ here for simplicity.

$$T_{j,l,k} = \frac{-2}{K^3 H^3(\boldsymbol{W}')}\phi'({\boldsymbol{w}'}_j^\top \boldsymbol{x})\phi'({\boldsymbol{w}'}_l^\top \boldsymbol{x})\phi'({\boldsymbol{w}'}_k^\top \boldsymbol{x}), \quad \text{where } j, l, k \text{ are not equal to each other} \tag{94}$$

$$T_{j,j,k} = \begin{cases} \frac{-2}{K^3 H^3(\boldsymbol{W}')}\phi'({\boldsymbol{w}'}_j^\top \boldsymbol{x})\phi'({\boldsymbol{w}'}_j^\top \boldsymbol{x})\phi'({\boldsymbol{w}'}_k^\top \boldsymbol{x}) + \frac{1}{K^2 H^2(\boldsymbol{W}')}\phi''({\boldsymbol{w}'}_j^\top \boldsymbol{x})\phi'({\boldsymbol{w}'}_k^\top \boldsymbol{x}), & j \neq k \\ \frac{-2}{K^3 H^3(\boldsymbol{W}')}(\phi'({\boldsymbol{w}'}_j^\top \boldsymbol{x}))^3 + \frac{3}{K^2 H^2(\boldsymbol{W}')}\phi''({\boldsymbol{w}'}_j^\top \boldsymbol{x})\phi'({\boldsymbol{w}'}_j^\top \boldsymbol{x}) - \frac{\phi'''({\boldsymbol{w}'}_j^\top \boldsymbol{x})}{KH(\boldsymbol{W}')}, & j = k \end{cases} \tag{95}$$

$$\boldsymbol{a}_j^\top \Delta_{j,l} \boldsymbol{a}_l = \mathbb{E}_{\boldsymbol{x} \sim \sum_{l=1}^L \mathcal{N}(\boldsymbol{\mu}_l, \sigma_l^2 \boldsymbol{I})}[(\sum_{k=1}^K T_{j,l,k}\langle \boldsymbol{x}, \boldsymbol{w}_k - \boldsymbol{w}_k^*\rangle) \cdot (\boldsymbol{a}_j^\top \boldsymbol{x})(\boldsymbol{a}_l^\top \boldsymbol{x})]$$

$$\leq \sqrt{\mathbb{E}_{\boldsymbol{x} \sim \sum_{l=1}^L \mathcal{N}(\boldsymbol{\mu}_l, \sigma_l^2 \boldsymbol{I})}[\sum_{k=1}^K T_{j,k,l}^2] \cdot \mathbb{E}[\sum_{k=1}^K (\langle \boldsymbol{x}, \boldsymbol{w}_k - \boldsymbol{w}_k^*\rangle(\boldsymbol{a}_j^\top \boldsymbol{x})(\boldsymbol{a}_l^\top \boldsymbol{x}))^2]}$$

$$\leq \sqrt{\mathbb{E}_{\boldsymbol{x} \sim \sum_{l=1}^L \mathcal{N}(\boldsymbol{\mu}_l, \sigma_l^2 \boldsymbol{I})}[\sum_{k=1}^K T_{j,k,l}^2]}\sqrt{\sum_{k=1}^K \sqrt{\mathbb{E}((\boldsymbol{w}_k - \boldsymbol{w}_k^*)^\top \boldsymbol{x})^4} \cdot \sqrt{\mathbb{E}[(\boldsymbol{a}_j^\top \boldsymbol{x})^4(\boldsymbol{a}_l^\top \boldsymbol{x})^4]}}$$

$$\leq C_8 \sqrt{\mathbb{E}_{\boldsymbol{x} \sim \sum_{l=1}^L \mathcal{N}(\boldsymbol{\mu}_l, \sigma_l^2 \boldsymbol{I})}[\sum_{k=1}^K T_{j,k,l}^2]}\sqrt{\sum_{k=1}^K ||\boldsymbol{w}_k - \boldsymbol{w}_k^*||_2^2 \cdot ||\boldsymbol{a}_j||_2^2 \cdot ||\boldsymbol{a}_l||_2^2}$$

$$\cdot \left(\sum_{l=1}^L \lambda_l(||\boldsymbol{\mu}_l||_\infty + \sigma_l)^4 \sum_{l=1}^L \lambda_l(||\boldsymbol{\mu}_l||_\infty + \sigma_l)^8\right)^{\frac{1}{4}} \tag{96}$$

for some constant $C_8 > 0$. All the three inequalities of (96) are derived from Cauchy-Schwarz inequality. Note that we have

$$\left|\frac{-2}{K^3 H^3(\boldsymbol{W})}(\phi'(\boldsymbol{w}_j^\top \boldsymbol{x}))^2 \phi'(\boldsymbol{w}_k^\top \boldsymbol{x})\right| \leq \frac{2\phi^2(\boldsymbol{w}_j^\top \boldsymbol{x})(1-\phi(\boldsymbol{w}_j^\top \boldsymbol{x}))^2 \phi(\boldsymbol{w}_k^\top \boldsymbol{x})(1-\phi(\boldsymbol{w}_k^\top \boldsymbol{x}))}{K^3 \frac{1}{K^3}\phi^2(\boldsymbol{w}_j^\top \boldsymbol{x})\phi(\boldsymbol{w}_k^\top \boldsymbol{x})} \tag{97}$$

$$= 2(1 - \phi(\boldsymbol{w}_j^\top \boldsymbol{x}))^2(1 - \phi(\boldsymbol{w}_k^\top \boldsymbol{x})) \leq 2$$

$$\left|\frac{-2}{K^3 H^3(\boldsymbol{W})}\phi'(\boldsymbol{w}_j^\top \boldsymbol{x})\phi'(\boldsymbol{w}_l^\top \boldsymbol{x})\phi'(\boldsymbol{w}_k^\top \boldsymbol{x})\right| \leq 2 \tag{98}$$

$$\left|\frac{3}{K^2 H^2(\boldsymbol{W})}\phi''(\boldsymbol{w}_j^\top \boldsymbol{x})\phi'(\boldsymbol{w}_k^\top \boldsymbol{x})\right|$$

$$\leq \left|\frac{3\phi(\boldsymbol{w}_j^\top \boldsymbol{x})(1-\phi(\boldsymbol{w}_j^\top \boldsymbol{x}))(1-2\phi(\boldsymbol{w}_j^\top \boldsymbol{x}))\phi(\boldsymbol{w}_k^\top \boldsymbol{x})(1-\phi(\boldsymbol{w}_k^\top \boldsymbol{x}))}{K^2 \frac{1}{K^2}\phi(\boldsymbol{w}_j^\top \boldsymbol{x})\phi(\boldsymbol{w}_k^\top \boldsymbol{x})}\right| \tag{99}$$

$$= \left|3(1 - \phi(\boldsymbol{w}_j^\top \boldsymbol{x}))(1 - 2\phi(\boldsymbol{w}_j^\top \boldsymbol{x}))(1 - \phi(\boldsymbol{w}_k^\top \boldsymbol{x}))\right| \leq 3$$

$$\Big|\frac{\phi'''(\boldsymbol{w}_j^\top \boldsymbol{x})}{KH(\boldsymbol{W})}\Big| \leq \Big|\frac{\phi(\boldsymbol{w}_j^\top \boldsymbol{x})(1-\phi(\boldsymbol{w}_j^\top \boldsymbol{x}))(1-6\phi(\boldsymbol{w}_j^\top \boldsymbol{x})+6\phi^2(\boldsymbol{w}_j^\top \boldsymbol{x}))}{K\frac{1}{K}\phi(\boldsymbol{w}_j^\top \boldsymbol{x})}\Big| \leq 1 \tag{100}$$

Therefore, by combining (94), (95) and (97) to (100), we have

$$|T_{j,l,k}| \leq C_7 \quad \Rightarrow \quad T_{j,l,k}^2 \leq C_7^2, \forall j,l,k \in [K], \tag{101}$$

for some constants $C_7 > 0$. By (88), (89), (96), (101) and the Cauchy-Schwarz's Inequality, we have

$$\|\nabla^2 f(\boldsymbol{W}) - \nabla^2 f(\boldsymbol{W}^*)\|$$

$$\leq C_8 \sqrt{C_7^2 K} \|\boldsymbol{W}-\boldsymbol{W}^*\|_F \Big(\sum_{l=1}^L \lambda_l(\|\boldsymbol{\mu}_l\|_\infty + \sigma_l)^4 \sum_{l=1}^L \lambda_l(\|\boldsymbol{\mu}_l\|_\infty + \sigma_l)^8\Big)^{\frac{1}{4}}$$

$$\cdot \max_{\|\boldsymbol{a}\|=1} \sum_{j=1}^K \sum_{l=1}^K \|\boldsymbol{a}_j\|_2 \|\boldsymbol{a}_l\|_2$$

$$\leq C_8 \sqrt{C_7^2 K} \cdot \|\boldsymbol{W}-\boldsymbol{W}^*\|_F \cdot \Big(\sum_{l=1}^L \lambda_l(\|\boldsymbol{\mu}_l\|_\infty + \sigma_l)^4 \sum_{l=1}^L \lambda_l(\|\boldsymbol{\mu}_l\|_\infty + \sigma_l)^8\Big)^{\frac{1}{4}} \cdot \Big(\sum_{j=1}^K \|\boldsymbol{a}_j\|\Big)^2$$

$$\leq C_8 \sqrt{C_7^2 K^3} \cdot \|\boldsymbol{W}-\boldsymbol{W}^*\|_F \cdot \Big(\sum_{l=1}^L \lambda_l(\|\boldsymbol{\mu}_l\|_\infty + \sigma_l)^4 \sum_{l=1}^L \lambda_l(\|\boldsymbol{\mu}_l\|_\infty + \sigma_l)^8\Big)^{\frac{1}{4}} \tag{102}$$

Hence, we have

$$\|\nabla^2 f(\boldsymbol{W}) - \nabla^2 f(\boldsymbol{W}^*)\| \leq C_5 K^{\frac{3}{2}} \Big(\sum_{l=1}^L \lambda_l(\|\boldsymbol{\mu}_l\|_\infty + \sigma_l)^4 \sum_{l=1}^L \lambda_l(\|\boldsymbol{\mu}_l\|_\infty + \sigma_l)^8\Big)^{\frac{1}{4}} \|\boldsymbol{W}-\boldsymbol{W}^*\|_F \tag{103}$$

for some constant $C_5 > 0$.

## C.6 Proof of Lemma 7

From (Fu et al. (2020)), we know

$$\nabla^2 f(\boldsymbol{W}^*) \succeq \min_{\|\boldsymbol{a}\|=1} \frac{4}{K^2} \mathbb{E}_{\boldsymbol{x}\sim\sum_{l=1}^L \lambda_l \mathcal{N}(\boldsymbol{\mu}_l,\sigma_l^2 \boldsymbol{I}_d)} \Big[\Big(\sum_{j=1}^K \phi'(\boldsymbol{w}_j^{*\top}\boldsymbol{x})(\boldsymbol{a}_j^\top \boldsymbol{x})\Big)^2\Big] \cdot \boldsymbol{I}_{dK} \tag{104}$$

with $\boldsymbol{a} = (\boldsymbol{a}_1^\top, \cdots, \boldsymbol{a}_K^\top)^\top \in \mathbb{R}^{dK}$. And

$$\nabla^2 f(\boldsymbol{W}^*) \preceq \Big(\max_{\|\boldsymbol{a}\|=1} \boldsymbol{a}^\top \nabla^2 f(\boldsymbol{W}^*)\boldsymbol{a}\Big) \cdot \boldsymbol{I}_{dK} \preceq C_4 \cdot \max_{\|\boldsymbol{a}\|=1} \mathbb{E}_{\boldsymbol{x}\sim\sum_{l=1}^L \lambda_l \mathcal{N}(\boldsymbol{\mu}_l,\sigma_l^2 \boldsymbol{I}_d)} \Big[\sum_{j=1}^K (\boldsymbol{a}_j^\top \boldsymbol{x})^2\Big] \cdot \boldsymbol{I}_{dK} \tag{105}$$

for some constant $C_4 > 0$. By applying Property 4, we can derive the upper bound in (105) as

$$C_4 \cdot \mathbb{E}_{\boldsymbol{x}\sim\sum_{l=1}^L \lambda_l \mathcal{N}(\boldsymbol{\mu}_l,\sigma_l^2 \boldsymbol{I}_d)} \Big[\sum_{j=1}^K (\boldsymbol{a}_j^\top \boldsymbol{x})^2\Big] \cdot \boldsymbol{I}_{dK} \preceq C_4 \cdot \sum_{l=1}^L \lambda_l(\|\boldsymbol{\mu}_l\|_\infty + \sigma_l)^2 \cdot \boldsymbol{I}_{dK} \tag{106}$$

To find a lower bound for (104), we can first transfer the expectation of the Gaussian Mixture Model to the weight sum of the expectations over general Gaussian distributions.

$$\min_{\|\boldsymbol{a}\|=1} \mathbb{E}_{\boldsymbol{x}\sim\sum_{l=1}^L \lambda_l \mathcal{N}(\boldsymbol{\mu}_l,\sigma_l^2 \boldsymbol{I}_d)} \Big[\Big(\sum_{j=1}^K \phi'(\boldsymbol{w}_j^{*\top}\boldsymbol{x})(\boldsymbol{a}_j^\top \boldsymbol{x})\Big)^2\Big]$$

$$= \min_{\|\boldsymbol{a}\|=1} \sum_{l=1}^L \lambda_l \mathbb{E}_{\boldsymbol{x}\sim\mathcal{N}(\boldsymbol{\mu}_l,\sigma_l^2 \boldsymbol{I}_d)} \Big[\Big(\sum_{j=1}^K \phi'(\boldsymbol{w}_j^{*\top}\boldsymbol{x})(\boldsymbol{a}_j^\top \boldsymbol{x})\Big)^2\Big] \tag{107}$$

Denote $\boldsymbol{U} \in \mathbb{R}^{d \times k}$ as the orthogonal basis of $\boldsymbol{W}^*$. For any vector $\boldsymbol{a}_i \in \mathbb{R}^d$, there exists two vectors $\boldsymbol{b}_i \in \mathbb{R}^K$ and $\boldsymbol{c}_i \in \mathbb{R}^{d-K}$ such that

$$\boldsymbol{a}_i = \boldsymbol{U}\boldsymbol{b}_i + \boldsymbol{U}_\perp \boldsymbol{c}_i \tag{108}$$

where $\boldsymbol{U}_\perp \in \mathbb{R}^{d \times (d-K)}$ denotes the complement of $\boldsymbol{U}$. We also have $\boldsymbol{U}_\perp^\top \boldsymbol{\mu}_l = 0$ by (9). Plugging (108) into RHS of (107), and then we have

$$
\begin{aligned}
&\mathbb{E}_{\boldsymbol{x} \sim \mathcal{N}(\boldsymbol{\mu}_l, \sigma_l^2 \boldsymbol{I}_d)} \Big[ \Big( \sum_{i=1}^K \boldsymbol{a}_i^\top \boldsymbol{x} \cdot \phi'(\boldsymbol{w}_i^{*\top} \boldsymbol{x}) \Big)^2 \Big] \\
&= \mathbb{E}_{\boldsymbol{x} \sim \mathcal{N}(\boldsymbol{\mu}_l, \sigma_l^2 \boldsymbol{I}_d)} \Big[ \Big( \sum_{i=1}^K (\boldsymbol{U}\boldsymbol{b}_i + \boldsymbol{U}_\perp \boldsymbol{c}_i)^\top \boldsymbol{x} \cdot \phi'(\boldsymbol{w}_i^{*\top} \boldsymbol{x}) \Big)^2 \Big] = A + B + C
\end{aligned}
\tag{109}
$$

$$A = \mathbb{E}_{\boldsymbol{x} \sim \mathcal{N}(\boldsymbol{\mu}_l, \sigma_l^2 \boldsymbol{I}_d)} \Big[ \Big( \sum_{i=1}^K \boldsymbol{b}_i^\top \boldsymbol{U}^\top \boldsymbol{x} \cdot \phi'(\boldsymbol{w}_i^{*\top} \boldsymbol{x}) \Big)^2 \Big] \tag{110}$$

$$
\begin{aligned}
C &= \mathbb{E}_{\boldsymbol{x} \sim \mathcal{N}(\boldsymbol{\mu}_l, \sigma_l^2 \boldsymbol{I}_d)} \Big[ 2 \Big( \sum_{i=1}^K \boldsymbol{c}_i^\top \boldsymbol{U}_\perp^\top \boldsymbol{x} \cdot \phi'(\boldsymbol{w}_i^{*\top} \boldsymbol{x}) \Big) \cdot \Big( \sum_{i=1}^K \boldsymbol{b}_i^\top \boldsymbol{U}^\top \boldsymbol{x} \cdot \phi'(\boldsymbol{w}_i^{*\top} \boldsymbol{x}) \Big) \Big] \\
&= \sum_{i=1}^K \sum_{j=1}^K \mathbb{E}_{\boldsymbol{x} \sim \mathcal{N}(\boldsymbol{\mu}_l, \sigma_l^2 \boldsymbol{I}_d)} \Big[ 2\boldsymbol{c}_i^\top \boldsymbol{U}_\perp^\top \boldsymbol{x} \Big] \mathbb{E}_{\boldsymbol{x} \sim \mathcal{N}(\boldsymbol{\mu}_l, \sigma_l^2 \boldsymbol{I}_d)} \Big[ \boldsymbol{b}_i^\top \boldsymbol{U}^\top \boldsymbol{x} \cdot \phi'(\boldsymbol{w}_i^{*\top} \boldsymbol{x})\phi'(\boldsymbol{w}_j^{*\top} \boldsymbol{x}) \Big] \quad (111) \\
&= \sum_{i=1}^K \sum_{j=1}^K \Big[ 2\boldsymbol{c}_i^\top \boldsymbol{U}_\perp^\top \boldsymbol{\mu}_l \Big] \mathbb{E}_{\boldsymbol{x} \sim \mathcal{N}(\boldsymbol{\mu}_l, \sigma_l^2 \boldsymbol{I}_d)} \Big[ \boldsymbol{b}_i^\top \boldsymbol{U}^\top \boldsymbol{x} \cdot \phi'(\boldsymbol{w}_i^{*\top} \boldsymbol{x})\phi'(\boldsymbol{w}_j^{*\top} \boldsymbol{x}) \Big] = 0
\end{aligned}
$$

where the last step is by $\boldsymbol{U}_\perp^\top \boldsymbol{\mu}_l = 0$ by (9).

$$
\begin{aligned}
B &= \mathbb{E}_{\boldsymbol{x} \sim \mathcal{N}(\boldsymbol{\mu}_l, \sigma_l^2 \boldsymbol{I}_d)} \Big[ \Big( \sum_{i=1}^K \boldsymbol{c}_i^\top \boldsymbol{U}_\perp^\top \boldsymbol{x} \cdot \phi'(\boldsymbol{w}_i^{*\top} \boldsymbol{x}))^2 \Big] \\
&= \mathbb{E}_{\boldsymbol{x} \sim \mathcal{N}(\boldsymbol{\mu}_l, \sigma_l^2 \boldsymbol{I}_d)} [(\boldsymbol{t}^\top \boldsymbol{s})^2] \qquad \text{by defining } \boldsymbol{t} = \sum_{i=1}^k \phi'(\boldsymbol{w}_i^{*\top} \boldsymbol{x}) \boldsymbol{c}_i \in \mathbb{R}^{d-K} \text{ and } \boldsymbol{s} = \boldsymbol{U}_\perp^\top \boldsymbol{x} \\
&= \sum_{i=1}^K \mathbb{E}[t_i^2 s_i^2] + \sum_{i \neq j} \mathbb{E}[t_i t_j s_i s_j] \\
&= \sum_{i=1}^K \mathbb{E}[t_i^2]\sigma_l^2 + \Big( \sum_{i=1}^K \mathbb{E}[t_i^2](\boldsymbol{U}_\perp^\top \boldsymbol{\mu}_l)_i^2 + \sum_{i \neq j} \mathbb{E}[t_i t_j](\boldsymbol{U}_\perp^\top \boldsymbol{\mu}_l)_i \cdot (\boldsymbol{U}_\perp^\top \boldsymbol{\mu}_l)_j \Big) \\
&= \mathbb{E}[\sum_{i=1}^{d-K} t_i^2 \sigma_l^2] + \mathbb{E}[(\boldsymbol{t}^\top \boldsymbol{U}_\perp^\top \boldsymbol{\mu}_l)^2] = \mathbb{E}[\sum_{i=1}^{d-K} t_i^2 \sigma_l^2]
\end{aligned}
\tag{112}
$$

The last step is by $\boldsymbol{U}_\perp^\top \boldsymbol{\mu}_l = 0$. The 4th step is because that $s_i$ is independent of $t_i$, thus $\mathbb{E}[t_i t_j s_i s_j] = \mathbb{E}[t_i t_j]\mathbb{E}[s_i s_j]$

$$\mathbb{E}[s_i s_j] = \begin{cases} (\boldsymbol{U}_\perp^\top \boldsymbol{\mu}_l)_i \cdot (\boldsymbol{U}_\perp^\top \boldsymbol{\mu}_l)_j, & \text{if } i \neq j \\ (\boldsymbol{U}_\perp^\top \boldsymbol{\mu}_l)_i^2 + \sigma_l^2, & \text{if } i = j \end{cases} \tag{113}$$

Since $\Big( \sum_{i=1}^k \boldsymbol{p}_i^\top \boldsymbol{x} \cdot \phi'(\sigma \cdot x_i) \Big)^2$ is an even function, so from Property 3 we have

$$\mathbb{E}_{\boldsymbol{x} \sim \mathcal{N}(\boldsymbol{\mu}_l, \sigma_l^2 \boldsymbol{I}_d)} \Big[ (\sum_{i=1}^k \boldsymbol{p}_i^\top \boldsymbol{x} \cdot \phi'(\sigma \cdot x_i))^2 \Big] = \mathbb{E}_{\boldsymbol{x} \sim \frac{1}{2}\mathcal{N}(\boldsymbol{\mu}_l, \sigma_l^2 \boldsymbol{I}_d) + \frac{1}{2}\mathcal{N}(-\boldsymbol{\mu}_l, \sigma_l^2 \boldsymbol{I}_d)} \Big[ (\sum_{i=1}^k \boldsymbol{p}_i^\top \boldsymbol{x} \cdot \phi'(\sigma \cdot x_i))^2 \Big] \tag{114}$$

Combining Lemma 4 and Property 3, we next follow the derivation for the standard Gaussian distribution in Page 36 of (Zhong et al., 2017b) and generalize the result to a Gaussian distribution

with an arbitrary mean and variance as follows.

$$
\begin{aligned}
A &= \mathbb{E}_{\boldsymbol{x} \sim \mathcal{N}(\boldsymbol{\mu}_l, \sigma_l^2 \boldsymbol{I}_d)} \Big[ \Big( \sum_{i=1}^{K} \boldsymbol{b}_i^{\top} \boldsymbol{U}^{\top} \boldsymbol{x} \cdot \phi'(\boldsymbol{w}_i^{*\top} \boldsymbol{x}) \Big)^2 \Big] \\
&= \int (2\pi\sigma_l^2)^{-\frac{K}{2}} \Big[ \Big( \sum_{i=1}^{K} \boldsymbol{b}_i^{\top} \boldsymbol{z} \cdot \phi'(\boldsymbol{v}_i^{\top} \boldsymbol{z}) \Big)^2 \Big] \exp\Big( -\frac{1}{2\sigma_l^2} \|\boldsymbol{z} - \boldsymbol{U}^{\top} \boldsymbol{\mu}_l\|^2 \Big) d\boldsymbol{z} \\
&= \int (2\pi\sigma_l^2)^{-\frac{K}{2}} \Big[ \Big( \sum_{i=1}^{K} \boldsymbol{b}_i^{\top} \boldsymbol{V}^{\dagger\top} \boldsymbol{s} \cdot \phi'(s_i) \Big)^2 \Big] \exp\Big( -\frac{1}{2\sigma_l^2} \|\boldsymbol{V}^{\dagger\top} \boldsymbol{s} - \boldsymbol{U}^{\top} \boldsymbol{\mu}_l\|^2 \Big) \Big| \det(\boldsymbol{V}^{\dagger}) \Big| d\boldsymbol{s} \\
&\geq \int (2\pi\sigma_l^2)^{-\frac{K}{2}} \Big[ \Big( \sum_{i=1}^{k} \boldsymbol{b}_i^{\top} \boldsymbol{V}^{\dagger\top} \boldsymbol{s} \cdot \phi'(s_i) \Big)^2 \Big] \exp\Big( -\frac{\|\boldsymbol{s} - \boldsymbol{V}^{\top} \boldsymbol{U}^{\top} \boldsymbol{\mu}_l\|^2}{2\delta_K^2(\boldsymbol{W}^*)\sigma_l^2} \Big) \Big| \det(\boldsymbol{V}^{\dagger}) \Big| d\boldsymbol{s} \\
&\geq \int (2\pi)^{-\frac{K}{2}} \sigma_l^{-K} \Big[ \Big( \sum_{i=1}^{k} \boldsymbol{b}_i^{\top} \boldsymbol{V}^{\dagger\top} (\delta_K(\boldsymbol{W}^*)\sigma_l) \boldsymbol{g} \cdot \phi'(\delta_K(\boldsymbol{W}^*)\sigma_l \cdot g_i) \Big)^2 \Big] \\
&\quad \cdot \exp\Big( -\frac{\|\boldsymbol{g} - \frac{\boldsymbol{W}^{*\top}\boldsymbol{\mu}_l}{\sigma_l \delta_K(\boldsymbol{W}^*)}\|^2}{2} \Big) \Big| \det(\boldsymbol{V}^{\dagger}) \Big| \sigma_l^K \delta_K^K(\boldsymbol{W}^*) d\boldsymbol{g} \\
&= \frac{\sigma_l^2}{\eta} \mathbb{E}_{\boldsymbol{g}} \Big[ \Big( \sum_{i=1}^{K} (\boldsymbol{b}_i^{\top} \boldsymbol{V}^{\dagger\top} \delta_K(\boldsymbol{W}^*)) \boldsymbol{g} \cdot \phi'(\sigma_l \delta_K(\boldsymbol{W}^*) \cdot g_i) \Big)^2 \Big] \\
&\geq \frac{\sigma_l^2}{\kappa^2 \eta} \rho\Big( \frac{\boldsymbol{W}^{*\top}\boldsymbol{\mu}_l}{\sigma_l \delta_K(\boldsymbol{W}^*)}, \sigma_l \delta_K(\boldsymbol{W}^*) \Big) \|\boldsymbol{b}\|^2.
\end{aligned}
\tag{115}
$$

The second step is by letting $\boldsymbol{z} = \boldsymbol{U}^{\top}\boldsymbol{x}$. The third step is by letting $\boldsymbol{s} = \boldsymbol{V}^{\top}\boldsymbol{z}$. The last to second step follows from $\boldsymbol{g} = \frac{\boldsymbol{s}}{\sigma_l \delta_K(\boldsymbol{W}^*)}$, where $\boldsymbol{g} \sim \mathcal{N}(\frac{\boldsymbol{W}^{*\top}\boldsymbol{\mu}_l}{\sigma_l \delta_K(\boldsymbol{W}^*)}, \boldsymbol{I}_K)$ and the last inequality is by Lemma 4. Similarly, we extend the derivation in Page 37 of (Zhong et al., 2017b) for the standard Gaussian distribution to a general Gaussian distribution as follows.

$$
B = \sigma_l^2 \mathbb{E}_{\boldsymbol{x} \sim \mathcal{N}(\boldsymbol{\mu}_l, \sigma_l^2 \boldsymbol{I}_d)}[\|\boldsymbol{t}\|^2] \geq \frac{\sigma_l^2}{\eta \kappa^2} \rho\Big( \frac{\boldsymbol{W}^{*\top}\boldsymbol{\mu}_l}{\sigma_l \delta_K(\boldsymbol{W}^*)}, \sigma_l \delta_K(\boldsymbol{W}^*) \Big) \|\boldsymbol{c}\|^2
\tag{116}
$$

Combining (109) - (112), (115) and (116), we have

$$
\min_{\|\boldsymbol{a}\|=1} \mathbb{E}_{\boldsymbol{x} \sim \mathcal{N}(\boldsymbol{\mu}_l, \sigma_l^2 \boldsymbol{I}_d)} \Big[ \Big( \sum_{i=1}^{k} \boldsymbol{a}_i^{\top} \boldsymbol{x} \cdot \phi'(\boldsymbol{w}_i^{*\top} \boldsymbol{x}) \Big)^2 \Big] \geq \frac{\sigma_l^2}{\eta \kappa^2} \rho\Big( \frac{\boldsymbol{W}^{*\top}\boldsymbol{\mu}_l}{\sigma_l \delta_K(\boldsymbol{W}^*)}, \sigma_l \delta_K(\boldsymbol{W}^*) \Big).
\tag{117}
$$

For the Gaussian Mixture Model $\boldsymbol{x} \sim \sum_{l=1}^{L} \mathcal{N}(\boldsymbol{\mu}_l, \sigma_l^2 \boldsymbol{I}_d)$, we have

$$
\min_{\|\boldsymbol{a}\|=1} \mathbb{E}_{\boldsymbol{x} \sim \sum_{l=1}^{L} \lambda_l \mathcal{N}(\boldsymbol{\mu}_l, \sigma_l^2 \boldsymbol{I}_d)} \Big[ \Big( \sum_{i=1}^{k} \boldsymbol{a}_i^{\top} \boldsymbol{x} \cdot \phi'(\boldsymbol{w}_i^{*\top} \boldsymbol{x}) \Big)^2 \Big] \geq \sum_{l=1}^{L} \lambda_l \frac{\sigma_l^2}{\eta \kappa^2} \rho\Big( \frac{\boldsymbol{W}^{*\top}\boldsymbol{\mu}_l}{\sigma_l \delta_K(\boldsymbol{W}^*)}, \sigma_l \delta_K(\boldsymbol{W}^*) \Big)
\tag{118}
$$

Therefore,

$$
\frac{4}{K^2} \sum_{l=1}^{L} \lambda_l \frac{\sigma_l^2}{\eta \kappa^2} \rho\Big( \frac{\boldsymbol{W}^{*\top}\boldsymbol{\mu}_l}{\sigma_l \delta_K(\boldsymbol{W}^*)}, \sigma_l \delta_k(\boldsymbol{W}^*) \Big) \cdot \boldsymbol{I}_{dK} \preceq \nabla^2 f(\boldsymbol{W}^*) \preceq C_4 \cdot \sum_{l=1}^{L} \lambda_l (\|\boldsymbol{\mu}_l\|_{\infty} + \sigma_l)^2 \cdot \boldsymbol{I}_{dK}
\tag{119}
$$

From (68) in Lemma 6, since that we have the condition $\|\boldsymbol{W} - \boldsymbol{W}^*\|_F \leq r$ and (53), we can obtain

$$
\begin{aligned}
&\|\nabla^2 f(\boldsymbol{W}) - \nabla^2 f(\boldsymbol{W}^*)\| \\
&\leq C_5 K^{\frac{3}{2}} \Big( \sum_{l=1}^{L} \lambda_l (\|\boldsymbol{\mu}_l\|_{\infty} + \sigma_l)^4 \sum_{l=1}^{L} \lambda_l (\|\boldsymbol{\mu}_l\|_{\infty} + \sigma_l)^8 \Big)^{\frac{1}{4}} \|\boldsymbol{W} - \boldsymbol{W}^*\|_F \\
&\leq \frac{4\epsilon_0}{K^2} \sum_{l=1}^{L} \lambda_l \frac{\sigma_l^2}{\eta \kappa^2} \rho\Big( \frac{\boldsymbol{W}^{*\top}\boldsymbol{\mu}_l}{\sigma_l \delta_K(\boldsymbol{W}^*)}, \sigma_l \delta_K(\boldsymbol{W}^*) \Big),
\end{aligned}
\tag{120}
$$

where $\epsilon_0 \in (0, \frac{1}{4})$. Then we have

$$||\nabla^2 f(\boldsymbol{W})|| \geq ||\nabla^2 f(\boldsymbol{W}^*)|| - ||\nabla^2 f(\boldsymbol{W}) - \nabla^2 f(\boldsymbol{W})||$$
$$\geq \frac{4(1-\epsilon_0)}{K^2} \sum_{l=1}^{L} \lambda_l \frac{\sigma_l^2}{\eta\kappa^2} \rho(\frac{\boldsymbol{W}^{*\top}\boldsymbol{\mu}_l}{\sigma_l\delta_K(\boldsymbol{W}^*)}, \sigma_l\delta_K(\boldsymbol{W}^*)) \tag{121}$$

$$||\nabla^2 f(\boldsymbol{W})|| \leq ||\nabla^2 f(\boldsymbol{W}^*)|| + ||\nabla^2 f(\boldsymbol{W}) - \nabla^2 f(\boldsymbol{W})||$$
$$\leq C_4 \cdot \sum_{l=1}^{L} \lambda_l(||\boldsymbol{\mu}_l||_\infty + \sigma_l)^2 + \frac{4}{K^2} \sum_{l=1}^{L} \lambda_l \frac{\sigma_l^2}{\eta\kappa^2} \rho(\frac{\boldsymbol{W}^{*\top}\boldsymbol{\mu}_l}{\sigma_l\delta_K(\boldsymbol{W}^*)}, \sigma_l\delta_K(\boldsymbol{W}^*)) \tag{122}$$
$$\lesssim C_4 \cdot \sum_{l=1}^{L} \lambda_l(||\boldsymbol{\mu}||_\infty + \sigma_l)^2$$

The last inequality of (122) holds since $C_4 \cdot \sum_{l=1} \lambda_l(||\boldsymbol{\mu}||_\infty + \sigma_l)^2 = \Omega(\sigma_{\max}^2)$, $\frac{4}{K^2}\sum_{l=1}^{L}\lambda_l \frac{\sigma_l^2}{\eta\kappa^2}\rho(\frac{\boldsymbol{W}^{*\top}\boldsymbol{\mu}_l}{\sigma_l\delta_K(\boldsymbol{W}^*)}, \sigma_l\delta_K(\boldsymbol{W}^*)) = O(\frac{\sigma_{\max}^2}{K^2})$ and $O(\sigma_{\max}^2) \geq \Omega(\frac{\sigma_{\max}^2}{K^2})$. Combining (121) and (122), we have

$$\frac{4(1-\epsilon_0)}{K^2} \sum_{l=1}^{L} \lambda_l \frac{\sigma_l^2}{\eta\kappa^2} \rho(\frac{\boldsymbol{W}^{*\top}\boldsymbol{\mu}_l}{\sigma_l\delta_K(\boldsymbol{W}^*)}, \sigma_l\delta_K(\boldsymbol{W}^*)) \cdot \boldsymbol{I} \preceq \nabla^2 f(\boldsymbol{W}) \preceq C_4 \cdot \sum_{l=1}^{L} \lambda_l(||\boldsymbol{\mu}||_\infty + \sigma_l)^2 \cdot \boldsymbol{I} \tag{123}$$

## C.7 Proof of Lemma 8

Let $N_\epsilon$ be the $\epsilon$-covering number of the Euclidean ball $\mathbb{B}(\boldsymbol{W}^*, r)$. It is known that $\log N_\epsilon \leq dK\log(\frac{3r}{\epsilon})$ from (Vershynin, 2010). Let $\mathcal{W}_\epsilon = \{\boldsymbol{W}_1, ..., \boldsymbol{W}_{N_\epsilon}\}$ be the $\epsilon$-cover set with $N_\epsilon$ elements. For any $\boldsymbol{W} \in \mathbb{B}(\boldsymbol{W}^*, r)$, let $j(\boldsymbol{W}) = \underset{j\in[N_\epsilon]}{\arg\min} ||\boldsymbol{W} - \boldsymbol{W}_{j(\boldsymbol{W})}||_F \leq \epsilon$ for all $\boldsymbol{W} \in \mathbb{B}(\boldsymbol{W}^*, r)$.

Then for any $\boldsymbol{W} \in \mathbb{B}(\boldsymbol{W}^*, r)$, we have

$$||\nabla^2 f_n(\boldsymbol{W}) - \nabla^2 f(\boldsymbol{W})||$$
$$\leq \frac{1}{n}||\sum_{i=1}^{n}[\nabla^2\ell(\boldsymbol{W};\boldsymbol{x}_i) - \nabla^2\ell(\boldsymbol{W}_{j(\boldsymbol{W})};\boldsymbol{x}_i)]||$$
$$+ ||\frac{1}{n}\sum_{i=1}^{n}\nabla^2\ell(\boldsymbol{W}_{j(\boldsymbol{W})};\boldsymbol{x}_i) - \mathbb{E}_{\boldsymbol{x}\sim\sum_{l=1}^{L}\lambda_l\mathcal{N}(\boldsymbol{\mu}_l,\sigma_l^2\boldsymbol{I}_d)}[\nabla^2\ell(\boldsymbol{W}_{j(\boldsymbol{W})};\boldsymbol{x}_i)]|| \tag{124}$$
$$+ ||\mathbb{E}_{\boldsymbol{x}\sim\sum_{l=1}^{L}\lambda_l\mathcal{N}(\boldsymbol{\mu}_l,\sigma_l^2\boldsymbol{I}_d)}[\nabla^2\ell(\boldsymbol{W}_{j(\boldsymbol{W})};\boldsymbol{x}_i)] - \mathbb{E}_{\boldsymbol{x}\sim\sum_{l=1}^{L}\lambda_l\mathcal{N}(\boldsymbol{\mu}_l,\sigma_l^2\boldsymbol{I}_d)}[\nabla^2\ell(\boldsymbol{W};\boldsymbol{x}_i)]||$$

Hence, we have

$$\mathbb{P}\Big(\sup_{\boldsymbol{W}\in\mathbb{B}(\boldsymbol{W}^*,r)} ||\nabla^2 f_n(\boldsymbol{W}) - \nabla^2 f(\boldsymbol{W})|| \geq t\Big) \leq \mathbb{P}(A_t) + \mathbb{P}(B_t) + \mathbb{P}(C_t) \tag{125}$$

where $A_t$, $B_t$ and $C_t$ are defined as

$$A_t = \{\sup_{\boldsymbol{W}\in\mathbb{B}(\boldsymbol{W}^*,r)} \frac{1}{n}||\sum_{i=1}^{n}[\nabla^2\ell(\boldsymbol{W};\boldsymbol{x}_i) - \nabla^2\ell(\boldsymbol{W}_{j(\boldsymbol{W})};\boldsymbol{x}_i)]|| \geq \frac{t}{3}\} \tag{126}$$

$$B_t = \{\sup_{\boldsymbol{W}\in\mathbb{B}(\boldsymbol{W}^*,r)} ||\frac{1}{n}\sum_{i=1}^{n}\nabla^2\ell(\boldsymbol{W}_{j(\boldsymbol{W})};\boldsymbol{x}_i) - \mathbb{E}_{\boldsymbol{x}\sim\sum_{l=1}^{L}\lambda_l\mathcal{N}(\boldsymbol{\mu}_l,\sigma_l^2\boldsymbol{I}_d)}[\nabla^2\ell(\boldsymbol{W}_{j(\boldsymbol{W})};\boldsymbol{x}_i)]|| \geq \frac{t}{3}\} \tag{127}$$

$$C_t = \{\sup_{\boldsymbol{W}\in\mathbb{B}(\boldsymbol{W}^*,r)} ||\mathbb{E}_{\boldsymbol{x}\sim\sum_{l=1}^{L}\lambda_l\mathcal{N}(\boldsymbol{\mu}_l,\sigma_l^2\boldsymbol{I}_d)}[\nabla^2\ell(\boldsymbol{W}_{j(\boldsymbol{W})};\boldsymbol{x}_i)]$$
$$- \mathbb{E}_{\boldsymbol{x}\sim\sum_{l=1}^{L}\lambda_l\mathcal{N}(\boldsymbol{\mu}_l,\sigma_l^2\boldsymbol{I}_d)}[\nabla^2\ell(\boldsymbol{W};\boldsymbol{x}_i)]|| \geq \frac{t}{3}\} \tag{128}$$

Then we bound $\mathbb{P}(A_t)$, $\mathbb{P}(B_t)$ and $\mathbb{P}(C_t)$ separately.

1) **Upper bound on** $\mathbb{P}(B_t)$. By Lemma 6 in (Fu et al., 2020), we obtain

$$
\begin{aligned}
&\left\| \left\| \frac{1}{n} \sum_{i=1}^{n} \nabla^2 \ell(\boldsymbol{W}; \boldsymbol{x}_i) - \mathbb{E}_{\boldsymbol{x} \sim \sum_{l=1}^{L} \lambda_l \mathcal{N}(\boldsymbol{\mu}_l, \sigma_l^2 \boldsymbol{I}_d)} [\nabla^2 \ell(\boldsymbol{W}; \boldsymbol{x}_i)] \right\| \right\| \\
&\leq 2 \sup_{\boldsymbol{v} \in \boldsymbol{V}_{\frac{1}{4}}} \left| \left\langle \boldsymbol{v}, \left( \frac{1}{n} \sum_{i=1}^{n} \nabla^2 \ell(\boldsymbol{W}; \boldsymbol{x}_i) - \mathbb{E}_{\boldsymbol{x} \sim \sum_{l=1}^{L} \lambda_l \mathcal{N}(\boldsymbol{\mu}_l, \sigma_l^2 \boldsymbol{I}_d)} [\nabla^2 \ell(\boldsymbol{W}; \boldsymbol{x}_i)] \right) \boldsymbol{v} \right\rangle \right|
\end{aligned}
\tag{129}
$$

where $\boldsymbol{V}_{\frac{1}{4}}$ is a $\frac{1}{4}$-cover of the unit-Euclidean-norm ball $\mathbb{B}(\boldsymbol{0}, 1)$ with $\log |\boldsymbol{V}_{\frac{1}{4}}| \leq dK \log 12$. Taking the union bound over $\mathcal{W}_\epsilon$ and $\boldsymbol{V}_{\frac{1}{4}}$, we have

$$
\begin{aligned}
\mathbb{P}(B_t) \leq & \mathbb{P} \left( \sup_{\boldsymbol{W} \in \mathcal{W}_\epsilon, \boldsymbol{v} \in \boldsymbol{V}_{\frac{1}{4}}} \left| \frac{1}{n} \sum_{i=1}^{n} G_i \right| \geq \frac{t}{6} \right) \\
\leq & \exp \left( dK \left( \log \frac{3r}{\epsilon} + \log 12 \right) \right) \sup_{\boldsymbol{W} \in \mathcal{W}_\epsilon, \boldsymbol{v} \in \boldsymbol{V}_{\frac{1}{4}}} \mathbb{P} \left( \left| \frac{1}{n} \sum_{i=1}^{n} G_i \right| \geq \frac{t}{6} \right)
\end{aligned}
\tag{130}
$$

where $G_i = \left\langle \boldsymbol{v}, (\nabla^2 \ell(\boldsymbol{W}, \boldsymbol{x}_i) - \mathbb{E}_{\boldsymbol{x} \sim \sum_{l=1}^{L} \lambda_l \mathcal{N}(\boldsymbol{\mu}_l, \sigma_l^2 \boldsymbol{I}_d)} [\nabla^2 \ell(\boldsymbol{W}, \boldsymbol{x}_i)] \boldsymbol{v} \right\rangle$ and $\mathbb{E}[G_i] = 0$. Here $\boldsymbol{v} = (\boldsymbol{u}_1^\top, \cdots, \boldsymbol{u}_K^\top)^\top \in \mathbb{R}^{dK}$.

$$
\begin{aligned}
|G_i| = & \left| \sum_{j=1}^{K} \sum_{l=1}^{K} \left[ \xi_{j,l} \boldsymbol{u}_j^\top \boldsymbol{x} \boldsymbol{x}^\top \boldsymbol{u}_l - \mathbb{E}_{\boldsymbol{x} \sim \sum_{l=1}^{L} \lambda_l \mathcal{N}(\boldsymbol{\mu}_l, \sigma_l^2 \boldsymbol{I})} (\xi_{j,l} \boldsymbol{u}_j^\top \boldsymbol{x} \boldsymbol{x}^\top \boldsymbol{u}_l) \right] \right| \\
\leq & C_9 \cdot \left[ \sum_{j=1}^{K} (\boldsymbol{u}_j^\top \boldsymbol{x})^2 + \sum_{j=1}^{K} \mathbb{E}_{\boldsymbol{x} \sim \sum_{l=1}^{L} \lambda_l \mathcal{N}(\boldsymbol{\mu}_l, \sigma_l^2 \boldsymbol{I}_d)} (\boldsymbol{u}_j^\top \boldsymbol{x})^2 \right]
\end{aligned}
\tag{131}
$$

for some $C_9 > 0$. The first step of (131) is by (44). The last step is by (93) and the Cauchy-Schwarz's Inequality.

$$
\begin{aligned}
\mathbb{E}[|G_i|^p] \leq & \sum_{l=1}^{p} \binom{p}{l} C_9 \cdot \mathbb{E}_{\boldsymbol{x} \sim \sum_{l=1}^{L} \lambda_l \mathcal{N}(\boldsymbol{\mu}_l, \sigma_l^2 \boldsymbol{I}_d)} \\
& \cdot \Big[ \big( \sum_{j=1}^{K} (\boldsymbol{u}_j^\top \boldsymbol{x})^2 \big)^l \Big] \Big( \sum_{j=1}^{K} \mathbb{E}_{\boldsymbol{x} \sim \sum_{l=1}^{L} \lambda_l \mathcal{N}(\boldsymbol{\mu}_l, \sigma_l^2 \boldsymbol{I}_d)} (\boldsymbol{u}_j^\top \boldsymbol{x})^2 \Big)^{p-l} \\
= & \sum_{l=1}^{p} \binom{p}{l} C_9 \cdot \mathbb{E}_{\boldsymbol{x} \sim \sum_{l=1}^{L} \lambda_l \mathcal{N}(\boldsymbol{\mu}_l, \sigma_l^2 \boldsymbol{I}_d)} \Big[ \sum_{l_1 + \cdots + l_K = l} \frac{l!}{\prod_{j=1}^{K} l_j!} \prod_{j=1}^{K} (\boldsymbol{u}_j^\top \boldsymbol{x})^{2l_j} \Big] \\
& \cdot \Big( \sum_{j=1}^{K} \mathbb{E}_{\boldsymbol{x} \sim \sum_{l=1}^{L} \lambda_l \mathcal{N}(\boldsymbol{\mu}_l, \sigma_l^2 \boldsymbol{I}_d)} (\boldsymbol{u}_j^\top \boldsymbol{x})^2 \Big)^{p-l} \\
= & \sum_{l=1}^{p} \binom{p}{l} C_9 \cdot \Big[ \sum_{l_1 + \cdots + l_K = l} \frac{l!}{\prod_{j=1}^{K} l_j!} \prod_{j=1}^{K} \mathbb{E}_{\boldsymbol{x} \sim \sum_{l=1}^{L} \lambda_l \mathcal{N}(\boldsymbol{\mu}_l, \sigma_l^2 \boldsymbol{I}_d)} (\boldsymbol{u}_j^\top \boldsymbol{x})^{2l_j} \Big] \\
& \cdot \Big( \sum_{j=1}^{K} \mathbb{E}_{\boldsymbol{x} \sim \sum_{l=1}^{L} \lambda_l \mathcal{N}(\boldsymbol{\mu}_l, \sigma_l^2 \boldsymbol{I}_d)} (\boldsymbol{u}_j^\top \boldsymbol{x})^2 \Big)^{p-l} \\
= & C_9 \cdot \sum_{l=1}^{p} \binom{p}{l} \Big( \sum_{j=1}^{K} \mathbb{E}_{\boldsymbol{x} \sim \sum_{l=1}^{L} \lambda_l \mathcal{N}(\boldsymbol{\mu}_l, \sigma_l^2 \boldsymbol{I}_d)} (\boldsymbol{u}_j^\top \boldsymbol{x})^2 \Big)^l \\
& \cdot \Big( \sum_{j=1}^{K} \mathbb{E}_{\boldsymbol{x} \sim \sum_{l=1}^{L} \lambda_l \mathcal{N}(\boldsymbol{\mu}_l, \sigma_l^2 \boldsymbol{I}_d)} (\boldsymbol{u}_j^\top \boldsymbol{x})^2 \Big)^{p-l} \\
= & C_9 \cdot \Big( \sum_{j=1}^{K} \mathbb{E}_{\boldsymbol{x} \sim \sum_{l=1}^{L} \lambda_l \mathcal{N}(\boldsymbol{\mu}_l, \sigma_l^2 \boldsymbol{I}_d)} (\boldsymbol{u}_j^\top \boldsymbol{x})^2 \Big)^p \\
\leq & C_9 \cdot \Big( \sum_{j=1}^{K} 1!! \|\boldsymbol{u}_j\|^2 \sum_{l=1}^{L} \lambda_l (\|\boldsymbol{\mu}_l\|_\infty + \sigma_l)^2 \Big)^p \\
\leq & C_9 \cdot \Big( \sum_{l=1}^{L} \lambda_l (\|\boldsymbol{\mu}_l\|_\infty + \sigma_l)^2 \Big)^p
\end{aligned}
\tag{132}
$$

where the second to last inequality results from Property 4. The last inequality is because $\boldsymbol{v} \in \boldsymbol{V}_{\frac{1}{4}}$, $\sum_{j=1}^{K} \|u_j\|^2 = \|\boldsymbol{v}\|^2 \leq 1$.

$$
\begin{aligned}
\mathbb{E}[\exp(\theta G_i)] &= 1 + \theta \mathbb{E}[G_i] + \sum_{p=2}^{\infty} \frac{\theta^p \mathbb{E}[|G_i|^p]}{p!} \\
&\leq 1 + \sum_{p=2}^{\infty} \frac{|e\theta|^p}{p^p} C_9 \cdot \Big( \sum_{l=1}^{L} \lambda_l (\|\boldsymbol{\mu}_l\|_\infty + \sigma_l)^2 \Big)^p \\
&\leq 1 + C_9 \cdot |e\theta|^2 \Big( \sum_{l=1}^{L} \lambda_l (\|\boldsymbol{\mu}_l\|_\infty + \sigma_l)^2 \Big)^2
\end{aligned}
\tag{133}
$$

where the first inequality holds from $p! \geq (\frac{p}{e})^p$ and (132), and the third line holds provided that

$$
\max_{p \geq 2} \Big\{ \frac{\frac{|e\theta|^{(p+1)}}{(p+1)^{(p+1)}} \cdot \Big( \sum_{l=1}^{L} \lambda_l (\|\boldsymbol{\mu}_l\|_\infty + \sigma_l)^2 \Big)^{p+1}}{\frac{|e\theta|^p}{p^p} \cdot \Big( \sum_{l=1}^{L} \lambda_l (\|\boldsymbol{\mu}_l\|_\infty + \sigma_l)^2 \Big)^p} \Big\} \leq \frac{1}{2}
\tag{134}
$$

Note that the quantity inside the maximization in (134) achieves its maximum when $p = 2$, because it is monotonously decreasing. Therefore, (134) holds if $\theta \leq \frac{27}{4e} \sum_{l=1}^{L} \lambda_l (\|\boldsymbol{\mu}_l\|_\infty + \sigma_l)^2$. Then

$$\mathbb{P}\Big(\frac{1}{n}\sum_{i=1}^{n} G_i \geq \frac{t}{6}\Big) = \mathbb{P}\Big(\exp(\theta \sum_{i=1}^{n} G_i) \geq \exp(\frac{n\theta t}{6})\Big) \leq e^{-\frac{n\theta t}{6}} \prod_{i=1}^{n} \mathbb{E}[\exp(\theta G_i)]$$
$$\leq \exp(C_{10}\theta^2 n \Big(\sum_{l=1}^{L} \lambda_l(\|\boldsymbol{\mu}_l\|_\infty + \sigma_l)^2\Big)^2 - \frac{n\theta t}{6})$$
(135)

for some constant $C_{10} > 0$. The first inequality follows from the Markov's Inequality. When $\theta = \min\{\frac{t}{12C_{10}\Big(\sum_{l=1}^{L}\lambda_l(\|\boldsymbol{\mu}_l\|_\infty+\sigma_l)^2\Big)^2}, \frac{27}{4e}\sum_{l=1}^{L}\lambda_l(\|\boldsymbol{\mu}_l\|_\infty + \sigma_l)^2\}$, we have a modified Bernstein's Inequality for the Gaussian Mixture Model as follows

$$\mathbb{P}(\frac{1}{n}\sum_{i=1}^{n} G_i \geq \frac{t}{6}) \leq \exp\Big(\max\{-\frac{C_{10}nt^2}{144\Big(\sum_{l=1}^{L}\lambda_l(\|\boldsymbol{\mu}_l\|_\infty + \sigma_l)^2\Big)^2},$$
$$- C_{11}n\sum_{l=1}^{L}\lambda_l(\|\boldsymbol{\mu}_l\|_\infty + \sigma_l)^2 \cdot t\}\Big)$$
(136)

for some constant $C_{11} > 0$. We can obtain the same bound for $\mathbb{P}(-\frac{1}{n}\sum_{i=1}^{n} G_i \geq \frac{t}{6})$ by replacing $G_i$ as $-G_i$. Therefore, we have

$$\mathbb{P}(|\frac{1}{n}\sum_{i=1}^{n} G_i| \geq \frac{t}{6}) \leq 2\exp\Big(\max\{-\frac{C_{10}nt^2}{144\Big(\sum_{l=1}^{L}\lambda_l(\|\boldsymbol{\mu}_l\|_\infty + \sigma_l)^2\Big)^2},$$
$$- C_{11}n\sum_{l=1}^{L}\lambda_l(\|\boldsymbol{\mu}_l\|_\infty + \sigma_l)^2 \cdot t\}\Big)$$
(137)

Thus, as long as

$$t \geq C_6 \cdot \max\{\sum_{l=1}^{L}\lambda_l(\|\boldsymbol{\mu}_l\|_\infty + \sigma_l)^2 \sqrt{\frac{dK\log\frac{36r}{\epsilon} + \log\frac{4}{\delta}}{n}}, \frac{dK\log\frac{36r}{\epsilon} + \log\frac{4}{\delta}}{\sum_{l=1}^{L}\lambda_l(\|\boldsymbol{\mu}_l\|_\infty + \sigma_l)^2 n}\} \quad (138)$$

for some large constant $C_6 > 0$, we have $\mathbb{P}(B_t) \leq \frac{\delta}{2}$.

2) **Upper bound on** $\mathbb{P}(A_t)$ **and** $\mathbb{P}(C_t)$. From Lemma 5, we can obtain

$$\sup_{\boldsymbol{W}\in\mathbb{B}(\boldsymbol{W}^*,r)} ||\mathbb{E}_{\boldsymbol{x}\sim\sum_{l=1}^{L}\lambda_l\mathcal{N}(\boldsymbol{\mu}_l,\sigma_l^2\boldsymbol{I}_d)}[\nabla^2\ell(\boldsymbol{W}_{j(\boldsymbol{W})};\boldsymbol{x})] - \mathbb{E}_{\boldsymbol{x}\sim\sum_{l=1}^{L}\lambda_l\mathcal{N}(\boldsymbol{\mu}_l,\sigma_l^2\boldsymbol{I}_d)}[\nabla^2\ell(\boldsymbol{W};\boldsymbol{x})]||$$

$$\leq \sup_{\boldsymbol{W}\in\mathbb{B}(\boldsymbol{W}^*,r)} \frac{||\mathbb{E}_{\boldsymbol{x}\sim\sum_{l=1}^{L}\lambda_l\mathcal{N}(\boldsymbol{\mu}_l,\sigma_l^2\boldsymbol{I}_d)}[\nabla^2\ell(\boldsymbol{W}_{j(\boldsymbol{W})};\boldsymbol{x})] - \mathbb{E}_{\boldsymbol{x}\sim\sum_{l=1}^{L}\lambda_l\mathcal{N}(\boldsymbol{\mu}_l,\sigma_l^2\boldsymbol{I}_d)}[\nabla^2\ell(\boldsymbol{W};\boldsymbol{x})]||}{||\boldsymbol{W} - \boldsymbol{W}_{j(\boldsymbol{W})}||_F}$$
$$\cdot \sup_{\boldsymbol{W}\in\mathbb{B}(\boldsymbol{W}^*,r)} ||\boldsymbol{W} - \boldsymbol{W}_{j(\boldsymbol{W})}||_F$$

$$\leq C_{12} \cdot d^{\frac{3}{2}} K^{\frac{5}{2}} \sqrt{\sum_{l=1}^{L}\lambda_l(\|\boldsymbol{\mu}_l\|_\infty + \sigma_l)^2 \sum_{l=1}^{L}\lambda_l(\|\boldsymbol{\mu}_l\|_\infty + \sigma_l)^4} \cdot \epsilon$$
(139)

Therefore, $C_t$ holds if

$$t \geq C_{12} \cdot d^{\frac{3}{2}} K^{\frac{5}{2}} \sqrt{\sum_{l=1}^{L}\lambda_l(\|\boldsymbol{\mu}_l\|_\infty + \sigma_l)^2 \sum_{l=1}^{L}\lambda_l(\|\boldsymbol{\mu}_l\|_\infty + \sigma_l)^4} \cdot \epsilon \quad (140)$$

We can bound the $A_t$ as below.

$$\mathbb{P}\Big(\sup_{\boldsymbol{W}\in\mathbb{B}(\boldsymbol{W}^*,r)}\frac{1}{n}||\sum_{i=1}^{n}[\nabla^2\ell(\boldsymbol{W}_{j(\boldsymbol{W})};\boldsymbol{x}_i)-\nabla^2\ell(\boldsymbol{W};\boldsymbol{x}_i)]||\geq\frac{t}{3}\Big)$$

$$\leq\frac{3}{t}\mathbb{E}_{\boldsymbol{x}\sim\sum_{l=1}^{L}\lambda_l\mathcal{N}(\boldsymbol{\mu}_l,\sigma_l^2\boldsymbol{I}_d)}\Big[\sup_{\boldsymbol{W}\in\mathbb{B}(\boldsymbol{W}^*,r)}\frac{1}{n}||\sum_{i=1}^{n}[\nabla^2\ell(\boldsymbol{W}_{j(\boldsymbol{W})};\boldsymbol{x}_i)-\nabla^2\ell(\boldsymbol{W};\boldsymbol{x}_i)]||\Big]$$

$$=\frac{3}{t}\mathbb{E}_{\boldsymbol{x}\sim\sum_{l=1}^{L}\lambda_l\mathcal{N}(\boldsymbol{\mu}_l,\sigma_l^2\boldsymbol{I}_d)}\Big[\sup_{\boldsymbol{W}\in\mathbb{B}(\boldsymbol{W}^*,r)}||\nabla^2\ell(\boldsymbol{W}_{j(\boldsymbol{W})};\boldsymbol{x}_i)-\nabla^2\ell(\boldsymbol{W};\boldsymbol{x}_i)||\Big] \qquad (141)$$

$$\leq\frac{3}{t}\mathbb{E}\Big[\sup_{\boldsymbol{W}\in\mathbb{B}(\boldsymbol{W}^*,r)}\frac{||\nabla^2\ell(\boldsymbol{W}_{j(\boldsymbol{W})};\boldsymbol{x}_i)-\nabla^2\ell(\boldsymbol{W};\boldsymbol{x}_i)||}{||\boldsymbol{W}-\boldsymbol{W}_{j(\boldsymbol{W})}||_F}\Big]\cdot\sup_{\boldsymbol{W}\in\mathbb{B}(\boldsymbol{W}^*,r)}||\boldsymbol{W}-\boldsymbol{W}_{j(\boldsymbol{W})}||_F$$

$$\leq\frac{C_{12}\cdot d^{\frac{3}{2}}K^{\frac{5}{2}}\sqrt{\sum_{l=1}^{L}\lambda_l(||\boldsymbol{\mu}_l||_\infty+\sigma_l)^2\sum_{l=1}^{L}\lambda_l(||\boldsymbol{\mu}_l||_\infty+\sigma_l)^4}\cdot\epsilon}{t}$$

Thus, taking

$$t\geq\frac{C_{12}\cdot d^{\frac{3}{2}}K^{\frac{5}{2}}\sqrt{\sum_{l=1}^{L}\lambda_l(||\boldsymbol{\mu}_l||_\infty+\sigma_l)^2\sum_{l=1}^{L}\lambda_l(||\boldsymbol{\mu}_l||_\infty+\sigma_l)^4}\cdot\epsilon}{\delta} \qquad (142)$$

ensures that $\mathbb{P}(A_t)\leq\frac{\delta}{2}$.

3) **Final step**
Let $\epsilon=\frac{\delta}{C_{12}\cdot d^{\frac{3}{2}}K^{\frac{5}{2}}\sqrt{\sum_{l=1}^{L}\lambda_l(||\boldsymbol{\mu}_l||_\infty+\sigma_l)^2\sum_{l=1}^{L}\lambda_l(||\boldsymbol{\mu}_l||_\infty+\sigma_l)^4}\cdot ndK}$ and $\delta=d^{-10}$, then from (138) and (142) we need

$$t>\max\{\frac{1}{ndK},\ C_6\cdot\sum_{l=1}^{L}\lambda_l(||\boldsymbol{\mu}_l||_\infty+\sigma_l)^2$$

$$\cdot\sqrt{\frac{dK\log(36rnd^{\frac{25}{2}}K^{\frac{7}{2}}\sqrt{\sum_{l=1}^{L}\lambda_l(||\boldsymbol{\mu}_l||_\infty+\sigma_l)^2\sum_{l=1}^{L}\lambda_l(||\boldsymbol{\mu}_l||_\infty+\sigma_l)^4})+\log\frac{4}{\delta}}{n}},$$

$$\frac{dK\log(36rnd^{\frac{25}{2}}K^{\frac{7}{2}}\cdot\sqrt{\sum_{l=1}^{L}\lambda_l(||\boldsymbol{\mu}_l||_\infty+\sigma_l)^2\sum_{l=1}^{L}\lambda_l(||\boldsymbol{\mu}_l||_\infty+\sigma_l)^4})+\log\frac{4}{\delta}}{\sum_{l=1}^{L}\lambda_l(||\boldsymbol{\mu}_l||_\infty+\sigma_l)^2n}\}$$

$$(143)$$

So by setting $t=\sum_{l=1}^{L}\lambda_l(||\boldsymbol{\mu}_l||_\infty+\sigma_l)^2\sqrt{\frac{dK\log n}{n}}$, as long as $n\geq C'\cdot dK\log dK$, we have

$$\mathbb{P}(\sup_{\boldsymbol{W}\in\mathbb{B}(\boldsymbol{W}^*,r)}||\nabla^2 f_n(\boldsymbol{W})-\nabla^2 f(\boldsymbol{W})||\geq C_6\cdot\sum_{l=1}^{L}\lambda_l(||\boldsymbol{\mu}_l||_\infty+\sigma_l)^2\sqrt{\frac{dK\log n}{n}})\leq d^{-10} \quad (144)$$

## D  PROOF OF LEMMA 2

We first present a lemma used in proving Lemma 2 in Section D.1 and then prove Lemma 2 in Section D.2.

### D.1  A USEFUL LEMMA USED IN THE PROOF

**Lemma 9** *If $r$ is defined in (53) for $\epsilon_0\in(0,\frac{1}{4})$, then with probability at least $1-d^{-10}$, we have*[4]

---

[4] $\nabla\tilde{f}_n(\boldsymbol{W})$ is defined as $\frac{1}{n}\sum_{i=1}^{n}(\nabla l(\boldsymbol{W},\boldsymbol{x}_i,y_i)+\nu_i)$ in algorithm 1

$$\sup_{\boldsymbol{W}\in\mathbb{B}(\boldsymbol{W}^*,r)} ||\nabla \tilde{f}_n(\boldsymbol{W}) - \nabla \tilde{f}(\boldsymbol{W})|| \leq C_{13} \cdot \sqrt{K\sum_{l=1}^{L}\lambda_l(||\boldsymbol{\mu}||_\infty + \sigma_l)^2}\sqrt{\frac{d\log n}{n}}(1+\xi) \quad (145)$$

*for some constant $C_{13} > 0$.*

**Proof:**
Note that $\nabla \tilde{f}_n(\boldsymbol{W}) = \nabla f_n(\boldsymbol{W}) + \frac{1}{n}\sum_{i=1}^{n}\nu_i$, $\nabla \tilde{f}(\boldsymbol{W}) = \nabla f(\boldsymbol{W}) + \mathbb{E}[\nu_i] = \nabla f(\boldsymbol{W})$. Therefore, we have

$$\sup_{\boldsymbol{W}\in\mathbb{B}(\boldsymbol{W}^*,r)} ||\nabla \tilde{f}_n(\boldsymbol{W}) - \nabla \tilde{f}(\boldsymbol{W})|| \leq \sup_{\boldsymbol{W}\in\mathbb{B}(\boldsymbol{W}^*,r)} ||\nabla f_n(\boldsymbol{W}) - \nabla f(\boldsymbol{W})|| + ||\frac{1}{n}\sum_{i=1}^{n}\nu_i|| \quad (146)$$

Then, similar to the idea of the proof of Lemma 8, we adopt an $\epsilon$-covering net of the ball $\mathbb{B}(\boldsymbol{W}^*, r)$ to build a relationship between any arbitrary point in the ball and the points in the covering set. We can then divide the distance between $\nabla f_n(\boldsymbol{W})$ and $\nabla f(\boldsymbol{W})$ into three parts, similar to (124). (147) to (149) can be derived in a similar way as (126) to (128), with "$\nabla^2$" replaced by "$\nabla$". Then we need to bound $\mathbb{P}(A'_t)$, $\mathbb{P}(B'_t)$ and $\mathbb{P}(C'_t)$ respectively, where $A'_t$, $B'_t$ and $C'_t$ are defined below.

$$A'_t = \{\sup_{\boldsymbol{W}\in\mathbb{B}(\boldsymbol{W}^*,r)} \frac{1}{n}||\sum_{i=1}^{n}[\nabla\ell(\boldsymbol{W};\boldsymbol{x}_i) - \nabla\ell(\boldsymbol{W}_{j(\boldsymbol{W})};\boldsymbol{x}_i)]|| \geq \frac{t}{3}\} \quad (147)$$

$$B'_t = \{\sup_{\boldsymbol{W}\in\mathbb{B}(\boldsymbol{W}^*,r)} ||\frac{1}{n}\sum_{i=1}^{n}\nabla\ell(\boldsymbol{W}_{j(\boldsymbol{W})};\boldsymbol{x}_i) - \mathbb{E}_{\boldsymbol{x}\sim\sum_{l=1}^{L}\lambda_l\mathcal{N}(\boldsymbol{\mu}_l,\sigma_l^2\boldsymbol{I}_d)}[\nabla\ell(\boldsymbol{W}_{j(\boldsymbol{W})};\boldsymbol{x}_i)]|| \geq \frac{t}{3}\} \quad (148)$$

$$C'_t = \{\sup_{\boldsymbol{W}\in\mathbb{B}(\boldsymbol{W}^*,r)} ||\mathbb{E}_{\boldsymbol{x}\sim\sum_{l=1}^{L}\lambda_l\mathcal{N}(\boldsymbol{\mu}_l,\sigma_l^2\boldsymbol{I}_d)}[\nabla\ell(\boldsymbol{W}_{j(\boldsymbol{W})};\boldsymbol{x}_i)] \\ - \mathbb{E}_{\boldsymbol{x}\sim\sum_{l=1}^{L}\lambda_l\mathcal{N}(\boldsymbol{\mu}_l,\sigma_l^2\boldsymbol{I}_d)}[\nabla\ell(\boldsymbol{W};\boldsymbol{x}_i)]|| \geq \frac{t}{3}\} \quad (149)$$

(a) Upper bound of $\mathbb{P}(B'_t)$. Applying Lemma 3 in (Mei et al., 2018a), we have

$$||\frac{1}{n}\sum_{i=1}^{n}\nabla\ell(\boldsymbol{W}_{j(\boldsymbol{W})};\boldsymbol{x}_i) - \mathbb{E}_{\boldsymbol{x}\sim\sum_{l=1}^{L}\lambda_l\mathcal{N}(\boldsymbol{\mu}_l,\sigma_l^2\boldsymbol{I}_d)}[\nabla\ell(\boldsymbol{W}_{j(\boldsymbol{W})};\boldsymbol{x}_i)]||$$

$$\leq 2\sup_{\boldsymbol{v}\in V_{\frac{1}{2}}}\left|\left\langle\frac{1}{n}\sum_{i=1}^{n}\nabla\ell(\boldsymbol{W}_{j(\boldsymbol{W})};\boldsymbol{x}_i) - \mathbb{E}_{\boldsymbol{x}\sim\sum_{l=1}^{L}\lambda_l\mathcal{N}(\boldsymbol{\mu}_l,\sigma_l^2\boldsymbol{I}_d)}[\nabla\ell(\boldsymbol{W}_{j(\boldsymbol{W})};\boldsymbol{x}_i),\boldsymbol{v}\right\rangle\right| \quad (150)$$

Define $G'_i = \left\langle\boldsymbol{v}, (\nabla\ell(\boldsymbol{W},\boldsymbol{x}_i) - \mathbb{E}_{\boldsymbol{x}\sim\sum_{l=1}^{L}\lambda_l\mathcal{N}(\boldsymbol{\mu}_l,\sigma_l^2\boldsymbol{I}_d)}[\nabla\ell(\boldsymbol{W},\boldsymbol{x}_i)])\right\rangle$. Here $\boldsymbol{v}\in\mathbb{R}^d$. To compute $\nabla\ell(\boldsymbol{W},\boldsymbol{x}_i)$, we require the derivation in Property 6. Then we can have an upper bound of $\zeta(\boldsymbol{W})$ in (43).

$$\zeta(\boldsymbol{W}) = \begin{cases} \left|-\frac{1}{K}\frac{1}{H(\boldsymbol{W})}\phi'(\boldsymbol{w}_j^\top\boldsymbol{x})\right| \leq \frac{\phi(\boldsymbol{w}_j^\top\boldsymbol{x})(1-\phi(\boldsymbol{w}_j^\top\boldsymbol{x}))}{K\cdot\frac{1}{K}\phi(\boldsymbol{w}_j^\top\boldsymbol{x})} \leq 1, & y = 1 \\ \left|\frac{1}{K}\frac{1}{1-H(\boldsymbol{W})}\phi'(\boldsymbol{w}_j^\top\boldsymbol{x})\right| \leq \frac{\phi(\boldsymbol{w}_j^\top\boldsymbol{x})(1-\phi(\boldsymbol{w}_j^\top\boldsymbol{x}))}{K\cdot\frac{1}{K}(1-\phi(\boldsymbol{w}_j^\top\boldsymbol{x}))} \leq 1, & y = 0 \end{cases} \quad (151)$$

Then we have an upper bound of $G'_i$.

$$|G'_i| = \left|\zeta_{j,l}\boldsymbol{v}^\top\boldsymbol{x} - \mathbb{E}_{\boldsymbol{x}\sim\sum_{l=1}^{L}\lambda_l\mathcal{N}(\boldsymbol{\mu}_l,\sigma_l^2\boldsymbol{I}_d)}[\zeta\boldsymbol{v}^\top\boldsymbol{x}]\right|$$

$$\leq |\boldsymbol{v}^\top\boldsymbol{x}| + \mathbb{E}_{\boldsymbol{x}\sim\sum_{l=1}^{L}\lambda_l\mathcal{N}(\boldsymbol{\mu}_l,\sigma_l^2\boldsymbol{I}_d)}[|\boldsymbol{v}^\top\boldsymbol{x}|] \quad (152)$$

Following the idea of (132) and (133), and by $\boldsymbol{v}\in V_{\frac{1}{2}}$, we have

$$\mathbb{E}[|G'_i|^p] \leq \left(\sum_{l=1}^{L}\lambda_l(||\boldsymbol{\mu}_l||_\infty + \sigma_l)^2\right)^{\frac{p}{2}} \quad (153)$$

$$\mathbb{E}[\exp(\theta G'_i)] \leq 1 + |e\theta^2|\sum_{l=1}^{L}\lambda_l(||\boldsymbol{\mu}_l||_\infty + \sigma_l)^2 \quad (154)$$

where (154) holds if $\theta \leq \frac{27}{4e}\sqrt{\sum_{l=1}^{L}\lambda_l(\|\boldsymbol{\mu}_l\|_\infty + \sigma_l)^2}$. Following the derivation of (130) and (135) to (138), we have

$$\mathbb{P}(|\frac{1}{n}\sum_{i=1}^{n}G_i'| \geq \frac{t}{6})$$

$$\leq 2\exp\Big(\max\Big\{-\frac{C_{14}nt^2}{144\sum_{l=1}^{L}\lambda_l(\|\boldsymbol{\mu}_l\|_\infty + \sigma_l)^2}, -C_{15}n\sqrt{\sum_{l=1}^{L}\lambda_l(\|\boldsymbol{\mu}_l\|_\infty + \sigma_l)^2 \cdot t}\Big\}\Big) \tag{155}$$

for some constant $C_{14} > 0$ and $C_{15} > 0$. Moreover, we can obtain $\mathbb{P}(B_t') \leq \frac{\delta}{2}$ as long as

$$t \geq C_{13} \cdot \max\Big\{\sqrt{\sum_{l=1}^{L}\lambda_l(\|\boldsymbol{\mu}_l\|_\infty + \sigma_l)^2}\sqrt{\frac{dK\log\frac{18r}{\epsilon} + \log\frac{4}{\delta}}{n}}, \frac{dK\log\frac{18r}{\epsilon} + \log\frac{4}{\delta}}{\sqrt{\sum_{l=1}^{L}\lambda_l(\|\boldsymbol{\mu}_l\|_\infty + \sigma_l)^2 \cdot n}}\Big\} \tag{156}$$

(b) For the upper bound of $\mathbb{P}(A_t')$ and $\mathbb{P}(C_t')$, we can first derive

$$\mathbb{E}_{\boldsymbol{x}\sim\sum_{l=1}^{L}\lambda_l\mathcal{N}(\boldsymbol{\mu}_l,\sigma_l^2\boldsymbol{I}_d)}\Big[\sup_{\boldsymbol{W}\neq\boldsymbol{W}'\in\mathbb{B}(\boldsymbol{W}^*,r)}\frac{\|\nabla\ell(\boldsymbol{W},\boldsymbol{x}) - \nabla\ell(\boldsymbol{W}',\boldsymbol{x})\|}{\|\boldsymbol{W} - \boldsymbol{W}'\|_F}\Big]$$

$$\leq \mathbb{E}_{\boldsymbol{x}\sim\sum_{l=1}^{L}\lambda_l\mathcal{N}(\boldsymbol{\mu}_l,\sigma_l^2\boldsymbol{I}_d)}\Big[\sup_{\boldsymbol{W}\neq\boldsymbol{W}'\in\mathbb{B}(\boldsymbol{W}^*,r)}\frac{|\zeta(\boldsymbol{W}) - \zeta(\boldsymbol{W}')| \cdot \|\boldsymbol{x}\|}{\|\boldsymbol{W} - \boldsymbol{W}'\|_F}\Big]$$

$$\leq \mathbb{E}_{\boldsymbol{x}\sim\sum_{l=1}^{L}\lambda_l\mathcal{N}(\boldsymbol{\mu}_l,\sigma_l^2\boldsymbol{I}_d)}\Big[\sup_{\boldsymbol{W}\neq\boldsymbol{W}'\in\mathbb{B}(\boldsymbol{W}^*,r)}\frac{\max_{1\leq j,l\leq K}\{|\xi_{j,l}(\boldsymbol{W}'')|\} \cdot \|\boldsymbol{x}\|^2\sqrt{K}\|\boldsymbol{W} - \boldsymbol{W}'\|_F}{\|\boldsymbol{W} - \boldsymbol{W}'\|_F}\Big]$$

$$\leq \mathbb{E}_{\boldsymbol{x}\sim\sum_{l=1}^{L}\lambda_l\mathcal{N}(\boldsymbol{\mu}_l,\sigma_l^2\boldsymbol{I}_d)}\Big[\sup_{\boldsymbol{W}\neq\boldsymbol{W}'\in\mathbb{B}(\boldsymbol{W}^*,r)}\frac{C_9 \cdot \|\boldsymbol{x}\|^2\sqrt{K}\|\boldsymbol{W} - \boldsymbol{W}'\|_F}{\|\boldsymbol{W} - \boldsymbol{W}'\|_F}\Big]$$

$$\leq C_9 \cdot 3\sqrt{K}d \cdot \sum_{l=1}^{L}\lambda_l(\|\boldsymbol{\mu}_l\|_\infty + \sigma_l)^2 \tag{157}$$

The first inequality is by (43). The second inequality is by the Mean Value Theorem. The third step is by (93). The last inequality is by Property 5. Therefore, following the steps in part (2) of Lemma 8, we can conclude that $C_t'$ holds if

$$t \geq 3C_9 \cdot \sqrt{K}d \cdot \sum_{l=1}^{L}\lambda_l(\|\boldsymbol{\mu}_l\|_\infty + \sigma_l)^2 \cdot \epsilon \tag{158}$$

Moreover, from (142) in Lemma 8 we have that

$$t \geq \frac{18C_9 \cdot \sqrt{K}d \cdot \sum_{l=1}^{L}\lambda_l(\|\boldsymbol{\mu}_l\|_\infty + \sigma_l)^2 \cdot \epsilon}{\delta} \tag{159}$$

ensures $\mathbb{P}(A_t') \leq \frac{\delta}{2}$. Therefore, let $\epsilon = \frac{\delta}{18C_9 \cdot \sqrt{K}d \cdot \sum_{l=1}^{L}\lambda_l(\|\boldsymbol{\mu}_l\|_\infty + \sigma_l)^2 \cdot \epsilon \cdot ndK}$, $\delta = d^{-10}$ and $t = C_{13}\sqrt{K\sum_{l=1}^{L}\lambda_l(\|\boldsymbol{\mu}_l\|_\infty + \sigma_l)^2}\sqrt{\frac{d\log n}{n}}$, if $n \geq C'' \cdot dK\log dK$ for some constant $C'' > 0$, we have

$$\mathbb{P}\big(\sup_{\boldsymbol{W}\in\mathbb{B}(\boldsymbol{W}^*,r)}\|\nabla f_n(\boldsymbol{W}) - \nabla f(\boldsymbol{W})\|\big) \geq C_{13} \cdot \sqrt{K\sum_{l=1}^{L}\lambda_l(\|\boldsymbol{\mu}_l\|_\infty + \sigma_l)^2}\sqrt{\frac{d\log n}{n}} \leq d^{-10} \tag{160}$$

By Hoeffding's inequality in (Vershynin, 2010) and Property 1, we have

$$\mathbb{P}\Big(\frac{1}{n}\sum_{i=1}^{n}\|\nu_i\|_F \geq C_{13} \cdot \sqrt{\sum_{l=1}^{L}\lambda_l(\|\boldsymbol{\mu}_l\|_\infty + \sigma_l)^2}\sqrt{\frac{dK\log n}{n}\xi}\Big)$$

$$\lesssim \exp(-C_{13}^2 \cdot \sum_{l=1}^{L}\lambda_l(\|\boldsymbol{\mu}_l\|_\infty + \sigma_l)^2\frac{\xi^2 dK\log n}{dK\xi^2})$$

$$\lesssim d^{-10} \tag{161}$$

Therefore,

$$
\begin{aligned}
\sup_{\boldsymbol{W}\in\mathbb{B}(\boldsymbol{W}^*,r)} \|\nabla\tilde{f}_n(\boldsymbol{W}) - \nabla\tilde{f}(\boldsymbol{W})\| &\leq C_{13}\cdot\sqrt{K\sum_{l=1}^L \lambda_l(\|\boldsymbol{\mu}_l\|_\infty+\sigma_l)^2}\sqrt{\frac{d\log n}{n}} + \frac{1}{n}\sum_{i=1}^n \|\nu_i\| \\
&\leq C_{13}\cdot\sqrt{K\sum_{l=1}^L \lambda_l(\|\boldsymbol{\mu}_l\|_\infty+\sigma_l)^2}\sqrt{\frac{d\log n}{n}} + \frac{1}{n}\sum_{i=1}^n \|\nu_i\|_F \\
&\leq C_{13}\cdot\sqrt{K\sum_{l=1}^L \lambda_l(\|\boldsymbol{\mu}_l\|_\infty+\sigma_l)^2}\sqrt{\frac{d\log n}{n}}(1+\xi)
\end{aligned}
\tag{162}
$$

## D.2 PROOF OF LEMMA 2

Following the proof of Theorem 2 in [Fu et al. (2020)], first we have the Taylor's expansion of $f_n(\widehat{\boldsymbol{W}}_n)$

$$
\begin{aligned}
f_n(\widehat{\boldsymbol{W}}_n) =& f_n(\boldsymbol{W}^*) + \left\langle \nabla\tilde{f}_n(\boldsymbol{W}^*), \mathrm{vec}(\widehat{\boldsymbol{W}}_n - \boldsymbol{W}^*)\right\rangle \\
&+ \frac{1}{2}\mathrm{vec}(\widehat{\boldsymbol{W}}_n - \boldsymbol{W}^*)\nabla^2 f_n(\boldsymbol{W}')\mathrm{vec}(\widehat{\boldsymbol{W}}_n - \boldsymbol{W}^*)
\end{aligned}
\tag{163}
$$

Here $\boldsymbol{W}'$ is on the straight line connecting $\boldsymbol{W}^*$ and $\widehat{\boldsymbol{W}}_n$. By the fact that $f_n(\widehat{\boldsymbol{W}}_n)\leq f_n(\boldsymbol{W}^*)$, we have

$$
\frac{1}{2}\mathrm{vec}(\widehat{\boldsymbol{W}}_n - \boldsymbol{W}^*)\nabla^2 f_n(\boldsymbol{W}')\mathrm{vec}(\widehat{\boldsymbol{W}}_n - \boldsymbol{W}^*) \leq \left|\nabla f_n(\boldsymbol{W}^*)^\top \mathrm{vec}(\widehat{\boldsymbol{W}}_n - \boldsymbol{W}^*)\right|
\tag{164}
$$

From Lemma 7 and Lemma 9, we have

$$
\begin{aligned}
&\frac{4}{K^2}\sum_{l=1}^L \lambda_l\frac{\sigma_l^2}{\eta\kappa^2}\rho\left(\frac{\boldsymbol{W}^{*\top}\boldsymbol{\mu}_l}{\sigma_l\delta_K(\boldsymbol{W}^*)}, \sigma_l\delta_K(\boldsymbol{W}^*)\right)\|\widehat{\boldsymbol{W}}_n - \boldsymbol{W}^*\|_F^2 \\
&\leq \frac{1}{2}\mathrm{vec}(\widehat{\boldsymbol{W}}_n - \boldsymbol{W}^*)\nabla^2 f_n(\boldsymbol{W}')\mathrm{vec}(\widehat{\boldsymbol{W}}_n - \boldsymbol{W}^*)
\end{aligned}
\tag{165}
$$

and

$$
\begin{aligned}
&\left|\nabla\tilde{f}_n(\boldsymbol{W}^*)^\top \mathrm{vec}(\widehat{\boldsymbol{W}}_n - \boldsymbol{W}^*)\right| \\
&\leq \|\nabla\tilde{f}_n(\boldsymbol{W}^*)\|\cdot\|\widehat{\boldsymbol{W}}_n - \boldsymbol{W}^*\|_F \\
&\leq (\|\nabla\tilde{f}_n(\boldsymbol{W}^*) - \nabla\tilde{f}(\boldsymbol{W}^*)\| + \|\nabla\tilde{f}(\boldsymbol{W}^*)\|)\cdot\|\widehat{\boldsymbol{W}}_n - \boldsymbol{W}^*\|_F \\
&\leq O\left(\sqrt{K\sum_{l=1}^L \lambda_l(\|\boldsymbol{\mu}_l\|_\infty+\sigma_l)^2}\sqrt{\frac{d\log n}{n}}(1+\xi)\right)\|\widehat{\boldsymbol{W}}_n - \boldsymbol{W}^*\|_F
\end{aligned}
\tag{166}
$$

The second to last step of (166) comes from the triangle inequality and the last step follows from the fact $\nabla f(\boldsymbol{W}^*) = 0$. Combining (164), (165) and (166), we have

$$
\|\widehat{\boldsymbol{W}}_n - \boldsymbol{W}^*\|_F \leq O\left(\frac{K^{\frac{5}{2}}\sqrt{\sum_{l=1}^L \lambda_l(\|\boldsymbol{\mu}_l\|_\infty+\sigma_l)^2}(1+\xi)}{\sum_{l=1}^L \lambda_l\frac{\sigma_l^2}{\eta\kappa^2}\rho\left(\frac{\boldsymbol{W}^{*\top}\boldsymbol{\mu}_l}{\sigma_l\delta_K(\boldsymbol{W}^*)}, \sigma_l\delta_K(\boldsymbol{W}^*)\right)}\sqrt{\frac{d\log n}{n}}\right)
\tag{167}
$$

Therefore, we have concluded that there indeed exists a critical point $\widehat{\boldsymbol{W}}$ in $\mathbb{B}(\boldsymbol{W}^*,r)$. Then we show the linear convergence of Algorithm 1 as below. By the update rule, we have

$$
\begin{aligned}
\boldsymbol{W}_{t+1} - \widehat{\boldsymbol{W}}_n &= \boldsymbol{W}_t - \eta_0\left(\nabla f_n(\boldsymbol{W}_t) + \frac{1}{n}\sum_{i=1}^n \nu_i\right) - (\widehat{\boldsymbol{W}}_n - \eta_0\nabla f_n(\widehat{\boldsymbol{W}}_n)) \\
&= \left(\boldsymbol{I} - \eta_0\int_0^1 \nabla^2 f_n(\boldsymbol{W}(\gamma))\right)(\boldsymbol{W}_t - \widehat{\boldsymbol{W}}_n) - \frac{\eta_0}{n}\sum_{i=1}^n \nu_i
\end{aligned}
\tag{168}
$$

where $\boldsymbol{W}(\gamma) = \gamma \widehat{\boldsymbol{W}}_n + (1-\gamma)\boldsymbol{W}_t$ for $\gamma \in (0,1)$. Since $\boldsymbol{W}(\gamma) \in \mathbb{B}(\boldsymbol{W}^*, r)$, by Lemma 1, we have

$$H_{\min} \cdot \boldsymbol{I} \preceq \nabla^2 f_n(\boldsymbol{W}(\gamma)) \leq H_{\max} \cdot \boldsymbol{I} \tag{169}$$

where $H_{\min} = \Omega\Big( \frac{1}{K^2} \sum_{l=1}^{L} \lambda_l \frac{\sigma_l^2}{\eta \kappa^2} \rho\big( \frac{\boldsymbol{W}^{*\top} \boldsymbol{\mu}_l}{\sigma_l \delta_K(\boldsymbol{W}^*)}, \sigma_l \delta_K(\boldsymbol{W}^*) \big) \Big)$, $H_{\max} = C_4 \cdot \sum_{l=1}^{L} \lambda_l(\|\boldsymbol{\mu}_l\|_\infty + \sigma_l)^2$. Therefore,

$$\|\boldsymbol{W}_{t+1} - \widehat{\boldsymbol{W}}_n\|_F = \|\boldsymbol{I} - \eta_0 \int_0^1 \nabla^2 f_n(\boldsymbol{W}(\gamma))\| \cdot \|\boldsymbol{W}_t - \widehat{\boldsymbol{W}}_n\|_F + \|\frac{\eta_0}{n} \sum_{i=1}^{n} \nu_i\|_F$$

$$\leq (1 - \eta_0 H_{\min})\|\boldsymbol{W}_t - \widehat{\boldsymbol{W}}_n\|_F + \|\frac{\eta_0}{n} \sum_{i=1}^{n} \nu_i\|_F \tag{170}$$

By setting $\eta_0 = \frac{1}{H_{\max}} = O\Big( \frac{1}{\sum_{l=1}^{L} \lambda_l(\|\boldsymbol{\mu}_l\|_\infty + \sigma_l)^2} \Big)$, we obtain

$$\|\widehat{\boldsymbol{W}}_{t+1} - \widehat{\boldsymbol{W}}_n\|_F \leq (1 - \frac{H_{\min}}{H_{\max}})\|\boldsymbol{W}_t - \widehat{\boldsymbol{W}}_n\|_F + \frac{\eta_0}{n} \sum_{i=1}^{n} \|\nu_i\|_F \tag{171}$$

Therefore, Algorithm 1 converges linearly to the local minimizer with an extra statistical error. By Hoeffding's inequality in (Vershynin, 2010) and Property 1, we have

$$\mathbb{P}\Big( \frac{1}{n} \sum_{i=1}^{n} \|\nu_i\|_F \geq \sqrt{\frac{dK \log n}{n}} \xi \Big) \lesssim \exp(-\frac{\xi^2 dK \log n}{dK \xi^2}) \lesssim d^{-10} \tag{172}$$

Therefore, with probability $1 - d^{-10}$ we can derive

$$\|\widehat{\boldsymbol{W}}_t - \widehat{\boldsymbol{W}}_n\|_F \leq (1 - \frac{H_{\min}}{H_{\max}})^t \|\boldsymbol{W}_0 - \widehat{\boldsymbol{W}}_n\|_F + \frac{H_{\max} \eta_0}{H_{\min}} \sqrt{\frac{dK \log n}{n}} \xi \tag{173}$$

## E    PROOF OF LEMMA 3

We need Lemma 10 to Lemma 14, which are stated in Section E.1, for the proof of Lemma 3. Section E.2 summarizes the proof of Lemma 3. The proofs of Lemma 10 to Lemma 12 are provided in Section E.3 to Section E.5. Lemma 13 and Lemma 14 are cited from (Zhong et al., 2017b). Although (Zhong et al., 2017b) considers the standard Gaussian distribution, the proofs of Lemma 13 and 14 hold for any data distribution. Therefore, these two lemmas can be applied here directly. The tensor initialization in (Zhong et al., 2017b) only holds for the standard Gaussian distribution. We exploit a more general definition of tensors from (Janzamin et al. (2014)) for the tensor initialization in our algorithm. We also develop new error bounds for the initialization.

### E.1    USEFUL LEMMAS IN THE PROOF

**Lemma 10** *Let $\boldsymbol{P}_2$ follow Definition 1. Let $S$ be a set of i.i.d. samples generated from the mixed Gaussian distribution $\sum_{l=1}^{L} \lambda_l \mathcal{N}(\boldsymbol{\mu}_l, \sigma_l^2 \boldsymbol{I})$. Let $\widehat{\boldsymbol{P}}_2$ be the empirical version of $\boldsymbol{P}_2$ using data set $S$. Then with probability at least $1 - 2n^{-\Omega(\delta_1^4 d)}$, we have*

$$\|\boldsymbol{P}_2 - \widehat{\boldsymbol{P}}_2\| \lesssim \sqrt{\frac{d \log n}{n}} \cdot \delta_1^2 \cdot \tau^6 \sqrt{D_2(\boldsymbol{\lambda}, \boldsymbol{M}, \boldsymbol{\sigma}) D_4(\boldsymbol{\lambda}, \boldsymbol{M}, \boldsymbol{\sigma})} \tag{174}$$

**Lemma 11** *Let $\boldsymbol{U} \in \mathbb{E}^{d \times K}$ be the orthogonal column span of $\boldsymbol{W}^*$. Let $\boldsymbol{\alpha}$ be a fixed unit vector and $\widehat{\boldsymbol{U}} \in \mathbb{R}^{d \times K}$ denote an orthogonal matrix satisfying $\|\boldsymbol{U}\boldsymbol{U}^\top - \widehat{\boldsymbol{U}}\widehat{\boldsymbol{U}}^\top\| \leq \frac{1}{4}$. Define $\boldsymbol{R}_3 = \boldsymbol{M}_3(\widehat{\boldsymbol{U}}, \widehat{\boldsymbol{U}}, \widehat{\boldsymbol{U}})$, where $\boldsymbol{M}_3$ is defined in Definition 1. Let $\widehat{\boldsymbol{R}}_3$ be the empirical version of $\boldsymbol{R}_3$ using data set $S$, where each sample of $S$ is i.i.d. sampled from the mixed Gaussian distribution $\sum_{l=1}^{L} \lambda_l \mathcal{N}(\boldsymbol{\mu}_l, \sigma_l^2 \boldsymbol{I})$. Then with probability at least $1 - n^{-\Omega(\delta^4)}$, we have*

$$\|\widehat{\boldsymbol{R}}_3 - \boldsymbol{R}_3\| \lesssim \delta_1^2 \cdot \big( \tau^6 \sqrt{D_6(\boldsymbol{\lambda}, \boldsymbol{M}, \boldsymbol{\sigma})} \big) \cdot \sqrt{\frac{\log n}{n}} \tag{175}$$

**Lemma 12** *Let $\widehat{\boldsymbol{M}}_1$ be the empirical version of $\boldsymbol{M}_1$ using dataset $S$. Then with probability at least $1 - 2n^{-\Omega(d)}$, we have*

$$||\widehat{\boldsymbol{M}}_1 - \boldsymbol{M}_1|| \lesssim \left(\tau^2\sqrt{D_2(\boldsymbol{\lambda}, \boldsymbol{M}, \boldsymbol{\sigma})}\right) \cdot \sqrt{\frac{d\log n}{n}} \tag{176}$$

**Lemma 13** *((Zhong et al., 2017b), Lemma E.6) Let $\boldsymbol{P}_2$ be defined in Definition 1 and $\widehat{\boldsymbol{P}}_2$ be its empirical version. Let $\boldsymbol{U} \in \mathbb{R}^{d \times K}$ be the column span of $\boldsymbol{W}^*$. Assume $||\boldsymbol{P}_2 - \widehat{\boldsymbol{P}}_2|| \leq \frac{\delta_K(\boldsymbol{P}_2)}{10}$. Then after $T = O(\log(\frac{1}{\epsilon}))$ iterations, the output of the Tensor Initialization Method 3, $\widehat{\boldsymbol{U}}$ will satisfy*

$$||\widehat{\boldsymbol{U}}\widehat{\boldsymbol{U}}^\top - \boldsymbol{U}\boldsymbol{U}^\top|| \lesssim \frac{||\widehat{\boldsymbol{P}}_2 - \boldsymbol{P}_2||}{\delta_K(\boldsymbol{P}_2)} + \epsilon \tag{177}$$

*which implies*

$$||(\boldsymbol{I} - \widehat{\boldsymbol{U}}\widehat{\boldsymbol{U}}^\top)\boldsymbol{w}_i^*|| \lesssim \left(\frac{||\boldsymbol{P}_2 - \widehat{\boldsymbol{P}}_2||}{\delta_K(\boldsymbol{P}_2)} + \epsilon\right)||\boldsymbol{w}_i^*|| \tag{178}$$

**Lemma 14** *((Zhong et al., 2017b), Lemma E.13) Let $\boldsymbol{U} \in \mathbb{R}^{d \times K}$ be the orthogonal column span of $\boldsymbol{W}^*$. Let $\widehat{\boldsymbol{U}} \in \mathbb{R}^{d \times K}$ be an orthogonal matrix such that $||\boldsymbol{U}\boldsymbol{U}^\top - \widehat{\boldsymbol{U}}\widehat{\boldsymbol{U}}^\top|| \lesssim \gamma_1 \lesssim \frac{1}{\kappa^2\sqrt{K}}$. For each $i \in [K]$, let $\widehat{\boldsymbol{v}}_i$ denote the vector satisfying $||\widehat{\boldsymbol{v}}_i - \widehat{\boldsymbol{U}}^\top\bar{\boldsymbol{w}}_i^*|| \leq \gamma_2 \lesssim \frac{1}{\kappa^2\sqrt{K}}$. Let $\boldsymbol{M}_1$ be defined in Lemma 12 and $\widehat{\boldsymbol{M}}_1$ be its empirical version. If $||\boldsymbol{M}_1 - \widehat{\boldsymbol{M}}_1|| \leq \gamma_3||\boldsymbol{M}_1|| \lesssim \frac{1}{4}||\boldsymbol{M}_1||$, then we have*

$$\left|||\boldsymbol{w}_i^*|| - \widehat{\alpha}_i\right| \leq (\kappa^4 K^{\frac{3}{2}}(\gamma_1 + \gamma_2) + \kappa^2 K^{\frac{1}{2}}\gamma_3)||\boldsymbol{w}_i^*|| \tag{179}$$

## E.2 PROOF OF LEMMA 3

$$\begin{aligned}
||\boldsymbol{w}_j^* - \widehat{\alpha}_j\widehat{\boldsymbol{U}}\widehat{\boldsymbol{v}}_j|| &= \left|\left|\boldsymbol{w}_j^* - ||\boldsymbol{w}_j^*||\widehat{\boldsymbol{U}}\widehat{\boldsymbol{v}}_j + ||\boldsymbol{w}_j^*||\widehat{\boldsymbol{U}}\widehat{\boldsymbol{v}}_j - \widehat{\alpha}_j\widehat{\boldsymbol{U}}\widehat{\boldsymbol{v}}_j\right|\right| \\
&\leq \left|\left|\boldsymbol{w}_j^* - ||\boldsymbol{w}_j^*||\widehat{\boldsymbol{U}}\widehat{\boldsymbol{v}}_j\right|\right| + \left|\left|||\boldsymbol{w}_j^*||\widehat{\boldsymbol{U}}\widehat{\boldsymbol{v}}_j - \widehat{\alpha}_j\widehat{\boldsymbol{U}}\widehat{\boldsymbol{v}}_j\right|\right| \\
&\leq ||\boldsymbol{w}_j^*||\left|\left|\bar{\boldsymbol{w}}_j^* - \widehat{\boldsymbol{U}}\widehat{\boldsymbol{v}}_j\right|\right| + \left|\left|||\boldsymbol{w}_j^*|| - \widehat{\alpha}_j\right|\right|||\widehat{\boldsymbol{U}}\widehat{\boldsymbol{v}}_j|| \\
&\leq ||\boldsymbol{w}_j^*||\left|\left|\bar{\boldsymbol{w}}_j^* - \widehat{\boldsymbol{U}}\widehat{\boldsymbol{U}}^\top\bar{\boldsymbol{w}}_j^* + \widehat{\boldsymbol{U}}\widehat{\boldsymbol{U}}^\top\bar{\boldsymbol{w}}_j^* - \widehat{\boldsymbol{U}}\widehat{\boldsymbol{v}}_j\right|\right| + \left|\left|||\boldsymbol{w}_j^*|| - \widehat{\alpha}_j\right|\right|||\widehat{\boldsymbol{U}}\widehat{\boldsymbol{v}}_j|| \\
&\leq \delta_1(\boldsymbol{W}^*)\left(\left|\left|\bar{\boldsymbol{w}}_j^* - \widehat{\boldsymbol{U}}\widehat{\boldsymbol{U}}^\top\bar{\boldsymbol{w}}_j^*\right|\right| + \left|\left|\widehat{\boldsymbol{U}}^\top\bar{\boldsymbol{w}}_j^* - \widehat{\boldsymbol{v}}_j\right|\right|\right) + \left|\left|||\boldsymbol{w}_j^*|| - \widehat{\alpha}_j\right|\right|
\end{aligned} \tag{180}$$

By Lemma 10, Lemma 13 and $\delta_K(\boldsymbol{P}_2) \lesssim \delta_K^2$, we have

$$\begin{aligned}
\left|\left|\bar{\boldsymbol{w}}_j^* - \widehat{\boldsymbol{U}}\widehat{\boldsymbol{U}}^\top\bar{\boldsymbol{w}}_j^*\right|\right| &\lesssim \frac{||\boldsymbol{P}_2 - \widehat{\boldsymbol{P}}_2||}{\delta_K(\boldsymbol{P}_2)} \lesssim \sqrt{\frac{d\log n}{n}} \cdot \frac{\delta_1^2}{\delta_K^2} \cdot \tau^6\sqrt{D_2(\boldsymbol{\lambda}, \boldsymbol{M}, \boldsymbol{\sigma})D_4(\boldsymbol{\lambda}, \boldsymbol{M}, \boldsymbol{\sigma})} \\
&= \sqrt{\frac{d\log n}{n}} \cdot \kappa^2 \cdot \tau^6\sqrt{D_2(\boldsymbol{\lambda}, \boldsymbol{M}, \boldsymbol{\sigma})D_4(\boldsymbol{\lambda}, \boldsymbol{M}, \boldsymbol{\sigma})}
\end{aligned} \tag{181}$$

Moreover, we have

$$\left|\left|\widehat{\boldsymbol{U}}^\top\bar{\boldsymbol{w}}_j^* - \widehat{\boldsymbol{v}}_j\right|\right| \leq \frac{K^{\frac{3}{2}}}{\delta_K^2(\boldsymbol{W}^*)}||\boldsymbol{R}_3 - \widehat{\boldsymbol{R}}_3|| \lesssim \kappa^2 \cdot \left(\tau^6\sqrt{D_6(\boldsymbol{\lambda}, \boldsymbol{M}, \boldsymbol{\sigma})}\right) \cdot \sqrt{\frac{K^3\log n}{n}} \tag{182}$$

in which the first step is by Theorem 3 in [Kuleshov et al. (2015)] and the second step is by Lemma 11. By Lemma 14, we have

$$\left|\left|||\boldsymbol{w}_j^*|| - \widehat{\alpha}_j\right|\right| \leq (\kappa^4 K^{\frac{3}{2}}(\gamma_1 + \gamma_2) + \kappa^2 K^{\frac{1}{2}}\gamma_3)||\boldsymbol{W}^*|| \tag{183}$$

Therefore, taking the union bound of failure probabilities in Lemmas 10, 11 and 12 and by $D_2(\boldsymbol{\lambda}, \boldsymbol{M}, \boldsymbol{\sigma})D_4(\boldsymbol{\lambda}, \boldsymbol{M}, \boldsymbol{\sigma}) \leq D_6(\boldsymbol{\lambda}, \boldsymbol{M}, \boldsymbol{\sigma})$ from Property 7, we have that if the sample size $n \geq \kappa^8 K^4 \tau^{12} D_6(\boldsymbol{\lambda}, \boldsymbol{M}, \boldsymbol{\sigma}) \cdot d\log^2 d$, then the output $\boldsymbol{W}_0 \in \mathbb{R}^{d \times K}$ satisfies

$$||\boldsymbol{W}_0 - \boldsymbol{W}^*|| \lesssim \kappa^6 K^3 \cdot \tau^6\sqrt{D_6(\boldsymbol{\lambda}, \boldsymbol{M}, \boldsymbol{\sigma})}\sqrt{\frac{d\log n}{n}}||\boldsymbol{W}^*|| \tag{184}$$

with probability at least $1 - n^{-\Omega(\delta_1^4)}$

### E.3 PROOF OF LEMMA 10

From Assumption 1, if the Gaussian Mixture Model is a symmetric probability distribution defined in (8), then $\boldsymbol{P}_2 = \boldsymbol{M}_3(\boldsymbol{I}, \boldsymbol{I}, \boldsymbol{\alpha})$. Therefore, by Definition 1, we have

$$
\begin{aligned}
||\widehat{\boldsymbol{M}}_3(\boldsymbol{I},\boldsymbol{I},\boldsymbol{\alpha}) - \boldsymbol{M}_3(\boldsymbol{I},\boldsymbol{I},\boldsymbol{\alpha})|| = &\Big|\Big| \frac{1}{n}\sum_{i=1}^n \Big[ y_i \cdot p(\boldsymbol{x})^{-1} \sum_{l=1}^L \lambda_l (2\pi\sigma_l)^{-\frac{d}{2}} \exp(-\frac{||\boldsymbol{x}-\boldsymbol{\mu}_l||^2}{2\sigma_l^2}) \\
& \Big( (\frac{\boldsymbol{x}-\boldsymbol{\mu}_l}{\sigma_l^2})^{\otimes 3} - (\frac{\boldsymbol{x}-\boldsymbol{\mu}_l}{\sigma_l^2})\widetilde{\otimes}\sigma_l^{-2}\boldsymbol{I}\Big)\Big](\boldsymbol{I},\boldsymbol{I},\boldsymbol{\alpha}) \\
& -\mathbb{E}\Big[ y \cdot p(\boldsymbol{x})^{-1} \sum_{l=1}^L \lambda_l (2\pi\sigma_l)^{-\frac{d}{2}} \exp(-\frac{||\boldsymbol{x}-\boldsymbol{\mu}_l||^2}{2\sigma_l^2}) \\
& \Big( (\frac{\boldsymbol{x}-\boldsymbol{\mu}_l}{\sigma_l^2})^{\otimes 3} - (\frac{\boldsymbol{x}-\boldsymbol{\mu}_l}{\sigma_l^2})\widetilde{\otimes}\sigma_l^{-2}\boldsymbol{I}\Big)\Big](\boldsymbol{I},\boldsymbol{I},\boldsymbol{\alpha})\Big|\Big|
\end{aligned}
\tag{185}
$$

Following (Zhong et al., 2017b), $\widetilde{\otimes}$ is defined such that for any $\boldsymbol{v} \in \mathbb{R}^{d_1}$ and $\boldsymbol{Z} \in \mathbb{R}^{d_1 \times d_2}$,

$$
\boldsymbol{v}\widetilde{\otimes}\boldsymbol{Z} = \sum_{i=1}^{d_2} (\boldsymbol{v}\otimes\boldsymbol{z}_i\otimes\boldsymbol{z}_i + \boldsymbol{z}_i\otimes\boldsymbol{v}\otimes\boldsymbol{z}_i + \boldsymbol{z}_i\otimes\boldsymbol{z}_i\otimes\boldsymbol{v}),
\tag{186}
$$

where $\boldsymbol{z}_i$ is the $i$-th column of $\boldsymbol{Z}$. By Definition 1, we have

$$
\begin{aligned}
&\Big|\Big| \Big[ y \cdot p(\boldsymbol{x})^{-1} \sum_{l=1}^L \lambda_l (2\pi\sigma_l)^{-\frac{d}{2}} \exp(-\frac{||\boldsymbol{x}-\boldsymbol{\mu}_l||^2}{2\sigma_l^2})\Big( (\frac{\boldsymbol{x}-\boldsymbol{\mu}_l}{\sigma_l^2})^{\otimes 3} - (\frac{\boldsymbol{x}-\boldsymbol{\mu}_l}{\sigma_l^2})\widetilde{\otimes}\sigma_l^{-2}\boldsymbol{I}\Big)\Big](\boldsymbol{I},\boldsymbol{I},\boldsymbol{\alpha})\Big|\Big| \\
&\lesssim \Big|\Big| \frac{\sum_{l=1}^L \lambda_l (2\pi\sigma_l^2)^{-\frac{d}{2}} \exp(-\frac{||\boldsymbol{x}-\boldsymbol{\mu}_l||^2}{2\sigma_l^2}) \cdot (\frac{\boldsymbol{x}-\boldsymbol{\mu}_l}{\sigma_l^2})^{\otimes 2}(\boldsymbol{\alpha}^\top(\frac{\boldsymbol{x}-\boldsymbol{\mu}_l}{\sigma_l^2}))}{\sum_{l=1}^L \lambda_l (2\pi\sigma_l^2)^{-\frac{d}{2}} \exp(-\frac{||\boldsymbol{x}-\boldsymbol{\mu}_l||^2}{2\sigma_l^2})} \Big|\Big| \\
&\lesssim ||\sigma_{\min}^{-6}(\boldsymbol{x}_i^\top\boldsymbol{\alpha})\boldsymbol{x}_i\boldsymbol{x}_i^\top||
\end{aligned}
\tag{187}
$$

The first step of (187) is because $(\frac{\boldsymbol{x}-\boldsymbol{\mu}_l}{\sigma_l^2})^{\otimes 2}(\boldsymbol{\alpha}^\top(\frac{\boldsymbol{x}-\boldsymbol{\mu}_l}{\sigma_l^2}))$ is the dominant term of the entire expression, and $y \leq 1$. The second step is because the expression can be considered as a normalized weighted summation of $(\frac{\boldsymbol{x}-\boldsymbol{\mu}_l}{\sigma_l^2})^{\otimes 2}(\boldsymbol{\alpha}^\top(\frac{\boldsymbol{x}-\boldsymbol{\mu}_l}{\sigma_l^2}))$ and $(\boldsymbol{x}_i^\top\boldsymbol{\alpha})\boldsymbol{x}_i\boldsymbol{x}_i^\top$ is its dominant term. Define $S_m(\boldsymbol{x}) = (-1)^m \frac{\nabla_{\boldsymbol{x}}^m p(\boldsymbol{x})}{p(\boldsymbol{x})}$, where $p(\boldsymbol{x})$ is the probability density function of the random variable $\boldsymbol{x}$. From Definition 1, we can verify that

$$
\boldsymbol{M}_j = \mathbb{E}[y \cdot S_m(\boldsymbol{x})] \quad j \in \{1,2,3\}
\tag{188}
$$

Then define $Gp_i = \big\langle \boldsymbol{v}, ([y_i \cdot S_3(\boldsymbol{x}_i)](\boldsymbol{I}_d,\boldsymbol{I}_d,\boldsymbol{\alpha}) - \mathbb{E}[[y_i \cdot S_3(\boldsymbol{x}_i)](\boldsymbol{I}_d,\boldsymbol{I}_d,\boldsymbol{\alpha})]\boldsymbol{v})\big\rangle$, where $||\boldsymbol{v}|| = 1$, then $\mathbb{E}[Gp_i] = 0$. Similar to the proof of (131), (132) and (133) in Lemma 8, we have

$$
|Gp_i|^p \lesssim \big| \sigma_{\min}^{-6}(\boldsymbol{x}_i^\top\boldsymbol{\alpha})(\boldsymbol{x}_i^\top\boldsymbol{v})^2 + \mathbb{E}_{\boldsymbol{x}\sim\sum_{l=1}^L \mathcal{N}(\boldsymbol{\mu}_l,\sigma_l^2\boldsymbol{I}_d)}[\sigma_{\min}^{-6}(\boldsymbol{x}_i^\top\boldsymbol{\alpha})(\boldsymbol{x}_i^\top\boldsymbol{v})^2] \big|^p
\tag{189}
$$

$$
\begin{aligned}
\mathbb{E}[|Gp_i|^p] &\lesssim \big( \mathbb{E}_{\boldsymbol{x}\sim\sum_{l=1}^L \mathcal{N}(\boldsymbol{\mu}_l,\sigma_l^2\boldsymbol{I}_d)}[\sigma_{\min}^{-6}(\boldsymbol{x}_i^\top\boldsymbol{\alpha})(\boldsymbol{x}_i^\top\boldsymbol{v})^2] \big)^p \\
&\leq \sigma_{\min}^{-6p} \mathbb{E}_{\boldsymbol{x}\sim\sum_{l=1}^L \mathcal{N}(\boldsymbol{\mu}_l,\sigma_l^2\boldsymbol{I}_d)}[(\boldsymbol{x}^\top\boldsymbol{\alpha})^2]^{\frac{p}{2}} \mathbb{E}_{\boldsymbol{x}\sim\sum_{l=1}^L \mathcal{N}(\boldsymbol{\mu}_l,\sigma_l^2\boldsymbol{I}_d)}[(\boldsymbol{x}^\top\boldsymbol{v})^4]^{\frac{p}{2}} \\
&\leq \tau^{6p}\sqrt{D_2(\boldsymbol{\lambda},\boldsymbol{M},\boldsymbol{\sigma})D_4(\boldsymbol{\lambda},\boldsymbol{M},\boldsymbol{\sigma})}^p
\end{aligned}
\tag{190}
$$

$$
\begin{aligned}
\mathbb{E}[\exp(\theta Gp_i)] &\lesssim 1 + \sum_{p=2}^\infty \frac{\theta^p \mathbb{E}[|Gp_i|^p]}{p!} \lesssim 1 + \sum_{p=2}^\infty \frac{|e\theta|^p \tau^{6p}(D_2(\boldsymbol{\lambda},\boldsymbol{M},\boldsymbol{\sigma})D_4(\boldsymbol{\lambda},\boldsymbol{M},\boldsymbol{\sigma}))^{\frac{p}{2}}}{p^p} \\
&\lesssim 1 + \theta^2\tau^{12}D_2(\boldsymbol{\lambda},\boldsymbol{M},\boldsymbol{\sigma})D_4(\boldsymbol{\lambda},\boldsymbol{M},\boldsymbol{\sigma})
\end{aligned}
\tag{191}
$$

Hence, similar to the derivation of (135), we have

$$\mathbb{P}\Big(\frac{1}{n}\sum_{i=1}^{n} Gp_i \ge t\Big) \le \exp\Big(-n\theta t + C_{16}n\theta^2\big(\tau^6\sqrt{D_2(\boldsymbol{\lambda},\boldsymbol{M},\boldsymbol{\sigma})D_4(\boldsymbol{\lambda},\boldsymbol{M},\boldsymbol{\sigma})}\big)^2\Big) \qquad (192)$$

for some constant $C_{16} > 0$. Let $\theta = \frac{t}{2C_{16}\big(\tau^6\sqrt{D_2(\boldsymbol{\lambda},\boldsymbol{M},\boldsymbol{\sigma})D_4(\boldsymbol{\lambda},\boldsymbol{M},\boldsymbol{\sigma})}\big)^2}$ and $t = \delta_1^2 \cdot$ $\big(\tau^6\sqrt{D_2(\boldsymbol{\lambda},\boldsymbol{M},\boldsymbol{\sigma})D_4(\boldsymbol{\lambda},\boldsymbol{M},\boldsymbol{\sigma})}\big) \cdot \sqrt{\frac{d\log n}{n}}$, then we have

$$||\widehat{\boldsymbol{M}}_3(\boldsymbol{I}_d,\boldsymbol{I}_d,\boldsymbol{\alpha}) - \boldsymbol{M}_3(\boldsymbol{I}_d,\boldsymbol{I}_d,\boldsymbol{\alpha})|| \le \delta_1^2 \cdot \big(\tau^6\sqrt{D_2(\boldsymbol{\lambda},\boldsymbol{M},\boldsymbol{\sigma})D_4(\boldsymbol{\lambda},\boldsymbol{M},\boldsymbol{\sigma})}\big) \cdot \sqrt{\frac{d\log n}{n}} \quad (193)$$

with probability at least $1 - 2n^{-\Omega(\delta_1^4 d)}$.

If the Gaussian Mixture Model is not a symmetric distribution which is defined in (8), then $\boldsymbol{P}_2 = \boldsymbol{M}_2$. We would have a similar result as follows.

$$||\widehat{\boldsymbol{M}}_2 - \boldsymbol{M}_2|| = \Big|\Big|\frac{1}{n}\sum_{i=1}^{n}[y_i \cdot S_2(\boldsymbol{x})] - \mathbb{E}[y \cdot S_2(\boldsymbol{x})]\Big|\Big| \qquad (194)$$

$$||y_i \cdot S_2(\boldsymbol{x}_i)|| \lesssim ||\sigma_{\min}^{-4}\frac{1}{K}\sum_{j=1}^{K}\phi(\boldsymbol{w}_j^{*\top}\boldsymbol{x}_i)\boldsymbol{x}_i\boldsymbol{x}_i^\top|| \qquad (195)$$

Then define $Gp_i' = \big\langle \boldsymbol{v}, ([y_i \cdot S_2(\boldsymbol{x}_i)] - \mathbb{E}\big[y_i \cdot S_2(\boldsymbol{x}_i)\big]\boldsymbol{v})\big\rangle$, where $||\boldsymbol{v}|| = 1$, then $\mathbb{E}[Gp_i'] = 0$. Similar to the proof of (131), (132) and (133) in Lemma 8, we have

$$|Gp_i'|^p \lesssim \big|\sigma_{\min}^{-4}(\boldsymbol{x}_i^\top\boldsymbol{v})^2 + \mathbb{E}_{\boldsymbol{x}\sim\sum_{l=1}^{L}\mathcal{N}(\boldsymbol{\mu}_l,\sigma_l^2\boldsymbol{I}_d)}[\sigma_{\min}^{-4}(\boldsymbol{x}_i^\top\boldsymbol{v})^2]\big|^p \qquad (196)$$

$$\mathbb{E}[|Gp_i'|^p] \lesssim \big(\mathbb{E}_{\boldsymbol{x}\sim\sum_{l=1}^{L}\mathcal{N}(\boldsymbol{\mu}_l,\sigma_l^2\boldsymbol{I}_d)}[\sigma_{\min}^{-4}(\boldsymbol{x}_i^\top\boldsymbol{v})^2]\big)^p \le \tau^{4p}D_2(\boldsymbol{\lambda},\boldsymbol{M},\boldsymbol{\sigma})^p \qquad (197)$$

$$\mathbb{E}[\exp(\theta Gp_i')] \lesssim 1 + \sum_{p=2}^{\infty}\frac{\theta^p\mathbb{E}[|Gp_i|^p]}{p!} \lesssim 1 + \sum_{p=2}^{\infty}\frac{|e\theta|^p\tau^{4p}D_2(\boldsymbol{\lambda},\boldsymbol{M},\boldsymbol{\sigma})^p}{p^p}$$
$$\lesssim 1 + \theta^2\tau^8 D_2(\boldsymbol{\lambda},\boldsymbol{M},\boldsymbol{\sigma})^2 \qquad (198)$$

Hence, similar to the derivation of (135), we have

$$\mathbb{P}\Big(\frac{1}{n}\sum_{i=1}^{n} Gp_i \ge t\Big) \le \exp\Big(-n\theta t + C_{17}n\theta^2\big(\tau^4 D_2(\boldsymbol{\lambda},\boldsymbol{M},\boldsymbol{\sigma})\big)^2\Big) \qquad (199)$$

for some constant $C_{17} > 0$. Let $\theta = \frac{t}{2C_{17}\big(\tau^4 D_2(\boldsymbol{\lambda},\boldsymbol{M},\boldsymbol{\sigma})\big)^2}$ and $t = \delta_1^2 \cdot \big(\tau^4 D_2(\boldsymbol{\lambda},\boldsymbol{M},\boldsymbol{\sigma})\big) \cdot \sqrt{\frac{d\log n}{n}}$, then we have

$$||\widehat{\boldsymbol{M}}_2 - \boldsymbol{M}_2|| \lesssim \delta_1^2 \cdot \tau^4 D_2(\boldsymbol{\lambda},\boldsymbol{M},\boldsymbol{\sigma}) \cdot \sqrt{\frac{d\log n}{n}} \qquad (200)$$

with probability at least $1 - 2n^{-\Omega(\delta_1^4 d)}$.

To sum up, from (193) and (200) we have

$$||\boldsymbol{P}_2 - \widehat{\boldsymbol{P}}_2|| \lesssim \sqrt{\frac{d\log n}{n}} \cdot \delta_1^2 \cdot \max\{\tau^4 D_2(\boldsymbol{\lambda},\boldsymbol{M},\boldsymbol{\sigma})), \tau^6\sqrt{D_2(\boldsymbol{\lambda},\boldsymbol{M},\boldsymbol{\sigma})D_4(\boldsymbol{\lambda},\boldsymbol{M},\boldsymbol{\sigma})}\}$$
$$\lesssim \sqrt{\frac{d\log n}{n}} \cdot \delta_1^2 \cdot \tau^6\sqrt{D_2(\boldsymbol{\lambda},\boldsymbol{M},\boldsymbol{\sigma})D_4(\boldsymbol{\lambda},\boldsymbol{M},\boldsymbol{\sigma})} \qquad (201)$$

with probability at least $1 - 2n^{-\Omega(\delta_1^4 d)}$.

### E.4 PROOF OF LEMMA 11

We consider each component of $y = \frac{1}{K}\sum_{i=1}^{K}\phi(\boldsymbol{w}_i^{*\top}\boldsymbol{x})$.
Define $\boldsymbol{T}_i(\boldsymbol{x}) : \mathbb{R}^d \to \mathbb{R}^{K\times K\times K}$ such that

$$\boldsymbol{T}_i(\boldsymbol{x}) = [\phi(\boldsymbol{w}_i^{*\top}\boldsymbol{x}) \cdot S_3(\boldsymbol{x})](\widehat{\boldsymbol{U}},\widehat{\boldsymbol{U}},\widehat{\boldsymbol{U}}) \tag{202}$$

We flatten $\boldsymbol{T}_i(\boldsymbol{x})$ : $\mathbb{R}^d \to \mathbb{R}^{K\times K\times K}$ along the first dimension to obtain function $\boldsymbol{B}_i(\boldsymbol{x})$ : $\mathbb{R}^d \to \mathbb{R}^{K\times K^2}$. Similar to the derivation of the last step of Lemma E.8 in (Zhong et al., 2017b), we can obtain $\|\boldsymbol{T}_i(\boldsymbol{x})\| \le \|\boldsymbol{B}_i(\boldsymbol{x})\|$. By (185), we have

$$\|\boldsymbol{B}_i(\boldsymbol{x})\| \lesssim \sigma_{\min}^{-6}\frac{1}{K}\sum_{j=1}^{K}\phi(\boldsymbol{w}_j^{*\top}\boldsymbol{x}_i)(\widehat{\boldsymbol{U}}^{\top}\boldsymbol{x})^3 \tag{203}$$

Define $Gr_i = \langle\boldsymbol{v},\boldsymbol{B}_i(\boldsymbol{x}_i)) - \mathbb{E}[\boldsymbol{B}_i(\boldsymbol{x}_i)]\boldsymbol{v})\rangle$, where $\|\boldsymbol{v}\| = 1$, so $\mathbb{E}[Gr_i] = 0$. Similar to the proof of (131), (132) and (133) in Lemma 8, we have

$$|Gr_i|^p \lesssim \left|\sigma_{\min}^{-6}(\boldsymbol{v}^{\top}\widehat{\boldsymbol{U}}^{\top}\boldsymbol{x})^3 + \mathbb{E}_{\boldsymbol{x}\sim\sum_{l=1}^{L}\mathcal{N}(\boldsymbol{\mu}_l,\sigma_l^2\boldsymbol{I}_d)}[\sigma_{\min}^{-6}(\boldsymbol{v}^{\top}\widehat{\boldsymbol{U}}^{\top}\boldsymbol{x})^3]\right|^p \tag{204}$$

$$\mathbb{E}[|Gr_i|^p] \lesssim \left(\mathbb{E}_{\boldsymbol{x}\sim\sum_{l=1}^{L}\mathcal{N}(\boldsymbol{\mu}_l,\sigma_l^2\boldsymbol{I}_d)}[\sigma_{\min}^{-6}(\boldsymbol{v}^{\top}\widehat{\boldsymbol{U}}^{\top}\boldsymbol{x})^3]\right)^p \lesssim \tau^{6p}\sqrt{D_6(\boldsymbol{\lambda},\boldsymbol{M},\boldsymbol{\sigma})}^p \tag{205}$$

$$\mathbb{E}[\exp(\theta Gr_i)] \lesssim 1 + \sum_{p=2}^{\infty}\frac{\theta^p\mathbb{E}[|Gr_i|^p]}{p!} \lesssim 1 + \sum_{p=2}^{\infty}\frac{|e\theta|^p\tau^{6p}D_6(\boldsymbol{\lambda},\boldsymbol{M},\boldsymbol{\sigma})^{\frac{p}{2}}}{p^p}$$
$$\le 1 + \theta^2(\tau^{12}\sqrt{D_6(\boldsymbol{\lambda},\boldsymbol{M},\boldsymbol{\sigma})})^2 \tag{206}$$

Hence, similar to the derivation of (135), we have

$$\mathbb{P}\Big(\frac{1}{n}\sum_{i=1}^{n}Gr_i \ge t\Big) \le \exp\Big(-n\theta t + C_{18}\theta^2\big(\tau^6\sqrt{D_6(\boldsymbol{\lambda},\boldsymbol{M},\boldsymbol{\sigma})}\big)^2\Big) \tag{207}$$

for some constant $C_{18} > 0$. Let $\theta = \frac{t}{C_{18}\big(\tau^6\sqrt{D_6(\boldsymbol{\lambda},\boldsymbol{M},\boldsymbol{\sigma})}\big)^2}$ and $t = \delta_1^2 \cdot \big(\tau^6\sqrt{D_6(\boldsymbol{\lambda},\boldsymbol{M},\boldsymbol{\sigma})}\big) \cdot \sqrt{\frac{\log n}{n}}$, then we have

$$\|\widehat{\boldsymbol{R}}_3 - \boldsymbol{R}_3\| \lesssim \delta_1^2 \cdot \big(\tau^6\sqrt{D_6(\boldsymbol{\lambda},\boldsymbol{M},\boldsymbol{\sigma})}\big) \cdot \sqrt{\frac{\log n}{n}} \tag{208}$$

with probability at least $1 - 2n^{-\Omega(\delta_1^4)}$.

### E.5 PROOF OF LEMMA 12

From the Definition 1, we have

$$\|\widehat{\boldsymbol{M}}_1 - \boldsymbol{M}_1\| = \Big\|\frac{1}{n}\sum_{i=1}^{n}[y_i \cdot S_1(\boldsymbol{x})] - \mathbb{E}[y \cdot S_1(\boldsymbol{x})]\Big\|. \tag{209}$$

Based on Definition 1,

$$\Big\|[y_i \cdot S_1(\boldsymbol{x}_i)]\Big\| \lesssim \Big\|\frac{\sum_{l=1}^{L}\lambda_l(2\pi\sigma_l^2)^{-\frac{d}{2}}\exp(-\frac{\|\boldsymbol{x}-\boldsymbol{\mu}_l\|^2}{2\sigma_l^2})\cdot(\frac{\boldsymbol{x}-\boldsymbol{\mu}_l}{\sigma_l^2})}{\sum_{l=1}^{L}\lambda_l(2\pi\sigma_l^2)^{-\frac{d}{2}}\exp(-\frac{\|\boldsymbol{x}-\boldsymbol{\mu}_l\|^2}{2\sigma_l^2})}\Big\| \lesssim \Big\|\sigma_{\min}^{-2}\frac{1}{K}\sum_{j=1}^{K}\phi(\boldsymbol{w}_j^{*\top}\boldsymbol{x}_i)\boldsymbol{x}_i\Big\| \tag{210}$$

Define $Gq_i = \big\langle\boldsymbol{v},([y_i \cdot S_1(\boldsymbol{x}_i)] - \mathbb{E}\big[[y_i \cdot S_1(\boldsymbol{x}_i)]\big]\boldsymbol{v})\big\rangle$, where $\|\boldsymbol{v}\| = 1$, so $\mathbb{E}[Gq_i] = 0$. Similar to the proof of (131), (132) and (133) in Lemma 8, we have

$$|Gq_i|^p \lesssim \big|\sigma_{\min}^{-2}(\boldsymbol{x}_i^{\top}\boldsymbol{v}) + \mathbb{E}_{\boldsymbol{x}\sim\sum_{l=1}^{L}\mathcal{N}(\boldsymbol{\mu}_l,\sigma_l^2\boldsymbol{I}_d)}[\sigma_{\min}^{-2}(\boldsymbol{x}_i^{\top}\boldsymbol{v})]\big|^p \tag{211}$$

$$\mathbb{E}[|Gq_i|^p] \lesssim \left(\mathbb{E}_{\boldsymbol{x} \sim \sum_{l=1}^{L} \mathcal{N}(\boldsymbol{\mu}_l, \sigma_l^2 \boldsymbol{I}_d)}[\sigma_{\min}^{-2}(\boldsymbol{x}_i^\top \boldsymbol{v})]\right)^p \leq \tau^{2p} \sqrt{D_2(\boldsymbol{\lambda}, \boldsymbol{M}, \boldsymbol{\sigma})}^p \tag{212}$$

$$\mathbb{E}[\exp(\theta G q_i)] \lesssim 1 + \sum_{p=2}^{\infty} \frac{\theta^p \mathbb{E}[|Gq_i|^p]}{p!} \lesssim 1 + \sum_{p=2}^{\infty} \frac{|e\theta|^p \tau^{2p} D_2(\boldsymbol{\lambda}, \boldsymbol{M}, \boldsymbol{\sigma})^{\frac{p}{2}}}{p^p} \tag{213}$$

$$\leq 1 + \theta^2 (\tau^2 \sqrt{D_2(\boldsymbol{\lambda}, \boldsymbol{M}, \boldsymbol{\sigma})})^2$$

Hence, similar to the derivation of (135), we have

$$\mathbb{P}\left(\frac{1}{n} \sum_{i=1}^{n} Gq_i \geq t\right) \leq \exp\left(-n\theta t + C_{19}\theta^2 \left(\tau^2 \sqrt{D_2(\boldsymbol{\lambda}, \boldsymbol{M}, \boldsymbol{\sigma})}\right)^2\right) \tag{214}$$

for some constant $C_{19} > 0$. Let $\theta = \frac{t}{C_{19}\left(\tau^2 \sqrt{D_2(\boldsymbol{\lambda}, \boldsymbol{M}, \boldsymbol{\sigma})}\right)^2}$ and $t = \left(\tau^2 \sqrt{D_2(\boldsymbol{\lambda}, \boldsymbol{M}, \boldsymbol{\sigma})}\right) \cdot \sqrt{\frac{d \log n}{n}}$, then we have

$$\|\widehat{\boldsymbol{M}}_1 - \boldsymbol{M}_1\| \lesssim \left(\tau^2 \sqrt{D_2(\boldsymbol{\lambda}, \boldsymbol{M}, \boldsymbol{\sigma})}\right) \cdot \sqrt{\frac{d \log n}{n}} \tag{215}$$

with probability at least $1 - 2n^{-\Omega(d)}$.

