# OpenReview forum: "Learning One-hidden-layer Neural Networks on Gaussian Mixture Models with Guaranteed Generalizability"
_ICLR.cc/2021/Conference — Reject_

### Official Review · AnonReviewer3 · 2020-10-13
**Interesting convergence guarantees, but the results are oversold and the numerical experiments rather weak (updated version)**

**Rating:** 4
**Confidence:** 4

**Review:**

**Update after response of the authors**

The authors partly corrected the points I mentioned, and I thank them for extensively addressing my comments. However I still believe that the paper oversells its results and analysis, although in a much less strong manner (e.g. in the last sentence of the abstract) and that many concerns remain. For instance, my concern on the tensor initialization has been partially addressed by the authors which showed that it performed as well as a random initialization close to the solution. The very small dimension (d = 5) for which these simulations were performed is however still a concern for verifying this analysis, which would need to be more rigorous and much more extensive to justify replacing the tensor initialization with the random one.
As a consequence of all the changes made by the authors, I have raised my grade from 3 to 4 (and I would grade the current state of the paper between 4 and 5).

**Summary:**

The paper focuses on the behavior of gradient descent in one-hidden-layer neural networks (with fixed second layer weights), in a “teacher-student” setup. In this setup, the labels are generated by an unknown teacher network with the same architecture, and the student aims at recovering the teacher weights. The main ambition of the paper is to provide guarantees for the convergence of gradient descent on the cross-entropy loss, for an input data which comes from a non-trivial model, here a mixture of Gaussians. The algorithm is moreover initialized with a tensor initialization method, which will be crucial to assess its performance. The paper provides a precise theorem to guarantee convergence of this algorithm, and studies how the learning process can depend on the parameters of the mixture of Gaussians. Importantly, the theoretical analysis also relies on the knowledge of these parameters. In particular, they derive limit regimes in which learning should be very hard, e.g. when the variances of the mixture are either very small or very large. They finally provide numerical evidence to support their claims.

Given the length and available time, I did not check the calculations given in the supplementary material.

**Overall decision:**

Considering all my criticisms and comments, I cannot recommend publication of this paper at ICLR 2021 as it is. For me to reconsider this decision, the authors would have to slightly improve the quality of the writing (see the remarks and typos), to improve the numerical analysis, and to provide a more convincing presentation of the impact and novelty of their theoretical results with respect to the previous literature.

**Strengths of the paper:**

- The authors provide an involved tensor initialization method to start the gradient descent algorithm, which provably reaches a basin of attraction of a local minimum very close to the true weights.

- The bounds derived in Theorem 1, both on the convergence of the algorithm and on the distance between the local minimum and the true weights are quite explicit, in particular as a function of the number n of samples, the dimension d of the data, and the number K of hidden neurons. Having these two bounds together is important and interesting, as it allows not only to probe the optimization but also the generalization properties. Moreover, the assumptions needed on the activation function (Assumption 1) are quite generic, and allow for a large class of activation functions.

- They show that the sample complexity needed for precise estimation with this algorithm is in the scale $d \ (\log d)^2$. I wonder if such a bound is sharp? Also perhaps the authors could comment on the work of [Mei, Bai & Montanari, ‘18], which showed that in this scaling, the landscape of some estimation problems is already trivialized (i.e. close to the population loss landscape). They showed it in particular for a single-hidden-node model: it could be interesting to compare their result with the present paper in the limit $K = 1$.

- Corollary 1 allows to study the impact of the parameters of the mixture of Gaussians (i.e. the structure of the data) on the learning procedure, with the mentioned algorithm. In particular, they show that either too large or too small variances can be detrimental to optimization. While it is intuitive that large variances would harm optimization, the finding on small variances is particularly interesting, as one would expect that small variances in the Gaussian mixture would imply an easier optimization problem. On this point, Fig. 2 is very qualitative, but shows the dependency of the sample complexity on the parameters of the mixture of Gaussians. In particular, Fig.(2b) manages to show the interesting divergence of the sample complexity when the variance goes either to $0$ or $\infty$.

- Figures 3 and 4 do a decent job at showing the dependency of the final convergence time on the parameters of the mixture, of the convergence rate on the size of the hidden layer, and of the distance of the true weights to the critical point achieved by gradient descent on the number of samples. The results are quite consistent with the theoretical analysis.

**Concerns and remarks:**

- In the introduction, the authors explain that they consider a setup in which the labels are generated by a ground truth network, and provide some recent literature on this hypothesis. They however do not mention that this assumption is known as the “teacher-student” setup, and it has been studied for a long time in the statistical learning community (in particular from a statistical physics point of view in which such a setup is very natural). The paper should correct this point by providing a much more exhaustive view of the literature on this topic. For instance, they can refer to [Seung, Sompolinsky & Tishby ‘92], [Engel & Van den Broeck ‘01]. See also [Goldt & al, NeurIPS’19] for recent applications to neural networks.

- Fig 1 (in the numerical experiments) is somehow unclear. While the authors pretend that the sample complexity needed to recover is indeed almost linear in d, this is not obvious from the picture. Moreover, the difference in orders of magnitudes between n (from 6000-60 000) and d (from 1 to 30) indicate that, even with a quasi-linear dependency, the prefactor would be huge, and the authors should discuss this point.

 My two main concerns are the following:

- First, as emphasized in the beginning of Section 4, the tensor initialization is crucial, as it actually returns an estimate already in the basin of attraction of a critical point very close to the ground truth.  However, in Section 5 (on the numerics), the authors precise “We use random initialization rather than tensor initialization to reduce the computational time”, without any justification of how this could impact the results. This reduces greatly the relevance of these numerical results, and their relation with the theoretical findings. This also raises the question: is the tensor initialization numerically tractable? If the tensor initialization provably returns an estimate in the correct basin of attraction, this algorithm is actually doing the most important part of the estimation, and replacing it with a random initialization close to the ground truth removes a lot of the relevance of the theoretical findings (as obviously, when starting in a convex region, gradient descent will work).

- Secondly, the following bold claims can be found in the abstract and the main text, and are very emphasized:

1) “Instead of following the conventional and restrictive assumption in the literature that the input features follow the standard Gaussian distribution, this paper, for the first time, analyzes a more general and practical scenario that the input features follow a Gaussian mixture model of a finite number of Gaussian distributions of various mean and variance.”
2) “This paper provides the first theoretical analysis of learning one-hidden-layer neural networks when the input distribution follows a Gaussian mixture model containing an arbitrary number of Gaussian distributions with arbitrary mean and variance.”

3) “This is the first theoretical characterization about how the input distribution affects the learning performance.”

4) “Theorem 1 provides the first theoretical guarantee of learning one-hidden-layer neural networks with the input following the Gaussian mixture model.”

    In my point of view, these are exaggerated statements. While it is true that there is ample room for new studies of non-trivial input distributions, several previous and impacting papers have followed very similar approaches, beyond [Du&al ‘17] which is the only paper cited by the authors in this context. For instance, the following (very incomplete) list of papers all either consider training a one-hidden-layer neural net on a dataset with a non-trivial covariance, or a mixture of Gaussians data model:
    1. Mei, Montanari & Nguyen [PNAS 2018] study one-hidden-layer networks in the mean-field limit trained on a large class of distributions, including a mixture of Gaussians with the same mean (see Fig 1 of the paper).
    2. Li & Liang [NeurIPS 2018] studied over-parameterized one-hidden-layer nets "when the data comes from mixtures of well-separated distributions".
    3. Yoshida & Okada [NeurIPS 2019] study one-hidden-layer neural networks with inputs drawn from a single Gaussian with arbitrary covariance.
    4. Goldt et al. (arXiv:1909.11500 and arXiv:2006.14709) study one-hidden-layer networks with inputs drawn from a wide class of generative models.
    5. Mignacco et al. (arXiv:2006.06098) give exact equations for the size-one minibatch SGD evolution in a perceptron (i.e. a single-node network) trained on a mixture of Gaussians.
    6. Ghorbani et al. (arXiv:2006.13409) consider labels which depend on low-dimensional projections of the inputs, which, I believe, is very related to a mixture of Gaussians.
Some of the papers in this list (e.g. [1,2,4,6]) also provide a rigorous analysis of some of their results.

While, to the best of my knowledge, the theoretical results of the present paper (i.e. global convergence guarantees for gradient descent with tensor initialization, for data coming from a mixture of Gaussians) are indeed new, their impact and novelty is, I believe, exaggerated. In particular, similar results already exist in the literature cited above, and the new results of this paper should be discussed in comparison to them.

**Minor points and questions:**

- The notations section is very long. While some precisions are useful, many are very standard (N, Z, R, or the transpose, or the L2 norm) and, I believe, do not have to be reminded. Similarly, the footnote at the end of page 5 is not necessary.

- The paper studies one-hidden-layer neural networks with fixed second layer weights. The authors should mention that this setup is known as the committee machine, and they should refer to some literature on this (besides some references given in the concerns section, one can for instance refer to [Aubin&al NeurIPS 2018, Schwarze&al ‘92,’93, Monasson&Zecchina ‘95] and many others).

- Is it possible to add a noise in the gradient descent algorithm without affecting the theoretical findings? Even an uncorrelated noise (i.e. Langevin dynamics), as I expect that the noise in plain SGD will not be easily tractable.

- Algorithm 1 requires a constant learning rate: I wonder if the authors tried (even just numerically) to see if the bounds could be improved by considering an adaptive learning rate (for instance with a linear decrease)?

**Typos:**

- At the beginning of the introduction: “Neural” → “neural”.
- Just after, I believe: “theoretical underpin of leaning neural networks” → “theoretical underpin of learning in neural networks”.
- Again, just after: “lack of the theoretical generalization guarantee” → “lack of theoretical generalization guarantees”.
- In the “Contributions” paragraph,  the “etc” when listing the applications of the Gaussian mixture model does not read well.
- In the “Contributions”: “One interesting finding is the”→ “One interesting finding is that”.
- Just after: “all the variance approaches” → “all the variances approach”.
- In Section 2: “Let kappa denote the number that kappa =...” → “Let kappa = …”
- Below eq.(6): sigma_l \in R → sigma_l \in R_+.
- End of page 4: “more details of” → “more details on”

---

> ### Author Response · Authors · 2020-11-23
> **Summary of the changes for the paper, and the replies to the first three questions.**
>
> Thank you very much for the comments. We have revised our paper extensively based on the comments from all the reviewers. Major changes include: 1. We rewrote the introduction with a more extensive literature review and stated our novelty and contributions in comparison with the literature. 2. We added a new experiment (section 5.1) on tensor initialization, justifying the effectiveness of our algorithm. 3. We clarified our problem setup and improved the presentation of our algorithm and main results. We are very excited to see the improvement of our paper based on reviewers’ comments and appreciate your feedback. We look forward to your evaluation and comments about the revision.
>
> The point-to-point answers to your questions are as follows.
>
> Questions in ''Strengths of the paper'':
>
> $\boldsymbol{Q1}$: They show that the sample complexity needed for precise estimation with this algorithm is in the scale $d \log d^2$. I wonder if such a bound is sharp? Also perhaps the authors could comment on the work of [Mei, Bai & Montanari, ‘18], which showed that in this scaling, the landscape of some estimation problems is already trivialized (i.e. close to the population loss landscape). They showed it in particular for a single-hidden-node model: it could be interesting to compare their result with the present paper in the limit K=1.
>
> $\boldsymbol{A1}$: Our sample complexity bound is in the same order as the sample complexity with the standard Gaussian input in Zhong et al. 2017 and Fu et al. 2020,  indicating that our method can handle input from the Gaussian mixture model without increasing the order of the sample complexity.  Our bound is almost order-wise optimal with respect to d because the degree of freedom is $dK$. The additional multiplier of $\log^2{d}$ results from the concentration bound in the proof technique.  We highly doubt whether the exact order d can be achieved due to the limitation of proof techniques.
>
> One main component in the proof of Theorem 1 is to show that if Eqn.11 holds, the landscape of the empirical risk is close to that of the population risk in a local neighborhood of $\boldsymbol{W}^*$. Mei et al, 2018 shows that when the sample complexity is $O(d \log{d})$, both functions are sufficiently close when $K=1$, but it is not clear if their approach can be extended to $K>1$. Here, focusing on the Gaussian mixture model, we explicitly quantify the impact of the parameters of the input distribution on the landscapes of these functions. We added the discussion to the main text after Theorem 1 and Appendix-C.
>
> Questions in ''Concerns and remarks''
>
> $\boldsymbol{Q2}$: In the introduction, the authors explain that they consider a setup in which the labels are generated by a ground-truth network, and provide some recent literature on this hypothesis. They however do not mention that this assumption is known as the “teacher-student” setup, and it has been studied for a long time in the statistical learning community (in particular from a statistical physics point of view in which such a setup is very natural). The paper should correct this point by providing a much more exhaustive view of the literature on this topic. For instance, they can refer to [Seung, Sompolinsky & Tishby ‘92], [Engel & Van den Broeck ‘01]. See also [Goldt & al, NeurIPS’19] for recent applications to neural networks.
>
> $\boldsymbol{A2}$: Thank you very much for introducing the “teacher-student” setup. We have revised the discussion of our problem using the teacher-student setup and cited the related work.
>
> $\boldsymbol{Q3}$: Fig 1 (in the numerical experiments) is somehow unclear. While the authors pretend that the sample complexity needed to recover is indeed almost linear in d, this is not obvious from the picture. Moreover, the difference in orders of magnitudes between n (from 6000-60 000) and d (from 1 to 30) indicate that, even with a quasi-linear dependency, the prefactor would be huge, and the authors should discuss this point.
>
> $\boldsymbol{A3}$: Thanks for the comment. We revised the result (Figure 2 in the revision here) to show the sample complexity when $d$ changes from 5 to 100. The linear trend is obvious now. We agree that the constant can be largely based on the selection of the network structure, the input distribution. We added this comment after Figure 2.

---

> > ### Author Response · Authors · 2020-11-23
> > **Replies to two your major concerns.**
> >
> > $\boldsymbol{Q4}$: My two main concerns are the following:
> > First, as emphasized in the beginning of Section 4, the tensor initialization is crucial, as it actually returns an estimate already in the basin of attraction of a critical point very close to the ground truth. However, in Section 5 (on the numerics), the authors precise “We use random initialization rather than tensor initialization to reduce the computational time”, without any justification of how this could impact the results. This reduces greatly the relevance of these numerical results, and their relation with the theoretical findings. This also raises the question: is the tensor initialization numerically tractable? If the tensor initialization provably returns an estimate in the correct basin of attraction, this algorithm is actually doing the most important part of the estimation, and replacing it with a random initialization close to the ground truth removes a lot of the relevance of the theoretical findings (as obviously, when starting in a convex region, gradient descent will work).
> >
> > $\boldsymbol{A4}$: Thank you for the helpful comment. We added experiments about tensor initialization in Section 5.1. Fig.1 shows the accuracy of the returned model by our algorithm. We compare tensor initialization with a random   initialization in a local region $\{\boldsymbol{W}\in\mathbb{R}^{d\times K}:\frac{||\boldsymbol{W}-\boldsymbol{W}^*||_F}{||\boldsymbol{W}^*||_F}\leq \epsilon\}$. Here $d=5$, $K=2$, $\lambda_1=\lambda_2=0.5$, $\boldsymbol{\mu}_1=-\boldsymbol{1}$ and $\boldsymbol{\mu}_2=\boldsymbol{0}$. Tensor initialization returns an initial point close to $\boldsymbol{W}^*$ with a relative error of $0.61$.
> > If the random initialization is also close to $\boldsymbol{W}^*$, e.g., $\epsilon=0.1$, the algorithm converges to a critical point from both initializations, and the linear convergence rate is the same. If the random initialization is far away,  e.g., $\epsilon=1.5$, the algorithm does not converge.  On a MacBook Pro with Intel(R) Core(TM) i5-7360U CPU at 2.30GHz and MATLAB 2017a, it takes 0.55 seconds to compute the tensor initialization.  We consider random initialization with $\epsilon=0.1$ in the following experiments to simplify the computation.
> >
> > $\boldsymbol{Q5}$: While, to the best of my knowledge, the theoretical results of the present paper (i.e. global convergence guarantees for gradient descent with tensor initialization, for data coming from a mixture of Gaussians) are indeed new, their impact and novelty is, I believe, exaggerated. In particular, similar results already exist in the literature cited above, and the new results of this paper should be discussed in comparison to them.
> >
> > $\boldsymbol{A5}$: We are very grateful for the related references you introduced to us. We have revised the inaccurate statements in the revision and add a discussion about the references and the comparison of our work. We have rewritten and reorganized the introduction, and we believe our contributions should be clearer in the revision. Please refer to the revised paper for details.
> > Specifically, the statements you mentioned as revised as follows
> >
> > $(\boldsymbol{1})$ “Instead of following the conventional and restrictive assumption in the literature that the input features follow the standard Gaussian distribution, this paper, for the first time, analyzes a more general and practical scenario that the input features follow a Gaussian mixture model of a finite number of Gaussian distributions of various mean and variance.”
> >
> > We change it to “This paper analyzes a  general and practical scenario that the input features follow a Gaussian mixture model of a finite number of Gaussian distributions of various mean and variance.”
> >
> > $(\boldsymbol{2})$ “This paper provides the first theoretical analysis of learning one-hidden-layer neural networks when the input distribution follows a Gaussian mixture model containing an arbitrary number of Gaussian distributions with arbitrary mean and variance.”
> >
> > We change it to “This paper analyzes a theoretical analysis …..”
> >
> > $(\boldsymbol{3})$ “This is the first theoretical characterization about how the input distribution affects the learning performance.”
> >
> > We change it to “to the best of our knowledge, this paper provides the first theoretical and explicit characterization about how the mean and variance of the input distribution affects the sample complexity and learning rate.”
> >
> > $(\boldsymbol{4})$ “Theorem 1 provides the first theoretical guarantee of learning one-hidden-layer neural networks with the input following the Gaussian mixture model.”
> >
> >
> > We change it to “To the best of our knowledge, Theorem 1 provides the first explicit characterization of the sample complexity and learning rate  when the input follows the Gaussian mixture model.”

---

> > > ### Author Response · Authors · 2020-11-23
> > > **Replies to your ''minor points and questions'' and ''typos''**
> > >
> > > ''Minor points and questions'':
> > >
> > > $\boldsymbol{Q6}$: The notations section is very long. While some precisions are useful, many are very standard (N, Z, R, or the transpose, or the L2 norm) and, I believe, do not have to be reminded. Similarly, the footnote at the end of page 5 is not necessary.
> > >
> > > $\boldsymbol{A6}$: We have revised accordingly.
> > >
> > > $\boldsymbol{Q7}$: The paper studies one-hidden-layer neural networks with fixed second layer weights. The authors should mention that this setup is known as the committee machine, and they should refer to some literature on this (besides some references given in the concerns section, one can for instance refer to [Aubin&al NeurIPS 2018, Schwarze&al ‘92,’93, Monasson&Zecchina ‘95] and many others).
> > >
> > > $\boldsymbol{A7}$: We have added the discussion of committee machined and cited these related works.
> > >
> > > $\boldsymbol{Q8}$: Is it possible to add a noise in the gradient descent algorithm without affecting the theoretical findings? Even an uncorrelated noise (i.e. Langevin dynamics), as I expect that the noise in plain SGD will not be easily tractable.
> > >
> > > $\boldsymbol{A8}$: Thank you for the interesting idea. We very much like your idea as it is indeed a good approximation and potentially manageable approach to analyze SGD. We revised our formulation and results accordingly in the revision.
> > >
> > > We assume an i.i.d. zero-mean random noise $\{\nu_i\}_{i=1}^n\in\mathbb{R}^{d\times K}$ with  bounded magnitude   $|(\nu_i)_{jk}|\leq\xi$ ($j \in [d], k\in [K])$ for some $\xi \geq 0$ when computing the gradient of the loss in (4) for every training sample $(\boldsymbol{x}_i, y_i)$. Then if $n$ is small, the average noise is large. If $n$ is large, the average noise is small. Then, there is an additional error term in (12) and (13) related to $\xi$.  With the noise in the gradient, there is an additional error term of $O(\xi\sqrt{d\log n/n})$. Please refer to the revision for details. We highlighted major changes in the main text and in the technical details in the Appendix.
> > >
> > >
> > > $\boldsymbol{Q9}$: Algorithm 1 requires a constant learning rate: I wonder if the authors tried (even just numerically) to see if the bounds could be improved by considering an adaptive learning rate (for instance with a linear decrease)?
> > >
> > > $\boldsymbol{A9}$: Thanks for the interesting idea. We tested the backtracking line search following Armijo’s rule numerically. We found that with both constant step size ($\eta_0=10$) and adaptive step size ($\eta_0=\frac{100}{2^j}, j=1,2,\cdots,10$), the algorithm converges linearly. To achieve a $10^{-6}$ error to the critical point, the gradient descent with constant step size needs 560 iterations, while the gradient descent with adaptive step size only requires 240 iterations. That means the convergence rate v is smaller for the adaptive step sizing, indicating a faster convergence. However, our current proof techniques do not extend directly to analyze adaptive step size. We are still working on it and will update if we get any theoretical results.
> > >
> > > We added the following comment to the paper
> > > “Algorithm 1 employs a constant step size. One can potentially speed up the convergence, i.e., reduce $v$, by using a variable step size. We leave the corresponding theoretical analysis for future work.”
> > >
> > > ''Typos'':
> > >
> > > $\boldsymbol{Q10}$: Typos
> > >
> > > $\boldsymbol{A10}$: Thank you. We have addressed these typos. We have also proofread the paper to correct some other typos.

---

### Official Review · AnonReviewer1 · 2020-10-27
**The paper is interesting and provides some new theoretical insight into the learning problem of fully connected neural networks when the input data are from mixture of Gaussian distributions.**

**Rating:** 7
**Confidence:** 4

**Review:**

In the paper, the authors provide theoretical analysis of learning one-hidden-layer neural networks when the input distribution follows a mixture of location-scale Gaussian distributions instead of single location-scale Gaussian distribution as in the previous work. I think the results in the paper are interesting

Here are my comments with the paper:

(1) The literature with Gaussian mixtures is quite poor. Except for the references with the application of Gaussian mixtures in the paper, I think the authors may consider adding a few more relevant references about theoretical aspect of Gaussian mixtures. For instance, the work of [1] provides convergence rate/ sample complexity for estimating unknown location and scale parameters when the data are generated from Gaussian mixtures. Furthermore, the work of [2] provide theoretical analysis of optimization algorithms, such as gradient descent/ EM/ Newton algorithms,  for learning location and scale parameters in Gaussian mixtures (Section 4.2 in this work).

(2) The assumption that the weight, location, scale of Gaussian mixtures as well as the number of components $L$ are known is quite strong in my opinion. In particular, the number of components $L$ in Gaussian mixtures is rarely known in practice; therefore, we usually choose some $\bar{L}$ as the number of components and $\bar{L}$ can be much larger than $L$. By doing that, we overspecify the number of components in Gaussian mixtures. This over-specification leads to the slow convergence rates of estimating weight, location, scale of Gaussian mixtures; see the references [1], [2], and [3]. For the settings that being considered by the authors, when $\bar{L} = L + 1$, if we use EM algorithm, the convergence rates of estimating these parameters from the EM algorithm for location parameter is $(d/n)^{1/4}$ and for scale parameter is $(d/n)^{1/2}$ when $d \geq 2$ (n stands for the sample size) (see references [2], [3], and [4] for details). Therefore, in light of the results in the paper, the total sample complexity is $\sqrt{d \log n/n} + (d/n)^{1/4}$ when the parameters of Gaussian mixtures are unknown. When the covariance matrices are not spherical and $\bar{L} > L$, the work of [1] show that the sample complexity of estimating location and covariance matrices depends on the solvability of a system of polynomial equations and eventually grows with $\bar{L} - L$. I think the authors should provide a clarification of these points in the paper.

(3) The paper specifically assumes that the covariance matrices of each component are $\sigma_{l}^2 I_{d}$, i.e., homogeneous among all dimension in each subpopulation. When the covariance matrices have a bit more realistic structures like $\text{diag}(\sigma_{l1}^2, \ldots, \sigma_{ld}^2) I_{d}$ for all $1 \leq l \leq L$, will the results in Theorem 1 still hold?

(4) In Theorem 1, what is the intuition behind $K^{5/2}$ and $\Gamma(\lambda, M, \sigma, W^{*})$$ on the difference between $\widehat{W}_{n}$ and $W^{*}$ as well as $D_{12}$ in the condition of sample size $n$?

(5) Can the authors provide some initial theoretical analysis/ discussion for the setting of multi-layer neural networks? I agree that one layer neural network is quite interesting; however, it will be useful for the readers to understand the challenges of extending the
current results to the multi-layers settings.

(6) A few minor comments:

- In Definition 1, what is $M_{3}(I_{d}, I_{d}, \alpha)$?



References:

[1] N. Ho and L. Nguyen. Convergence rates of parameter estimation for some weakly identifiable finite mixtures. Annals of Statistics, 44(6), 2726-2755, 2016

[2] N. Ho, R. Dwivedi, K. Khamaru, M. J. Wainwright, M. I . Jordan, B. Yu. Instability, computational efficiency and statistical accuracy. Arxiv preprint Arxiv: 2005.11411.

[3] R. Dwivedi, N. Ho, K. Khamaru, M. J. Wainwright, M. I. Jordan, B. Yu. Singularity, misspecification, and the convergence rate of EM. To appear, Annals of Statistics.

[4] R. Dwivedi, N. Ho, K. Khamaru, M. J. Wainwright, M. I. Jordan, B. Yu. Sharp analysis of Expectation-Maximization for weakly identifiable models. AISTATS, 2020.

---

> ### Author Response · Authors · 2020-11-23
> **Summary of the changes for the paper, and the replies to the first two comments.**
>
> Thank you very much for the comments. We have revised our paper extensively based on the comments from all the reviewers. Major changes include: 1. We rewrote the introduction with a more extensive literature review and stated our novelty and contributions in comparison with the literature. 2. We added a new experiment (section 5.1) on tensor initialization, justifying the effectiveness of our algorithm. 3. We clarified our problem setup and improved the presentation of our algorithm and main results. We are very excited to see the improvement of our paper based on reviewers’ comments and appreciate your feedback. We look forward to your evaluation and comments about the revision.
>
> The point-to-point answers to your questions are as follows.
>
> $\boldsymbol{Q1}$: The literature with Gaussian mixtures is quite poor. Except for the references with the application of Gaussian mixtures in the paper, I think the authors may consider adding a few more relevant references about the theoretical aspect of Gaussian mixtures. For instance, the work of [1] provides convergence rate/ sample complexity for estimating unknown location and scale parameters when the data are generated from Gaussian mixtures. Furthermore, the work of [2] provides theoretical analysis of optimization algorithms, such as gradient descent/ EM/ Newton algorithms, for learning location and scale parameters in Gaussian mixtures (Section 4.2 in this work).
>
> $\boldsymbol{A1}$: Thank you for introducing related works. We have indeed these papers in the revision. “The parameters of the mixture model can be estimated from data and the theoretical characterization of the learning methods and the required number of samples have been well investigated, e.g.,   Ho & Nguyen (2016); Ho et al. (2005); Dwivedi et al.(2020a;b).
>
> $\boldsymbol{Q2}$: The assumption that the weight, location, scale of Gaussian mixtures as well as the number of components $L$ are known is quite strong in my opinion. In particular, the number of components $L$ in Gaussian mixtures is rarely known in practice; therefore, we usually choose some $\bar{L}$ as the number of components $L$ and can be much larger than $L$. By doing that, we overspecify the number of components in Gaussian mixtures. This over-specification leads to the slow convergence rates of estimating weight, location, scale of Gaussian mixtures; see the references [1], [2], and [3]. For the settings that being considered by the authors, when $\bar{L}=L+1$, if we use EM algorithm, the convergence rates of estimating these parameters from the EM algorithm for location parameter is  $(\frac{d}{n})^\frac{1}{4}$ and for scale parameter is $(\frac{d}{n})^\frac{1}{2}$ when $d\geq 2$ (n stands for the sample size) (see references [2], [3], and [4] for details). Therefore, in light of the results in the paper, the total sample complexity is $\sqrt{\frac{d\log n}{n}}+(\frac{d}{n})^\frac{1}{4}$ when the parameters of Gaussian mixtures are unknown. When the covariance matrices are not spherical and $\bar{L}>L$, the work of [1] show that the sample complexity of estimating location and covariance matrices depends on the solvability of a system of polynomial equations and eventually grows with
> $\bar{L}-L$. I think the authors should provide clarification of these points in the paper.
>
> $\boldsymbol{A2}$: Thank you for the excellent point. We have added the following discussion to the paper. We need to clarify that $d\log n/n$ is the estimation error, not the sample complexity. We are not sure if the estimation error of W^* and the distribution parameters can be added directly, so we discuss them separately as follows.
> “...The above results assume the parameters of the Gaussian mixture are known. These parameters can be estimated by the EM algorithm (Redner & Walker, 1984) and the moment-based method (Hsu & Kakade, 2013) in practice.  The EM algorithm returns model parameters within Euclidean distance $O((\frac{d}{n})^\frac{1}{2})$ when the number of mixture components $L$ is known. When $L$ is unknown,  one usually chooses some estimate $\bar{L}$ as the number of components and $\bar{L}$ can be much larger than $L$. In this over-specified setting, the estimation error by the EM algorithm scales as $O((\frac{d}{n})^\frac{1}{4})$. Please refer to (Ho & Nguyen, 2016; Ho et al., 2005; Dwivedi et al., 2020a;b) for details.”

---

> > ### Author Response · Authors · 2020-11-23
> > **Replies to the comment 3 to 6**
> >
> > $\boldsymbol{Q3}$: The paper specifically assumes that the covariance matrices of each component are $\sigma_l^2\boldsymbol{I_d}$, i.e., homogeneous among all dimension in each subpopulation. When the covariance matrices have a bit more realistic structures like $diag(\sigma_{l1}^2,…,\sigma_{ld}^2)\boldsymbol{I}_d$  for all $1\leq l \leq L$, will the results in Theorem 1 still hold?
> >
> > $\boldsymbol{A3}$: Yes. The result still holds. The only differences lie in Property 3 and Lemma 7. The extension is not difficult. The discussion in Corollary 1 will be the same if we replace $\sigma_l$ with $\sigma_{l(i)}$ in $diag(\sigma_{l1}^2,…,\sigma_{ld}^2)\boldsymbol{I}_d$. For simplicity of analysis and the presentation of the results, we use the same $\sigma$ in the paper.
> >
> > $\boldsymbol{Q4}$: In Theorem 1, what is the intuition behind $K^\frac{5}{2}$ and $\Gamma(\lambda, M, \sigma, W*)$ on the difference between  $\widehat{W_n}$ and $W^*$ as well as $D_{12}$ in the condition of sample size $n$?
> >
> > $\boldsymbol{A4}$: Note that the output of $K$ neurons are added together to generate the output label, and $W$ is in $R^{d \times K}$. Intuitively, when $K$ increases, it is more challenging for $\widehat{W_n}$ to be close to $W^*$. The power of $5/2$ results from our proof technique, and we do not try to optimize over $K$ in the proof as our main focus is on $n$ and $d$.
> > We removed $\Gamma$ and $D_{12}$ in the main text to simplify the representation of Theorem 1. In the revision, the sample complexity depends on the $\mathcal{B}$ parameter, and Corollary 1 discusses how the parameters of the Gaussian mixture model affects $\mathcal{B}$ and thus the sample complexity.
> >
> > $\boldsymbol{Q5}$: Can the authors provide some initial theoretical analysis/ discussion for the setting of multi-layer neural networks? I agree that one layer neural network is quite interesting; however, it will be useful for the readers to understand the challenges of extending the current results to the multi-layers settings.
> >
> > $\boldsymbol{A5}$: Most works in this line of work consider one-hidden-layer networks because for multiple-hidden-layer networks, due to the concatenation of nonlinear activation functions, the analysis of the landscape of the empirical risk and the design of a proper initialization is more challenging and may require the development of new tools. Because we want to study the impact of input distributions, we start with one-hidden-layer neural networks in this paper. We believe the results can be extended to a two-layer-network (one hidden layer plus a pooling layer) with some modifications. We are investigating multiple hidden layers now and we expect some new tools need to be developed. We added the discussion to the conclusions in the paper.
> >
> >
> > $\boldsymbol{Q6}$ A few minor comments: In Definition 1, what is  $M_3(\boldsymbol{I_d},\boldsymbol{I_d},\alpha)$?
> >
> > $\boldsymbol{A6}$: Sorry for the confusion. $M_3(\boldsymbol{I_d},\boldsymbol{I_d},\alpha)$ is a projection of the third-order tensor $M_3$ onto the vector $\alpha$, so it will be a matrix. We define this notation in Eqn 1. This notation also appears in Zhong et al. 2017 and Fu et al. 2020.

---

> > ### Comment · AnonReviewer1 · 2020-11-23
> > **Revise the references**
> >
> > Thank the authors for the detailed rebuttal.  The authors have addressed all of my concerns. I decide to keep my current score 7 with the paper.
> >
> > For the reference Ho et al., "Instability, computational efficiency and statistical accuracy", there is a typo with the year. It should be in 2020. Furthermore, there are inconsistencies in style in the bibliography (some authors' first names are without abbreviation while the other first names are abbreviated). Please fix the references.

---

> > > ### Author Response · Authors · 2020-11-23
> > > **We have revised the references**
> > >
> > > Thank you for your comments and appreciation of our work. We have fixed the references in the submission.

---

### Official Review · AnonReviewer4 · 2020-10-28
**Reviews**

**Rating:** 6
**Confidence:** 3

**Review:**

This paper considers the problem of learning one-hidden-layer neural networks with Gaussian mixture input in the teacher-student setting. The authors consider the neural network with sigmoid activation functions and the learning algorithm is gradient descent plus tensor initialization.  There is a line of research studying such a problem, and the main contribution of the current paper is to extend the standard Gaussian input distribution to the mixture of Gaussian input distribution. My main concern is the contribution of the current paper. The main techniques used in this paper seem to be based on existing approaches in Fu et al, 2020 and Zhong et al, 2017, and the Gaussian mixture input setting considered in this paper seems not to be very interesting and realistic. Here are some problems I have for the current paper:
1. Please clarify the main differences of the current analysis compared with Fu et al, 2020 and Zhong et al, 2017.
2. Please elaborate more about Assumption 1. Intuitively, what can we imply from this assumption and why you think it is a mild assumption?
3. The description of the tensor initialization at the end of page 4 is not very clear.
4. Please add some comments on functions in Definition 2,3,4. It is unclear what are the meanings of these functions.
5. Thereom 1 looks not correct since there is no requirement on the step size in Algorithm1. In addition, why the parameter v can belong to (0,1)?
6. Since the paper claims to use tensor initialization, the experiments should also include such results.
7. The presentation of the current paper is good. However, there are some concurrent works [1,2] also study the training and generalization of neural networks, the authors may want to discuss them in the introduction section.

Reference:
[1] Zou, Difan, et al. "Gradient descent optimizes over-parameterized deep ReLU networks." Machine Learning 109.3 (2020): 467-492.
[2] Cao, Yuan, and Quanquan Gu. "Generalization bounds of stochastic gradient descent for wide and deep neural networks." Advances in Neural Information Processing Systems. 2019.

---

> ### Author Response · Authors · 2020-11-23
> **Summary of the changes for the paper, and the replies to the first three comments.**
>
> Thank you very much for the comments. We have revised our paper extensively based on the comments from all the reviewers. Major changes include: 1. We rewrote the introduction with a more extensive literature review and stated our novelty and contributions in comparison with the literature. 2. We added a new experiment (section 5.1) on tensor initialization, justifying the effectiveness of our algorithm. 3. We clarified our problem setup and improved the presentation of our algorithm and main results. We are very excited to see the improvement of our paper based on reviewers’ comments and appreciate your feedback. We look forward to your evaluation and comments about the revision.
>
> The point-to-point answers to your questions are as follows.
>
> $\boldsymbol{Q1}$: Please clarify the main differences of the current analysis compared with Fu et al, 2020 and Zhong et al, 2017.
> Thank you for your comments.
>
> $\boldsymbol{A1}$: Our paper is an extension of Zhong et al, 2017 and Fu et al, 2020. Although we follow the basic framework of analysis in these two papers, we made new technical contributions from the following three aspects.
>
> 1.If we directly apply the existing matrix concentration inequalities in these works in bounding the error between the empirical loss and the population loss, the resulting sample complexity would be $O(d^3)$ and cannot reflect the influence of each component of the Gaussian mixture distribution. We develop a new bound for high-order moments of the Gaussian mixture model and prove a new version of Bernstein’s inequality (see Eqn 136) so that the final bound is $d\log^2 d$. 2. The analysis of the Hessian of the population loss in these works can not be extended to the Gaussian mixture model. We developed new tools using some good properties of symmetric distribution and even function, and our approach can be applied to other activations like tanh or erf. 3. The tensor initialization in these works only holds for the standard Gaussian distribution.    We exploit a more general definition of tensors from Janzamin et al, 2014 for the tensor initialization in our algorithm. We also develop new error bounds for the initialization.
>
> We added the above discussion to both the main text and the Appendix.
>
> $\boldsymbol{Q2}$: Please elaborate more about Assumption 1. Intuitively, what can we imply from this assumption and why you think it is a mild assumption?
>
> $\boldsymbol{A2}$: Thanks for the comment. Because we decompose tensor $\boldsymbol{M}_3$ to estimate $\boldsymbol{w}_j^*$, we need $\boldsymbol{M}_3$ to be nonzero, which is guaranteed by Assumption 1. Assumption 1.1 implies that $\boldsymbol{M}_3$ is nonzero.  Assumption 1.2 implies that if the Gaussian mixture model is symmetric, $\boldsymbol{P}_2$ is nonzero.
>
> By mild we mean  given $L$, if Assumption 1 is not met for some $(\boldsymbol{\lambda}_0,\boldsymbol{M}_0,\boldsymbol{\sigma}_0)$, there exists  an infinite number of $(\boldsymbol{\lambda}',\boldsymbol{M}',\boldsymbol{\sigma}')$   in any neighborhood of  $(\boldsymbol{\lambda}_0,\boldsymbol{M}_0,\boldsymbol{\sigma}_0)$ such that Assumption 1 holds for $(\boldsymbol{\lambda}',\boldsymbol{M}',\boldsymbol{\sigma}')$.
>
> We revised the paper accordingly.
>
> $\boldsymbol{Q3}$: The description of the tensor initialization at the end of page 4 is not very clear.
>
> $\boldsymbol{A3}$: Thanks for the comment. We revised the tensor initialization in the paper and we hope it is clearer this time. Briefly speaking,  our tensor initialization method is extended from (Janzamin et al., 2014) and (Zhong et al., 2017b). Our method is quite similar to the tensor method in Zhong et al. 2017. The idea is to compute quantities ($\boldsymbol{M}_j$ in Eqn.10 is a $j$th order tensor) that are tensors of  $\boldsymbol{w}_i^*$ and then apply tensor decomposition method to estimate  $\boldsymbol{w}_i^*$. Because $\boldsymbol{M}_j$  can only be estimated from training samples, tensor decomposition does not return $\boldsymbol{w}_i^*$ exactly but provides a close approximation as an initialization. The main idea of this algorithm is to obtain the direction and magnitude of $\boldsymbol{w}_j^*$ separately, by computing some tensors composed of inputs $x_i$ and outputs $y_i$. To obtain the direction information of  $\boldsymbol{w}_i^*$, we apply the KCL method for tensor decomposition. To acquire the magnitude information, we solve a linear system of equations.

---

> > ### Author Response · Authors · 2020-11-23
> > **Replies to the comments 4 to 6.**
> >
> > $\boldsymbol{Q4}$: Please add some comments on functions in Definition 2,3,4. It is unclear what are the meanings of these functions.
> >
> > $\boldsymbol{A4}$: Thanks for the comments. We first moved these definitions to the appendix and simplified that representation of Theorem 1 in the paper. Now the sample complexity depends on the $\mathcal{B}$ parameter only, and the convergence rate depends on $v$ only. Corollary 1 discusses how the parameters of the Gaussian mixture model affects $\mathcal{B}$ (and thus the sample complexity) and $v$ (and thus the convergence rate).
> >
> > Regarding the original definitions, these three functions appear as factors in the lower bound for the sample complexity. $\rho$ function is defined to compute the lower bound of the Hessian of the population risk with a Gaussian input. $\Gamma$ function is the weighted sum of $\rho$ function under the mixture Gaussian model. $\Gamma$ function is positive and upper bounded by a small value. It is increasing when $|\mu_{l(i)}|$ increases. When $\sigma_l$ increases, $\Gamma$ increases first and then decreases. $\Gamma$ goes to zero if all $||\boldsymbol{\mu}_l||_\infty$ or all $\sigma_l$ go to infinity. $D$ function is a normalized parameter for the means and variances. It is lower bounded by 1. It is an increasing function of $||\boldsymbol{\mu}_l||_\infty$ and a decreasing function of $\sigma_l$. We have moved these definitions to Appendix Section A. For simplicity of understanding Theorem 1, we use $\mathcal{B}(\boldsymbol{\lambda},\boldsymbol{M},\boldsymbol{\sigma},\boldsymbol{W^*})$ and $\mathcal{v}(\boldsymbol{\lambda},\boldsymbol{M},\boldsymbol{\sigma},\boldsymbol{W^*})$ to integrate these three quantities above.
> >
> > $\boldsymbol{Q5}$: Theorem 1 looks not correct since there is no requirement on the step size in Algorithm1. In addition, why the parameter v can belong to (0,1)?
> >
> > $\boldsymbol{A5}$: Thank you for pointing it out. We have added a requirement of the step size in the statement of Theorem 1 and Algorithm 1. We revised the description of the rate v in Theorem 1 so that it is easier to see that it is less than 1. The exact expression of $v$ is from equations (165) and (167). One can see from these two equations that it is between 0 and 1.
> >
> > $\boldsymbol{Q6}$: Since the paper claims to use tensor initialization, the experiments should also include such results.
> >
> > $\boldsymbol{A6}$: Thank you for the helpful comment. We added experiments about tensor initialization in Section 5.1. Fig.1 shows the accuracy of the returned model by our algorithm. We compare tensor initialization with a random   initialization in a local region $\{\boldsymbol{W}\in\mathbb{R}^{d\times K}:\frac{||\boldsymbol{W}-\boldsymbol{W}^*||_F}{||\boldsymbol{W}^*||_F}\leq \epsilon\}$. Here $d=5$, $K=2$, $\lambda_1=\lambda_2=0.5$, $\boldsymbol{\mu}_1=-\boldsymbol{1}$ and $\boldsymbol{\mu}_2=\boldsymbol{0}$. Tensor initialization returns an initial point close to $\boldsymbol{W}^*$ with a relative error of $0.61$.
> > If the random initialization is also close to $\boldsymbol{W}^*$, e.g., $\epsilon=0.1$, the algorithm converges to a critical point from both initializations, and the linear convergence rate is the same. If the random initialization is far away,  e.g., $\epsilon=1.5$, the algorithm does not converge.  On a MacBook Pro with Intel(R) Core(TM) i5-7360U CPU at 2.30GHz and MATLAB 2017a, it takes 0.55 seconds to compute the tensor initialization.  We consider random initialization with $\epsilon=0.1$ in the following experiments to simplify the computation.

---

> > > ### Author Response · Authors · 2020-11-23
> > > **Replies to the last comment**
> > >
> > > $\boldsymbol{Q7}$: The presentation of the current paper is good. However, there are some concurrent works [1,2] also study the training and generalization of neural networks, the authors may want to discuss them in the introduction section.
> > > Reference: [1] Zou, Difan, et al. "Gradient descent optimizes over-parameterized deep ReLU networks." Machine Learning 109.3 (2020): 467-492. [2] Cao, Yuan, and Quanquan Gu. "Generalization bounds of stochastic gradient descent for wide and deep neural networks." Advances in Neural Information Processing Systems. 2019.
> > >
> > > $\boldsymbol{A7}$: Thank you for introducing these references.  [1] was originally cited in our paper but might somehow get cut out during the revisions among authors. We have added the discussion of these references in the revision.  Specifically, [1] shows that gradient descent can find the global minimum of the empirical risk function from random initialization in over-parameterized networks. However, there is no discussion about whether the learned model with a zero training error would achieve a small test error or not. In fact, many works in the literature focus on the optimization landscape of the empirical risk and how to find the global minimum of the empirical risk, but do not discuss the test accuracy of the learned model. [2] provides the bounds of the generalization error of the learned model by stochastic gradient descent (SGD) in deep neural networks,  based on the assumption that there exists a good model with a small test error around the initialization of the SGD algorithm, and  no discussion is provided about how to find such an initialization. In contrast, our tensor initialization method in this paper provides an initialization that is close to the ground-truth teacher model such that our algorithm can find this model with a zero test error.  We have added these references and the discussion in the revision.

---

### Official Review · AnonReviewer2 · 2020-10-31
**This paper investigates the training convergence of the general one-hidden-layer fully connected neural network  in a more general and practical scenario that the input features follow a Gaussian Mixture Model (GMM) distribution and proves that the GMM induced learning algorithm converges linearly to a critical point  of the empirical risk function. This result makes a progress on the training convergence study of this network.**

**Rating:** 6
**Confidence:** 4

**Review:**

This paper analyzes the convergence behaviour of the general one-hidden-layer fully connected neural network  in the practical scenario that the input features follow a Gaussian Mixture Model (GMM) distribution. Under certain assumptions, the authors prove that the GMM nduced learning algorithm converges linearly to a critical point  of the empirical risk function, which really  makes a progress on the training convergence study of this network.  However, I  have the following concerns: (1).   What is the ground-truth weights?  For a large traning set, it is possible that there may be no such ground-truth weights. Moroever, the ground-truth weights cannot  form a critical point of the empiirical risk function. So, this assumption is not reasonable. (2).  The assumptin of the Gaussian mixture model is special, not general, and its parameters are  assumed to be known a priori. This  may be too strict and limits the significance of the result. (3). There are some errors  in the mathematical denotations like Eq.(4).

---

> ### Author Response · Authors · 2020-11-22
> **Summary of the changes for the paper, and the replies to the first two comments.**
>
> Thank you very much for the comments. We have revised our paper extensively based on the comments from all the reviewers. Major changes include 1. We rewrote the introduction with a more extensive literature review and stated our novelty and contributions in comparison with the literature. 2. We added a new experiment (section 5.1) on tensor initialization, justifying the effectiveness of our algorithm. 3. We clarified our problem setup and improved the presentation of our algorithm and main results. We are very excited to see the improvement of our paper based on reviewers’ comments and appreciate your feedback. We look forward to your evaluation and comments about the revision.
>
> The point-to-point answers to your questions are as follows.
>
> Q1: What is the ground-truth weights? For a large training set, it is possible that there may be no such ground-truth weights.
>
> A1: Moreover, the ground-truth weights cannot form a critical point of the empirical risk function. So, this assumption is not reasonable.
> Thank you for the comment. We added the clarification in the revision that we follow a teacher-student setup which has been widely used in the literature. Specifically, the training data are assumed to be generated by a teacher neural network of unknown weights, and the learning is performed on a student network to learn the weights by minimizing the empirical risk of the training data. This teacher-student setup has been studied in the statistical learning community for a long time (Engel & Broeck, 2001; Seung et al., 1992) and has been applied to study neural networks recently (Goldt et al., 2019; Zhong et al., 2017b; a; Zhang et al., 2019; 2020b; Fu et al., 2020; Zhang et al., 2020a).
>
> Q2: The assumption of the Gaussian mixture model is special, not general, and its parameters are assumed to be known a priori. This may be too strict and limits the significance of the result.
>
> A2: Because the learning performance clearly depends on the distribution of input data, some assumption about the input distribution is needed for the theoretical analysis. Most existing works consider standard Gaussian distribution because it is relatively easier to analyze. Extending from the standard Gaussian to the Gaussian mixture model requires developing new analytical tools, as we did in this paper.  There are some other existing works on distributions that are not standard, but there is ample room for new studies about distributions that is not standard Gaussian.  We added the discussion of related work about other input distribution in the paper as follows,
>
> “(Du et al.,2017) considers rotationally invariant distributions, but the results only apply to a perceptron (i.e. a single-node network). (Mei et al., 2018b) analyzes the generalization error of one-hidden-layer networks in the mean-field limit trained on a large class of distributions, including a mixture of Gaussian distributions with the same mean. The results only hold in the high-dimensional region where both the number of neuron K and the input dimension d are sufficiently large, and no sample complexity analysis is provided.  (Li & Liang, 2018) studies the generalization error of over-parameterized one-hidden-layer nets when the data comes from mixtures of well-separated distribution,  but the separation requirement excludes Gaussian distributions and Gaussian mixture models.  (Yoshida & Okada, 2019) analyzes the Plateau Phenomenon of training neural networks that the decrease of the loss slows down significantly partway and speeds up again in one-hidden-layer neural networks with inputs drawn from a single Gaussian with arbitrary covariance.  (Goldt et al., a;b) analyzes the dynamics of learning one-hidden-layer networks with SGD when the inputs are drawn from a wide class of generative models. (Mignacco et al.) provides analytical equations for the size-one minibatch SGD evolution in a perceptron trained on the Gaussian mixture model.  (Ghorbani et al.)  considers inputs with low-dimensional structures and compares neural networks with kernel methods.”
>
> We also added the discussion about how to estimate the parameters of the Gaussian mixture model from the data and the corresponding error bound analysis as follows,
>
> “...The above results assume the parameters of the Gaussian mixture are known. These parameters can be estimated by the EM algorithm (Redner & Walker, 1984) and the moment-based method (Hsu & Kakade, 2013) in practice.  The EM algorithm returns model parameters within Euclidean distance $O((\frac{d}{n})^\frac{1}{2})$ when the number of mixture components $L$ is known. When $L$ is unknown,  one usually chooses some estimate $\bar{L}$ as the number of components, and $\bar{L}$ can be much larger than $L$. In this over-specified setting, the estimation error by the EM algorithm scales as $O((\frac{d}{n})^\frac{1}{4})$. Please refer to (Ho & Nguyen, 2016; Ho et al., 2005; Dwivedi et al., 2020a;b) for details.”

---

> > ### Author Response · Authors · 2020-11-22
> > **Replies to the last comment**
> >
> > Q3: There are some errors in the mathematical denotations like Eq.(4).
> >
> > A3: Thank you. We have corrected (4).

---

### Decision · Program_Chairs · 2021-01-07
**Final Decision**

**Decision:**

Reject

**Comment:**

This paper gives a way to learn one-hidden-layer neural networks on when the input comes from Gaussian mixture model. The main algorithm uses [Janzamin et al. 2014] as an initialization and then performs gradient descent. The main contribution of this paper is 1. to give a characterization of sample complexity for estimating the moment tensors when the input distribution comes from a mixture of Gaussian; 2. to give a local convergence result when the samples come from a mixture of Gaussian. The paper claims certain behavior in the input data would make the problem harder and slow down the convergence, although the claim is based on an upperbound and would be stronger if there is some corresponding lowerbound.

---

> ### Author Response · Authors · 2021-01-15
> **Challenges in obtaining lower bounds as suggested**
>
> Dear Program Chairs,
>
> Although we agree with the comment that our analysis of the sample complexity depends on the upper bound, we need to emphasize that using a lower bound as suggested is either extremely challenging or leads to a trivial bound.
>
> Note that all the existing works in this line of research such as [Zhong et al., 2017], [Zhang et al., 2019], [Zhang et al., 2020a; 2020b], [Fu et al., 2020] consider the upper bound for the sample complexity. All these bounds, including ours, are in the order of  $d\log d$, which is almost order-wise optimal with respect to $d$. Therefore, the related works and our paper use the upper bound to analyze the sample complexity.
>
> One trivial lower bound is $Kd$, which is the number of unknown weights, but this bound does not depend on the input data and thus can not be used to analyze the impact of data. If one wants to find a tighter lower bound for the sample complexity that depends on the input data distribution,  the potential statement, if obtained, shall be something like "for a sample size smaller than this lower bound, no algorithm whatsoever can find $W^*$, as long as the input data follows the Gaussian mixture model," which also means that "if the data follows from a different distribution, then there may exist an algorithm to find $W^*$ with this number of samples. " We find obtaining such a statement is really challenging and would require a completely different set of tools. We feel it is beyond the scope of this paper and will leave this for future work.
>
>  Thank you.
>
> Authors